# Phosphate deprivation restricts bacterial degradation of the marine polysaccharide fucoidan

Yi Xu [1,2,3,4,6], Bowei Gu[1,3,4,6], Huiying Yao[2,6], Mikkel Schultz-Johansen[1,3,4], Isabella Wilkie [4], Leesa Jane Klau[5], Yuerong Chen[1,3,4], Luis H. Orellana[4], Finn Lillelund Aachmann [5], Mahum Farhan[1], Greta Reintjes [1], Silvia Vidal-Melgosa[1,3,4], Dairong Qiao[2], Yi Cao [2] ✉ & Jan-Hendrik Hehemann [1,3] ✉

Brown algae and diatoms convert carbon dioxide into the polysaccharide fucoidan, which sequesters carbon in the ocean despite the prevalence of marine bacterial fucoidanase genes. Bacteria with fucoidanase genes also have high-affinity phosphate transporters, suggesting that phosphate could impact fucoidan degradation and subsequent carbon sequestration. Here, to test this hypothesis, we assembled a system consisting of a microalga that produces and a bacterium that degrades fucoidan. The fixation of carbon dioxide into fucoidan by the microalga *Glossomastix* sp. PLY432 occurred independent of the phosphate concentration. In contrast, the fucoidan-degrading Verrucomicrobiaceae bacterium 227 was inhibited by a lack of phosphate. Degradation of the structurally simpler polysaccharide laminarin was less affected by the phosphate concentration. Phosphate deprivation enabled the fixation of carbon dioxide in fucoidan and disabled its degradation. These conclusions suggest that phosphate deprivation could be a potential strategy to promote the fixation and sequestration of carbon dioxide as fucoidan.

Photosynthetic algae exude substantial quantities of fucoidan that shows potential as a carbon sink[1–4]. Fucoidan is a family of diverse polysaccharides that share alpha-configured and sulfated fucose. During diatom blooms, fucoidan accumulates and forms particles, while the structurally simple polysaccharide laminarin, a beta-glucan, gets degraded rapidly[2]. Within sinking particles, fucoidan can reach deep waters and store carbon in hundred-to-thousand-year-old sediments[3,5]. The stability of fucoidan has been independently supported by incubation experiments. During research ship expeditions, less complex polysaccharides including laminarin are degraded faster than fucoidan by extant microbial communities[6–8]. Moreover, fucoidan degraders are highly specialized bacteria that use hundreds of different fucoidanases to degrade fucoidan[9]. Bacteria with homologue genes of those fucoidanases can be detected during algal blooms locally and globally in the TaraOcean dataset[9]. The presence of fucoidanases indicates a globally pervasive potential for fucoidan degradation by

[1]Faculty of Biology and Chemistry, University of Bremen, Bremen, Germany. [2]Microbiology and Metabolic Engineering Key Laboratory of Sichuan Province, Key Laboratory of Bio-Resource and Eco-Environment of Ministry of Education, College of Life Sciences, Sichuan University, Chengdu, Sichuan, China. [3]Center for Marine Environmental Sciences, MARUM, University of Bremen, Bremen, Germany. [4]Max Planck Institute for Marine Microbiology, Bremen, Germany. [5]Norwegian Biopolymer Laboratory (NOBIPOL), Department of Biotechnology and Food Science, NTNU Norwegian University of Science and Technology, Trondheim, Norway. [6]These authors contributed equally: Yi Xu, Bowei Gu, Huiying Yao. ✉e-mail: cyi@scu.edu.cn; jhhehemann@marum.de

**Fig. 1 | The heterokont microalga *Glossomastix* sp. PLY432 secretes a fucose-rich mucus that accumulates. a**, Phylogenetic relationship of heterokont algae (class level) based on the V4–V5 region of 18S rRNA gene sequences. Numbers in parentheses indicate species contained in collapsed branches. *Saccharomyces* is added as an outgroup. Bootstrap values below 70% are not shown. **b**, Microscopy image showing golden *Glossomastix* cells wrapped in transparent mucus. Similar results were observed in different fields of view across 3 independent replicates. **c**, Decantation of a 3-month-old *Glossomastix* culture reveals the viscous nature of fucose-rich mucus.

bacteria under certain conditions. However, during algal blooms, bacterial fucoidanases are downregulated in the presence of fucoidan[2]. The bacteria primarily express laminarinases and other enzymes for the degradation of simpler polysaccharides[2]. Molecular mechanisms that decrease bacterial synthesis of fucoidan-degrading enzymes and thereby promote the fucoidan carbon sink remain unknown.

Previous research showed that transporters for phosphate acquisition and enzymes for laminarin degradation are highly expressed by bacteria during algal blooms[10]. Hence, bacteria may require phosphate to degrade polysaccharides including laminarin and fucoidan. It has been suggested that the accumulation of algae-derived mucilage polysaccharides occurs during algae blooms when phosphate becomes limiting[11]. Under these conditions, the algae show reduced or absent growth (cell divisions) yet continue carbon dioxide fixation into released polysaccharides. However, this accumulation would also require that bacteria fail to degrade the polysaccharides that are released by algae. This bacterial inability to degrade polysaccharides in response to phosphate limitation has not been tested. To investigate the role of phosphate in fucoidan degradation at the molecular level, we assembled a model system suggesting that phosphate deprivation of microbes could promote the fixation and sequestration of carbon dioxide in the form of fucoidan.

## Results

### *Glossomastix* exudes a fucoidan matrix

Substantial quantities of a polysaccharide are required for purification, structural elucidation, isolation of bacteria and physiological experiments. We needed to identify an alga that reliably produces substantial amounts of fucoidan for our model system. Previous reports showed that *Glossomastix* sp. PLY432 isolated from the English Channel (https://roscoff-culture-collection.org/rcc-strain-details/3688) produces substantial quantities of a fucose containing polysaccharide of unknown structure[12], which we characterized below as a heterofucan belonging to the fucoidan family of polysaccharides. Hence, we chose *Glossomastix* sp. PLY432 as our microalgae for the model system. The *Glossomastix* genus contains one 18S rRNA sequence (GenBank: AF438325.1)[13]. For phylogenetic placement and validation, we sequenced the V4–V5 region of the 18S rRNA gene. Phylogeny, supported by a maximum likelihood analysis, ordered the sequence into the Pinguiophyceae class within the Ochrophyta. The sequence was 96.31% pairwise nucleotide identical to *Glossomastix chrysoplasta* (Fig. 1a). Spherical *Glossomastix* cells of 4.9–8.5-µm size are encapsulated by an extracellular matrix (Supplementary Table 1). Microscopy visualized a transparent, globular

sphere with a radius of 3.4–6.4 µm housing 1 or 2 algal cells (Fig. 1b and Extended Data Fig. 1a). Over time, the culture formed a weak gel, and macroscopic viscosity and microscopic aggregates increased beyond 3 months (Fig. 1c and Extended Data Fig. 1b).

### Phosphate deprivation promotes fucoidan exudation

During 2 months of growth in the medium with organic phosphate, the total glycan carbon content present in the secreted carbohydrates increased 36.42-fold, from 4.14 mg l$^{-1}$ to 150.72 mg l$^{-1}$ (Fig. 2a). The concentration of fucoidan, composed of fucose, rhamnose and galacturonate, increased during the stationary phase (Fig. 2b). Galactose, glucose and glucuronate did not follow this trend (Supplementary Fig. 1). The carbon accumulation rate was 0.52 pg cell$^{-1}$ day$^{-1}$, for fucose 14 µM day$^{-1}$ ($r = 1.00$, $P < 0.001$), for rhamnose 4 µM ($r = 1$, $P < 0.0001$) and for galacturonate 1 µM ($r = 1$, $P < 0.0001$) (Fig. 2c), or ~0.68 pg of fucose, ~0.19 pg of rhamnose and ~0.06 pg of galacturonate. On the basis of the volume and carbon content of diverse microalgae[14–16], we estimate that a *Glossomastix* cell contains 24.49 pg carbon (Supplementary Table 2). Using this figure, we calculated that one cell releases 2.12% of its fixed carbon dioxide in form of fucoidan. *Glossomastix* also continued exuding substantial quantities of fucoidan during the stationary phase when they were growing in the same medium with inorganic phosphate (Fig. 2d).

It has been previously shown that phosphate deprivation can enhance the amount of polysaccharide release by microalgae in the laboratory[17–21] and environmental settings[2,11,22]. Hence, we tested the influence of phosphate concentration on glycans/fucoidan exudation by *Glossomastix*. We restricted access to organic phosphate and found that the algae entered the stationary phase faster (Fig. 2a and Extended Data Fig. 2a–c). Neither the maximum synthesis rate of glycans per cell and day ($r = -0.42$, $P > 0.5$) nor the growth rate ($r = -0.89$, $P > 0.5$) were significantly affected by the decrease in organic phosphate concentration (Fig. 2e and Extended Data Fig. 2d). Moreover, a semi-continuous growth experiment showed that the fucoidan yield per cell was higher in cultures with lower phosphate concentrations after the cultivation cycle ended (Fig. 2f and Extended Data Fig. 3).

### Fucoidan structure

To solve its structure, we purified fucoidan. Detailed fucoidan purification steps are outlined in Methods and Extended Data Fig. 4a. A single broad peak after anionic exchange (AEX) and size exclusion chromatography (SEC) (Extended Data Fig. 4b) in addition to one broad band on a gel (Extended Data Fig. 5c) proved that the fucoidan was purified. An enzyme-linked immunosorbent assay (ELISA) showed

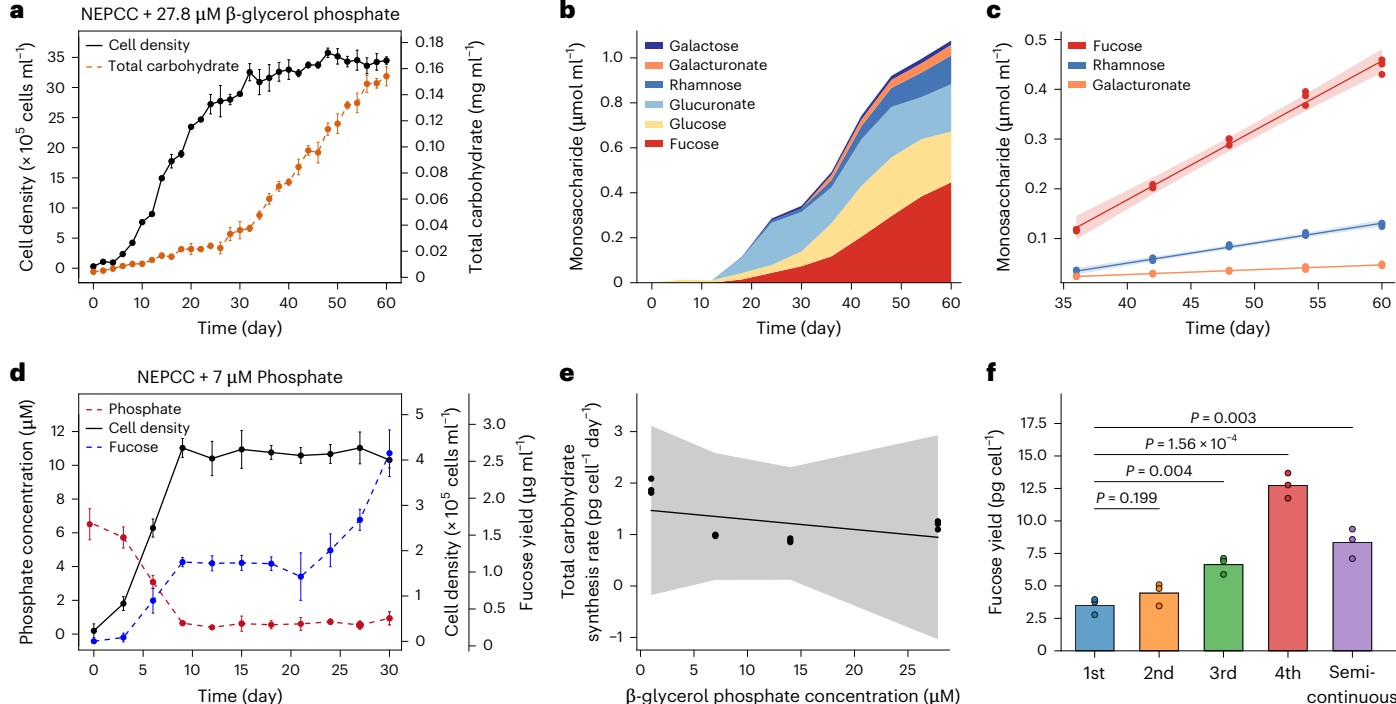

**Fig. 2 | Phosphate deprivation promotes synthesis of fucoidan by *Glossomastix*. a**, Dynamic monitoring of total carbohydrate content and cell density in *Glossomastix* cultures with the addition of β-glycerol phosphate as a phosphorus source. The experiment was performed in independent triplicates ($n = 3$), and data are mean ± s.d. **b**, Monosaccharide composition of exudates collected from *Glossomastix* culture (27.8 μM β-glycerol phosphate as phosphate source). Exudate samples were acid hydrolysed and monosaccharides were quantified by HPAEC-PAD. Each value is the mean of independent triplicates ($n = 3$). **c**, Linear regression analysis for monosaccharides in *Glossomastix* cultures (days 36–60, $n = 3$): fucose (slope = 0.014, $r = 1$, $P = 1.058 \times 10^{-5}$), rhamnose (slope = 0.004, $r = 1$, $P = 9.677 \times 10^{-5}$) and galacturonate (slope = 0.001, $r = 1$, $P = 7.021 \times 10^{-8}$). The shaded area represents the 95% confidence interval of the linear regression. **d**, Monitoring of growth by cell counts, phosphate concentration and fucose content in *Glossomastix* cultures. Inorganic phosphate

($KH_2PO_4$, 7 μM) was used as the phosphate source. Experiments were performed in independent triplicates ($n = 3$) and data are mean ± s.d. **e**, Correlation analysis of net carbohydrate synthesis rates of *Glossomastix* with different concentrations of phosphate ($r = -0.48$, $P = 0.5206$). $P$ values (**c**,**e**) were derived from two-sided Pearson correlation test. Shading indicates the 95% confidence interval around the linear regression line. **f**, Fucose yield of *Glossomastix* in a semi-continuous growth and batch mode. In the 1st batch culture, the starting phosphate concentration was 14 μM. In the 2nd, 3rd and 4th batch cultures and the semi-continuous growth, the actual phosphate concentrations were lower than the theoretical ones (7 μM, 3.5 μM, 1.75 μM and 0.875 μM, respectively). Data were extracted from the last incubation time point for each group of samples (Extended Data Fig. 2b). Experiments were performed in independent triplicates ($n = 3$). Each group of samples was compared with the first batch of samples using two-sided $t$-test.

that monoclonal antibodies (mAbs) BAM1 and BAM2 recognize the *Glossomastix* fucoidan (Fig. 3a). BAM2 binds a sulphated epitope in alpha-configured fucoidan, while BAM1 binds a non-sulfated epitope[23]. BAM2 showed a higher binding signal to both our fucoidan and the positive control fucoidan compared with BAM1 (Fig. 3a and Supplementary Fig. 2).

To know the building blocks of the fucoidan, we used compositional analysis by HPAEC-PAD (high-performance anion-exchange chromatography with pulsed amperometric detection). HPAEC-PAD analysis showed that *Glossomastix* fucoidan is composed (w/w) of 20.41 ± 0.06% fucose, 3.33 ± 0.03% galacturonate, 4.69 ± 0.02% rhamnose and 8.80 ± 0.06% glucuronate. The molar ratio was 44.27% fucose, 10.18% rhamnose, 1.25% galactose, 6.11% galacturonate, 16.15% glucuronate and 21.98% sulfate (Fig. 3b). The composition was reproducible between batches from different years (Extended Data Fig. 4c). The elemental composition (w/w) was 37.41% carbon, 3.29% sulfate and 3.33% nitrogen. Monomer and elemental composition are comparable to fucoidans from diatoms and brown algae[2,24,25] (Fig. 3c). The reproducible composition enabled structure determination by nuclear magnetic resonance spectroscopy.

To add molecular resolution, combinations of NMR spectra including HSQC, H2BC, IP-COSY, TOCSY and HMBC were used to resolve conserved parts of the structure. Desulfation reduced the complexity of the proton NMR spectra[26,27] (Fig. 3d,e). HSQC showed signals in

the anomeric region A ($\delta_H/\delta_C$: 5.29, 103.0), B ($\delta_H/\delta_C$: 5.23, 103.0), C ($\delta_H/\delta_C$: 5.23, 100.7) and D ($\delta_H/\delta_C$: 5.01, 102.2) (Supplementary Fig. 3a and Table 3). Chemical shifts identified monomers and their configuration A: α-D-GlcA*p*; B: α-L-Fuc*p*; C: α-D-GalA*p* and D: β-L-Rha*p*; inter-residue HMBC correlations, verified by NOESY correlations connected monomers with glycosidic bonds. The main chain is a repeating trimer of: -4)-α-L-Fuc*p*-(1-4)-α-D-GalA*p*-(1-2)-β-L-Rha*p*-(1-, showing that it is a heterofucoidan (hereafter, fucoidan) (Fig. 3f). The α-D-GlcA*p* residue was not correlated in HMBC or NOESY spectra, so its position and linkage remain unresolved.

The sulfated structure showed a higher chemical shift for proton (-0.6 ppm) and carbon (-7.4–8.0 ppm) from α-L-Fuc*p* (B'3) at position 3 ($\delta_H/\delta_C$: 4.52, 78.8) and α-D-GalA*p* (C'3) at position 3 ($\delta_H/\delta_C$: 4.69, 80.69), indicating sulfation on both of these monomers (Supplementary Fig. 3b and Table 3). Combined main signals indicate a trisaccharide of -4)-α-L-Fuc*p*(3-SO₃)-(1-4)-α-D-GalA*p*(3-SO₃)-(1-2)-β-L-Rha*p*-(1- (Fig. 3f). Acetyl $CH_3$ groups were observed ($\delta_H/\delta_C$: 2.23, 23.3 and 2.21, 23.1) but not assigned to the trisaccharide. Minor signals indicate less abundant variations to the proposed trisaccharide core structure and potential branching. Variation is consistent with HPAEC-PAD, finding relatively more fucose and rhamnose than galacturonate. Antibodies, chromatography and NMR revealed a structure related to other fucoidans but also with structural differences[9,28].

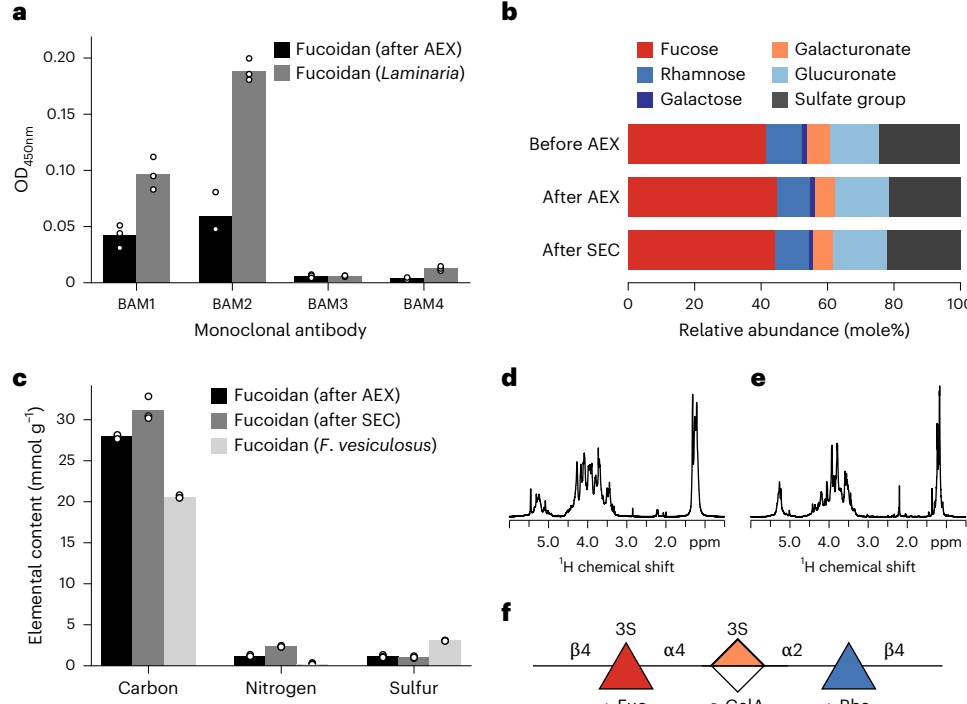

**Fig. 3 | Antibodies, chromatography and NMR present a model structure for the fucoidan sequestration pathway. a**, The binding of four fucoidan-specific monoclonal antibodies (BAM1–4) to AEX-purified *Glossomastix* fucoidan and to fucoidan from the macroalgae genus *Laminaria* (control) was assessed using ELISA. Fucoidan was used at a concentration of 50 µg ml[−1] and the experiment was conducted in 3 technical replicates ($n = 3$). OD, optical density. **b**, Monosaccharide composition of *Glossomastix* exudate (>30 kDa) before and after purification with AEX and SEC. Glucosamine content less than 0.1% is not shown. Monosaccharide profiling was performed with HPAEC-PAD following

acid hydrolysis of fucoidan and data shown are the means of triplicates ($n = 3$). **c**, Elemental analysis of purified *Glossomastix* fucoidan and fucoidan from *Fucus vesiculosus* (F8190, Sigma-Aldrich) in 3 technical replicates ($n = 3$). **d,e**, 1D proton spectrum of purified fucoidan sample (**d**) and desulfated fucoidan (**e**) with water suppression. The samples were dissolved in $D_2O$ (200 µl, 99.96% D), spectrum recorded at 25 °C and 800 MHz, [1]H chemical shift internally referenced to the residual water signal (4.75 ppm). **f**, Structural representation of glycan fragment assigned by NMR spectroscopy of SEC-purified *Glossomastix* fucoidan.

## Isolation of a *Verrucomicrobium* that digests fucoidan

Bacteria with enzymes to digest fucoidan from *Glossomastix* remain unknown. To isolate a fucoidan-degrading bacterium, we used seawater and intertidal pore water from multiple sites along a beach on the island of Helgoland (Extended Data Fig. 5a), where algal blooms and fucoidan accumulation were reported[2]. The enrichment cultures showed growth indicative of fucoidan degradation (Extended Data Fig. 5b,c). We isolated a single colony of Verrucomicrobiaceae bacterium 227 (hereafter V_227) that degraded the fucoidan (Extended Data Fig. 5d,e). When we added nutrients including phosphate but also trace elements and vitamins, V_227 degraded 80% of the fucoidan (Fig. 4a). Another 5% was consumed by the bacterium during the stationary phase. After 122 h, 14.43% ± 3.35% of the fucoidan remained in solution. HPAEC-PAD analysis showed that V_227 consumed 90.69% fucose, 87.51% rhamnose, 43.77% galacturonate, 100% galactose and 73.17% glucoronate, but some of these monomers were consumed only when they were a part of the fucoidan polymer (Fig. 4b). V_227 did not grow with individual fucose, rhamnose or galacturonate. Instead, the fucoidan was imported and digested with enzymes inside the cell (Fig. 4c). Microscopy showed that V_227 incubated with the FLA-fucoidan became fluorescent (Fig. 4d).

## Genome phosphate limitation

DNA sequencing of V_227 gave two contigs, one chromosome with 6.34 Mbp, and a megaplasmid with 0.42 Mbp. Phylogeny showed that V_227 belongs to the Akkermansiaceae family[29]. This family also includes the human commensal *Akkermansia muciniphila* that digests mucin, a fucoidan homologue, which is secreted by human mammalian epithelial cells[30]. Within the family, V_227 belongs to the SW10 clade containing so

far only few sequences. V_227 shares 73.54% average nucleotide identity (ANI) with a metagenome assembled genome (MAG) (WKH.23) from a river in Southern China[31] and is a cultivated member strain within this genus (Fig. 5). Average amino acid identity (AAI) values were also low when comparing V_227 to known members of the SW10 genus (Extended Data Fig. 6a), reinforcing its novelty at the species level. V_227 is the only species carrying two copies of an endo-fucoidanase (GH168), thus probably representing a niche among members of the SW10 genus. Moreover, GH141 (alpha-fucosidase) was detected in other members of the SW10 genus with overall similar genetic contexts, including CBMs and sulfatases in the same genetic contexts (Extended Data Fig. 6b,c).

We used the genome sequence of V_227 to screen for its presence in global circumnavigation TARA and Helgoland spring bloom datasets from the surface ocean. The depth and breadth of coverage were low in the queried datasets: for example, <0.5× and <3% (Fig. 5). The relatively low abundance of Verrucomicrobiota including their sulfatases, fucosidases and fucoidanases[9] remains intriguing considering that fucose-containing glycans hold up to 20% (~20 µM) of the dissolved organic carbon in the surface ocean[32,33]. This relatively high abundance of potential substrates and low abundance of Verrucomicrobiota—the phylum with members specialized for the consumption of fucoidans in the ocean[29]—may be related to phosphate concentration. Especially in oligotrophic regions of the surface ocean with a phosphate concentration of 0–300 nM[32,34,35], and during later phases of coastal algal blooms, phosphate can play an important role in restricting bacterial growth[36], although the degree of phosphate limitation varies among systems.

The distribution and types of gene in the chromosome and plasmid indicate that glycan digestion is coupled to phosphate acquisition. The genome contains genes for the tricarboxylic acid

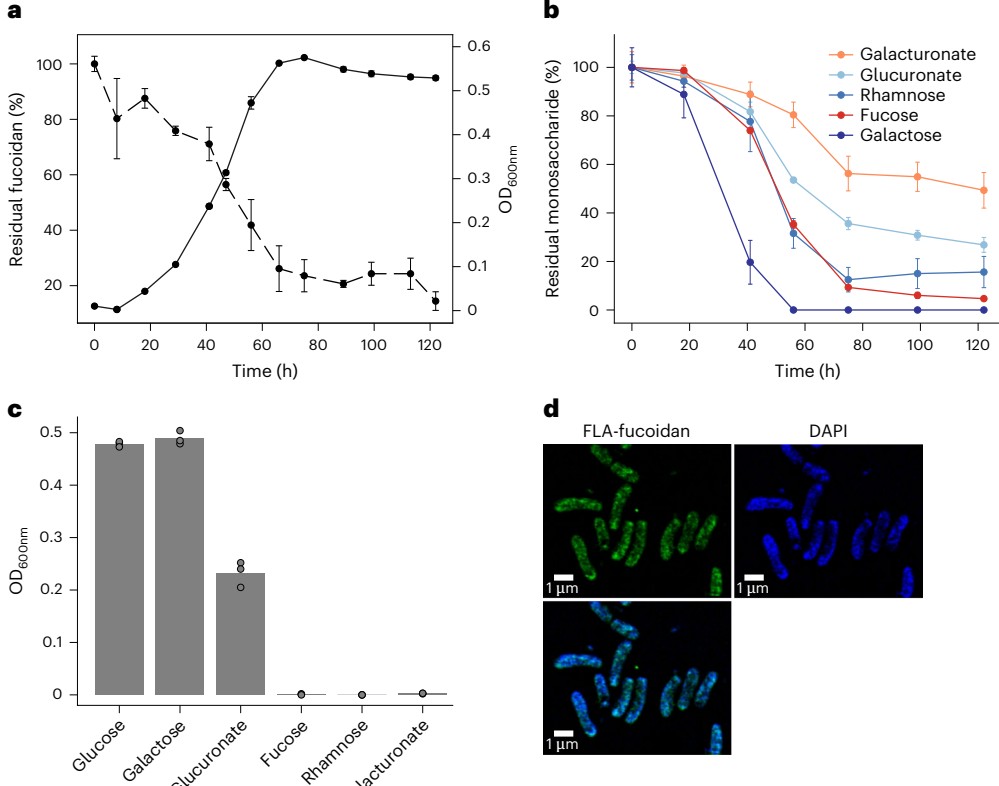

**Fig. 4 | Selfish mode of fucoidan digestion indicates that Verrucomicrobiaceae bacterium 227 (V_227) is adapted to nutrient limitation. a**, Growth of V_227 in MMT-YE medium with 0.05% (w/v) fucoidan (solid line) and the evolution of relative content of fucoidan in the culture supernatant (dashed line). The experiment was performed in independent triplicates ($n = 3$). Data represent mean ± s.d. **b**, Concentrations of total fucose, rhamnose, galacturonate, glucuronate and galactose in the culture supernatant over bacterial growth. The experiment was performed in independent triplicates ($n = 3$) and data are mean ± s.d. **c**, Growth of V_227 with different monosaccharides as substrates. The experiment was performed in independent triplicates ($n = 3$). **d**, Selfish uptake of fucoidan by V_227. Super-resolution images of V_227 cells on day 3. DAPI (blue) signal shows nucleic acid. FLA-probe (green) signal is released from fluoresceinamine-labelled fucoidan. Merged DAPI and FLA-probe signal is shown at the bottom. Similar results were observed in different fields of view across 3 independent replicates.

(TCA) cycle, and some genes for ATP synthase are on the plasmid (Supplementary Tables 4 and 5). We identified 208 carbohydrate active enzymes and sulfatases and found both enzymes on chromosome and plasmid (Supplementary Table 4). The annotations are consistent with glycans digested by V_227 (Supplementary Fig. 4). Transporters for the acquisition of phosphate and other nutrients are located on the chromosome and plasmid. The genome contains 206 genes annotated as membrane transporters, including 8 TonB-dependent receptors. TonB-dependent transport proteins require energy to import ferric chelates, vitamin $B_{12}$, nickel complexes, peptides and oligosaccharides[37]. A total of 12 major facilitator superfamily transporters were predicted to import oligosaccharides, monosaccharides and amino acids[38]. The chromosome also contained 3 $Na^+$/phosphate symporters (NptA) with low affinity for nutrient-rich environments[39,40]. For phosphate-limited environments, there are two homologues of the high-affinity ABC-type phosphate transport system composed of four proteins in an operon (PstS,A,B,C). One operon is located on the chromosome and the other on the plasmid. One regulatory protein, PhoU, for the regulation of the PstSABC proteins, may regulate the two operons (Supplementary Table 6). Notably, homologues of this high-affinity system are present in 211 of the 246 genomes shown in the tree (Fig. 5 and Supplementary Table 7), including the intestinal *Akkermansia muciniphila*, suggesting that phosphate limitation may be common.

## Phosphate deprivation restricts bacterial degradation of fucoidan

To test whether phosphate has an effect, we increasingly reduced the concentration and monitored the growth rate with laminarin or

*Glossomastix* fucoidan as sole carbon source. Due to its biogeochemical and ecological relevance[41], we used the less complex laminarin as a control glycan. For laminarin, the growth rate remained relatively constant at 0.15 h$^{-1}$ between 1 μM and 50 μM phosphate concentration ($r = -0.76$, $P < 0.05$). With fucoidan, the growth rate was overall lower and decreased linearly from 0.13 h$^{-1}$ at 50 μM to 0.06 h$^{-1}$ at 1 μM phosphate concentration ($r = 0.98$, $P < 0.001$) (Fig. 6a). At 50 μM phosphate concentration, both glycans enabled similar growth rates, and we stopped increasing the concentration as the growth rates plateaued. The convergence of growth rates shows that the effect is not detectable in bacterial growth media that have higher phosphate concentration. The lowest phosphate concentration of 1 μM tested here is still higher than that in nutrient-limited algal blooms[42,43]. Yet, at 1 μM phosphate, the mean growth rate was 2.98-fold slower with fucoidan than with laminarin. Intrigued by this observation, we tested two other glycans, pectin and xylan (Extended Data Fig. 7). Similar to that in laminarin, growth rate in pectin and xylan did not decrease together with the decreasing phosphate concentration. For unknown reasons, the data show that only fucoidan significantly slowed down the bacteria compared to laminarin and other glycans when the cells were phosphate starved.

Next, we investigated the relationship between biomass yield and phosphate concentration. Growth yield curves showed that biomass decreased with the phosphate concentration for laminarin and fucoidan. However, at phosphate concentrations of 10 μM and lower, the fucoidan significantly decreased the biomass compared to laminarin (Fig. 6b and Extended Data Figs. 8 and 9). With fucoidan at 1 μM phosphate, we counted $(1 \pm 4) \times 10^6$ cells ml$^{-1}$ (linear model: $1.88 \times 10^6$), and with

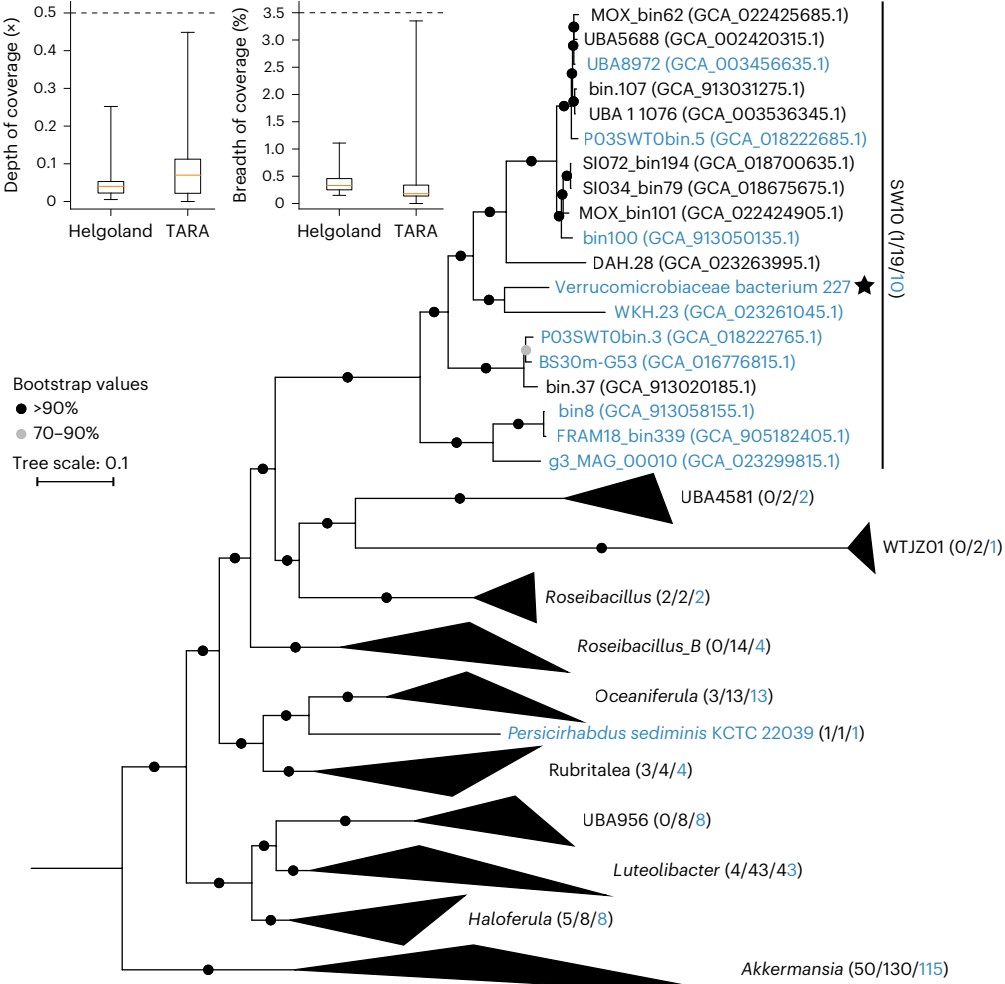

**Fig. 5 | High-affinity phosphate transporters common in glycan-utilizing Akkermansiaceae reveal that they adapt to environments with nM concentration of phosphate.** The global abundance of V_227 in the ocean. Coverage depth (left) and breadth (middle) in the TARA dataset ($n$ = 157) and Helgoland spring bloom dataset ($n$ = 68). Area below the dashed line indicates very low abundance in the dataset. Boxplot shows the median (centre), the 25th and 75th percentiles (bounds of box), and the minimum and maximum values (whiskers). Phylogenetic tree (right) based on 120 conserved genes from 246 selected Akkermansiaceae genomes. Parentheses show number of culturable and number of total samples. The blue colour indicates that the genome contains the high-affinity phosphate transport system (PstSABC), or the number of genomes containing this PstSABC system on each clade. Verrucomicrobiaceae bacterium 227 (V_227) is the only culturable bacterium in the genus SW10.

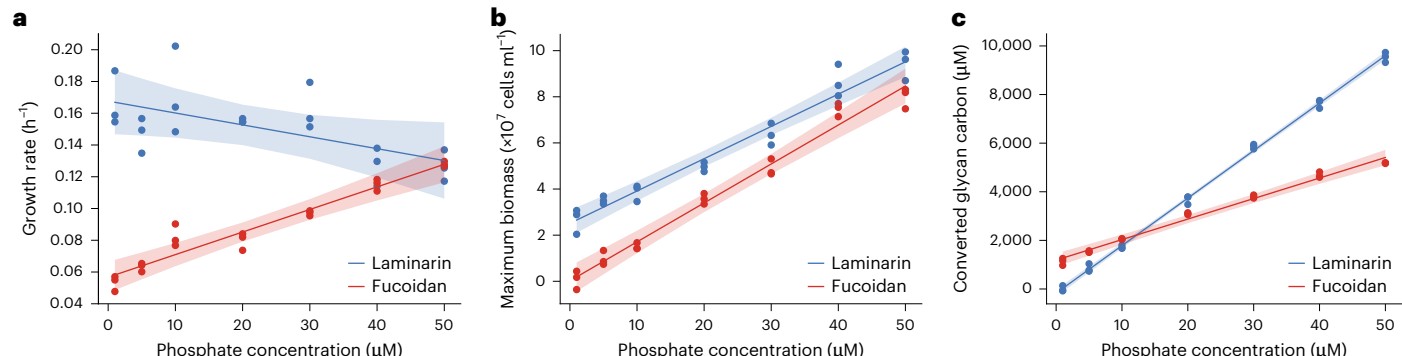

**Fig. 6 | Phosphate deprivation restricts bacterial digestion of fucoidan.**
**a**, Correlation analysis between growth rate and phosphate concentration on laminarin ($r$ = −0.76, $P$ = 0.04786) and fucoidan ($r$ = 0.98, $P$ = 0.0001361). Fitting was performed using the mean growth rate. **b**, Correlation analysis between maximum biomass and phosphate concentration on laminarin ($r$ = 0.99, $P$ = 1.033 × 10$^{-5}$) and fucoidan ($r$ = 0.99, $P$ = 7.915 × 10$^{-6}$). Fitting was performed using the mean of maximum biomass. **c**, Correlation analysis between converted glycan carbon and phosphate concentration on laminarin (slope = 195.46, $r$ = 1, $P$ = 2.288 × 10$^{-9}$) and fucoidan (slope = 84.43, $r$ = 0.99, $P$ = 3.457 × 10$^{-6}$). Fitting was performed using the mean consumption of polysaccharide carbon. The experiment was performed in independent triplicates ($n$ = 3). $P$ values were derived from two-sided Pearson correlation test. Shading indicates the 95% confidence interval around the linear regression line.

laminarin, we counted $(27 \pm 6) \times 10^6$ cells ml$^{-1}$ (linear model: $26.54 \times 10^6$), a 27-fold difference (linear model: 14.13) (Supplementary Table 8). Using the 14.13-fold difference from the linear model and assuming that the phosphate was quantitatively converted into bacterial biomass, a cell growing with fucoidan contains 0.53 fmol and a cell with laminarin contains 0.04 fmol of phosphate (Supplementary Table 8). These numbers fall within the range measured for different phyla of heterotrophic bacteria from different environments and in different growth phases[44]. Using the conservative calculation, fucoidan requires roughly 10× more phosphate to support the same number of cells. The converse conclusion is that cells grown with laminarin can allocate 10× more phosphate to replication and other biochemical reactions that require phosphate. The question remains whether phosphate starvation stabilizes fucoidan and other complex glycans.

Quantitative glycan carbon accounting shows that fucoidan is more stable with limited phosphate. To measure the conversion of glycan carbon relative to phosphate, we quantified the glycan parts that remained in a given phosphate concentration. Residual fucoidan and laminarin carbon was quantified by HPAEC-PAD analysis of their constituent monosaccharides. The initial glycan carbon concentration minus the residual glycan carbon was used to calculate how much glycan carbon was converted per phosphate by the bacterium. The data show a fixed ratio of glycan carbon converted per phosphate. For laminarin carbon, the conversion ratio was C:P = 195.46:1 ($r = 1$, $P < 0.0001$) (Fig. 6c). In the case of fucoidan, the conversion ratio was C:P = 84.43:1 ($r = 1$, $P < 0.0001$). The data indicate that for both glycans, carbon above the linear ratio line remained stable and carbon below this line was converted. To verify these results, we tested pectin and xylan and found that growth yield and carbon conversion for all four tested glycans were constrained by the amount of phosphate (Extended Data Fig. 7). Comparing growth yield (Fig. 6b) with carbon conversion (Fig. 6c) shows that the bacterium converts significantly more laminarin than fucoidan carbon to yield comparable bacterial biomass. Theoretically, glycolysis of glucose and galactose before the TCA cycle generates 2 ATPs compared to only 1 ATP from fucose, rhamnose, glucuronate and galacturonate. Glucuronate and galacturonate yield 0 NADH where the other monomers yield two (Extended Data Fig. 10). Given that more laminarin than fucoidan is required to build similar quantities of bacterial biomass, some of the carbon energy of laminarin is lost. The TCA cycle rapidly extracts energy and releases carbon dioxide from glucose, which may account for the lost carbon. Compared to laminarin, fucoidan is twice as stable and therefore of superior quality to store carbon.

## Discussion

Combined results show that glycan types have different qualities for carbon sequestration, and we provide ways to quantify this quality. The linear relationship of converted glycan carbon per phosphate indicates a structure-specific stability ratio (converted carbon atoms/phosphate molecule). The ratio may change, for example, in a dynamic microbiome, where diverse bacteria compete for the same inorganic nutrients[45] that consequently fluctuate and sometimes restrict or stimulate growth, and by extension the digestion of complex glycans[46]. Complex glycans are synthesized on the outer membrane of eukaryotic cells where they form the extracellular matrix. Algae and other eukaryotes have evolved to increase their glycan structural complexity as a survival strategy in diverse environments, including oceans, soils, intestines and the rhizosphere. In these environments, diverse types of cell including prokaryotes compete for the limited amount of phosphate[47], which is required for growth and to digest complex glycans. The ratio might be useful to quantify the stability of glycans in the presence of microbes for carbon sequestration, as prebiotics, as materials and as emerging drugs.

Aligned growth rates between eukaryotes and prokaryotes are a fundamental tenet for sustainable co-existence of species, a balance easily disrupted by resource imbalance. Bacteria have smaller cells with more surface area for receptors and transporters to obtain nutrients from the environment[48]. Algae compete with heterotrophic bacteria for nutrients and are at a disadvantage except for carbon dioxide. Support for this hypothesis comes from nutrient fertilization experiments conducted with ships and mesocosms across the ocean. Ocean fertilization with nutrients can stimulate algal blooms[22,49]. However, experiments in the Arctic[50], Mediterranean[36] and Southern Atlantic Ocean[51] found that addition of glucose, phosphate or iron stimulated heterotrophic bacteria. The scientists who conducted the experiment in the Arctic proposed that the addition of glucose enabled heterotrophic bacteria to consume limited nutrients faster than algae, shifting the system away from photosynthesis and carbon capture towards heterotrophy and release of carbon dioxide. Ocean and laboratory experiments indicate that the carbon to nutrient ratio as well as the complexity of the molecule that provides carbon energy can tune the balance between heterotrophic bacteria and algae.

So how do algae persist and compete for nutrients given that heterotrophic bacteria such as SAR11 are more abundant, resulting in a large surface area covered by nanomole affinity transporters for the uptake of inorganic nutrients[48]? Earlier studies of quantitative glycan accounting found that algae-derived, fucose-containing glycans of unknown structure are stable for years in the global ocean surface. These glycans hold 20% (~20 μM) of the dissolved organic carbon in this system, which is largely phosphate deprived (0–300 nM)[32–35]. This persistence is in line with nutrients such as phosphate being required for the degradation of such complex glycans[52–54].

We found that from a microalgal perspective, the fucoidan can, relative to the more labile laminarin, reduce bacterial growth rate and their cell yield. Hence, an extracellular matrix composed of fucoidan could give algae a competitive advantage by restricting bacterial growth and thereby providing microalgae with more time to access limited nutrients. In contrast, bacteria that consume structurally more labile carbon compounds[55] would rapidly consume essential nutrients. These nutrients then become limiting for the slower fucoidan-degrading bacteria, decreasing their ability to digest the extracellular matrix composed of fucoidan. In this way, the limited nutrients contribute to stabilizing the fucoidan matrix around the algal cells, which physically excludes and biochemically constrains bacteria. Consequentially, when fucoidan is the primary carbon source in the extracellular matrix, it can increase the space and time available for algae to access limited nutrients. We conclude that fucoidan may contribute to the competitive nature of these algae, which could contribute to their potential to sequester carbon in the ocean.

## Methods

### Cultivation of *Glossomastix*

The microalgal strain, *Glossomastix* sp. PLY432, was obtained from the Roscoff Culture Collection (RCC3688) and cultivated in NEPCC medium (MediaDive: 1724)[56] at 15 °C, with 140 μmol m$^{-2}$ s$^{-1}$ photosynthetic photon flux density provided by cool-white fluorescent lamps under a 12 h/12 h light/dark cycle. Pre-cultures grown for 2 weeks were used to inoculate fresh NEPCC medium. For large-scale cultures, 1 l culture was grown in 2 l Fernbach flasks with 20 ml pre-culture. For small-scale cultures, 150 ml NEPCC medium in 250 ml cell culture flasks was inoculated with pre-culture to an initial concentration of $3.05 \times 10^4$ cells ml$^{-1}$. Growth and carbohydrate production of *Glossomastix* was monitored over 60 days by sampling 2 ml of the mixed culture every second day, or every fourth day when *Glossomastix* was grown in phosphate-limited medium. Phosphate-limited NEPCC medium was obtained by adjusting the final concentration of β-glycerol phosphate. The initial concentration of algal cells in phosphate-limited cultures was $2.25 \times 10^4$ cells ml$^{-1}$. Cell counting was performed with a Neubauer haemocytometer (Marienfeld-superior, 0640111) using 10 μl of culture. Total carbohydrate was determined using 200 μl of culture as described below. The remainder of the 2-ml samples was stored at −20 °C

for total monosaccharide determination. After 3 months of growth, *Glossomastix* cultures were sampled from the top and bottom part of the culture flask to assess low and high viscosity fractions, respectively. Cell morphology was examined with an EVOS FL Auto Microscope (Thermo Fisher) and the size of mucus layers was evaluated manually using ImageJ[57].

For the semi-continuous growth experiments with inorganic phosphate, $KH_2PO_4$ was used instead of β-glycerol phosphate as the phosphate source. Pre-cultures were grown for 1 week and inoculated at 5% (v/v) into fresh NEPCC medium supplemented with 14 μM $KH_2PO_4$ (100 ml + 5 ml) as Batch 1. Each treatment was set up in triplicate. Batch 1 was cultured for 6 days until exponential growth. At this stage, 50 ml of culture was transferred to Batch 2 (50 ml fresh medium without phosphate) while retaining the remainder for continued cultivation. For sampling, 1 ml of cultures was collected for cell counting (10 μl), phosphorus analysis (0.5 ml filtered, −20 °C) and fucose quantification (remainder, −20 °C). Subsequent batches (2–5) followed an identical 3-day transfer protocol: each time, 50 ml from the previous batch was transferred to fresh medium (maintaining a 1:1 dilution) while continuing cultivation of the remaining culture, with consistent sampling at each transfer point.

## Phylogenetic analysis of *Glossomastix*

*Glossomastix* sp. PLY432 genomic DNA was extracted using the DNeasy Blood and Tissue kit (Qiagen). The V4–V5 region of the 18S rRNA gene was amplified from purified genomic DNA using the 18S universal primers (574F-CGGTAAYTCCAGCTCYAV and 1192R-CAGGTGAGTTTTCCCGTGTT) in Q5 High-Fidelity 2× Master Mix according to manufacturer instructions[58,59]. PCR products were then purified with the QIAquick PCR Purification kit (Qiagen) and sequenced at Eurofins Genomics. The 18S rRNA genes of 34 species from 5 classes of the Ochrophyta (Pinguiophyceae, Eustigmatophyceae, Bacillariophyceae (Diatom), Phaeophyceae (Brown algae) and Xanthophyceae) were obtained from NCBI and used to construct a phylogenetic tree along with the 18S rRNA gene of PLY432. The 18S rRNA gene of six species from *Saccharomyces* was likewise retrieved and used as an outgroup. Details of the genes are listed in Supplementary Table 9. Sequences were aligned using MUSCLE (v.3.8.31) in MPI Bioinformatics Toolkit, with non-aligned regions removed[60–62]. The phylogenetic tree was constructed via IQ-TREE 2 (ref. 63) under the automatic optimal model selection (Tne+I+G4), calculated with 1,000 bootstrap replications and visualized with TVBOT[64].

## Total carbohydrate quantification

The total carbohydrate content in *Glossomastix* cultures was measured continuously during growth using the phenol–sulfuric acid method[65]. Briefly, 0.2 ml sample, 5% phenol and concentrated sulfuric acid were combined in a 2-ml tube (1:1:5) and gently mixed. After 10 min at room temperature, the samples were placed in a water bath at 30 °C for 20 min. The content of each tube was cooled to r.t. and transferred to cuvettes, and the absorbance of each sample was then detected at optical density $(OD)_{490}$ using a BioSpectrometer (Eppendorf AG), or 100 μl of sample in a 96-well plate was read using the SpectraMax iD3 plate reader. A calibration curve was constructed by analysing the linear relationship between the concentration and the $OD_{490 nm}$ of the 99.99% standard glucose stock in the range of 0.02–0.5 mg ml⁻¹.

## Phosphate quantification assay

Phosphate concentration was quantified using a colorimetric assay modified from ref. 66. Briefly, a standard curve was generated using $KH_2PO_4$ solutions ranging from 0 to 40 μM. In a 48-well plate, 200 μl of freshly prepared reagent (a mixture of 10% ascorbic acid, 2.5% ammonium molybdate, 6 N sulfuric acid and deionized water at the ratio of 1:1:1:2) was added to 200 μl of each phosphate standard or sample. The plate was sealed with parafilm and incubated at 37 °C with shaking at 500 r.p.m. for 1.5 h. Absorbance was recorded at 820 nm via a

SpectraMax iD3 plate reader and phosphate concentrations of samples were calculated on the basis of the standard curve. This assay has a detection limit of ~1 μM, below which phosphate concentrations cannot be accurately quantified.

## Deoxy-sugar quantification assay

The quantification of deoxy-sugars was performed using an L-cysteine assay modified from refs. 67,68. For the assay, a standard curve was prepared using L-fucose solutions ranging from 0 to 10 μg ml⁻¹ in ultra-pure water. Samples (100 μl) and standards were aliquoted into 1.5 ml microcentrifuge tubes, followed by the addition of 234 μl 97% $H_2SO_4$. The tubes were allowed to cool to r.t. before being heated at 99 °C for 10 min with shaking (800 r.p.m.). After rapid cooling on ice, 10 μl 3% (v/v) L-cysteine HCl was added to each tube. The samples were vortexed and incubated in the dark at r.t. for 90 min. Absorbance measurements were taken at 396 nm and 427 nm using a microplate reader in a 96-well plate. The difference in absorbances (Abs 396–427) was calculated, and deoxy-sugar concentrations were determined on the basis of the standard curve. To ensure accuracy, all measurements were performed in triplicate and a calibration curve with an $R^2 > 0.99$ was used for quantification.

## Extraction of total polysaccharides from *Glossomastix*

After 2 months of cultivation, EDTA was added to 1 l *Glossomastix* culture (50 mM final concentration) and autoclaved (121 °C, 15 min). Whatman glass microfibre filters (Grade GF/F, 0.70 μm) and Millipore Express PLUS membranes (0.22 μm) were used to separate polysaccharide fractions from the supernatant sequentially. The filtered culture supernatant was concentrated on an Amicon stirred cell (Millipore) equipped with a 30-kDa ultrafiltration membrane to collect the high molecular weight (HMW) polysaccharide fraction. The concentrate was continuously dialysed with ultra-pure water (UPW) until the conductivity of the filtrate no longer changed. Conductivity was measured using the SevenCompact Duo S213 meter. The final concentrate was collected and made up to 100 ml with UPW, followed by stirring overnight to detach polymers from the membrane. The desalted samples were lyophilised and stored at r.t. until further analysis. Further separation of high-purity polysaccharides was carried out using AEX and SEC (Extended Data Fig. 4a).

## AEX

Crude polysaccharide extracts were further concentrated and purified by anion-exchange chromatography on an XK 26/40 column packed with 90 ml ANX FF resin. The packed column attached to an ÄKTA pure system was first equilibrated with Tris-HCl buffer (50 mM, pH 7.5, degassed) at 5 ml min⁻¹, followed by sample application. Crude sample in 100 ml Tris-HCl buffer (1 g l⁻¹) was filtered (0.22 μm) to remove insoluble materials and then loaded onto the equilibrated resin. Following sample injection, the column was washed with two column volumes of Tris-HCl buffer to remove unconsolidated fractions, followed by two column volumes of Tris-HCl buffer containing 0.5 M NaCl. Finally, fraction collection started immediately using two column volumes of Tris-HCl buffer containing 5 M NaCl as elution buffer to wash the column. The eluates were concentrated and desalted via Amicon stirred cells with 30-kDa ultrafiltration membrane as described above, and lyophilised.

## SEC

Final purification of polysaccharides was performed using size exclusion chromatography on two HiPrep 16/60 Sephacryl S-400 HR (120 ml per column) connected in series to a Knauer FPLC system (Azura Bio Purification System) equipped with a refractive index detector. Lyophilised sample (100 mg after AEX) was dissolved in 2 ml Tris-HCl buffer, filtered (0.22 μm) and loaded onto the columns that were equilibrated with 300 ml Tris-HCl buffer (50 mM, pH 7.5, degassed) at a rate of

1 ml min⁻¹ before sample injection. Columns were eluted with 300 ml Tris-HCl buffer and the polysaccharide-containing fractions were pooled, desalted and concentrated via Amicon stirred cells as above.

## Quantification of monosaccharides with HPAEC-PAD

For the quantification of monosaccharides, samples were analysed on an ICS-5000+ system (Dionex) with pulsed amperometric detection (PAD) equipped with a CarboPac PA10 analytical column ($2 \times 250$ mm) and a CarboPac PA10 guard column ($2 \times 50$ mm)[69]. In brief, 200 µl lyophilised pure polysaccharide samples (1 mg ml⁻¹) or 200 µl microalgae culture (with cells) were hydrolysed with 200 µl 2 M HCl at 100 °C for 24 h in a pre-combusted (450 °C, 4 h) vial. Supernatants from 100 µl bacterial cultures were hydrolysed with 100 µl 2 M HCl. After complete acid hydrolysis, 100 µl *Glossomastix* culture samples were dried by speed vacuum to remove HCl and then resuspended in 100 µl UPW, followed by a 1:100 (v/v) dilution. The other samples were diluted with UPW at a ratio of 1:200 (v/v) and then centrifuged at 14,800 r.p.m. ($\sim21,000 \times g$) (Thermo Scientific Fresco 21 microcentrifuge) for 10 min. Note that after acid hydrolysis, *Glossomastix* culture samples were dried by speed vacuum and then resuspended in 100 µl UPW, followed by a 100-fold dilution. Subsequently, 100 µl supernatant was analysed by direct injection onto the HPAEC-PAD system. Monosaccharide standard (Supplementary Table 10) mix ranging from 1–10 to 1,000 µg l⁻¹ was used to identify peaks by retention time and to construct standard curves for quantifying the amount of monosaccharide products in the reaction mixture.

## Desulfation of fucoidan

A complete desulfation of fucoidan was conducted using a modified version of the solvolytic desulfation protocol outlined in ref. 70. First, sodium cations were exchanged with pyridinium ions by dissolving 20 mg of fucoidan in water and passing the solution through an AG 50 W cation exchange resin (Bio-Rad) pre-equilibrated with pyridine (Sigma-Aldrich). The eluate was neutralized with 0.3 ml pyridine and subjected to lyophilisation. Subsequently, the fucoidan–pyridinium salt was dissolved in 15 ml of DMSO (Sigma-Aldrich), and 75 µl of UPW was introduced. The mixture was incubated at 80 °C for 30 min and then subjected to dialysis (8,000 molecular weight cut-off) against 1 M NaCl and UPW before lyophilisation.

## NMR characterization of *Glossomastix* fucoidan

SEC-purified fucoidan is used only for structural and elemental analysis. Purified fucoidan and desulfated fucoidan (10 mg) were first dissolved in 1 ml 99.9% $D_2O$ (Sigma-Aldrich) and lyophilised to reduce the residual water signal. Subsequently, the samples were dissolved in 200 µl $D_2O$ (D-99.96%; Sigma-Aldrich) and transferred to a 3 mm LabScape Stream NMR tube (Bruker LabScape). For NMR analyses, all homo- and heteronuclear experiments were acquired on a Bruker AV-IIIHD 800 MHz spectrometer (Bruker BioSpin) equipped with a 5 mm cryogenic CP-TCI z-gradient probe. Chemical shifts were calibrated using the residual water signal for ¹H (4.75 ppm at 25 °C). The ¹³C chemical shift was internally referenced to DSS (4,4-dimethyl-4-silapentane-1-sulfonic acid) using the absolute frequency ratio[71] (¹³C/¹H = 0.251449530). For chemical shift assignment, the following one- and two-dimensional NMR experiments were recorded at a temperature of 25 °C: 1D proton with water suppression (zgesgp), ¹H-¹³C HSQC with multiplicity editing (hsqcedetgpsisp2.3), ¹H-¹³C heteronuclear two-bond correlation (H2BC) spectroscopy (h2bcetgpl3pr), ¹H-¹³C heteronuclear multiple bond coherence (HMBC) with suppression of one-bond correlations (hmbcetgpl3nd), ¹H-¹H in-phase correlation spectroscopy (IP-COSY) with water suppression with excitation sculpting (ipcosyesgp-tr)[72], ¹H-¹H total correlation spectroscopy (TOCSY) with 70 ms mixing time and water suppression (clmlevphpr), ¹H-¹³C HSQC-TOCSY with 70 ms mixing time protons (hsqcdietgpsisp.2), ¹H-¹H nuclear Overhauser effect spectroscopy (NOESY) with 80 ms mixing time and water suppression

with excitation sculpting and gradients (noesyesgpph). The spectra were recorded, processed and analysed using the TopSpin 3.5 or 4.0.1 software (Bruker BioSpin).

## Sulfate quantification

The released sulfate from acid-hydrolysed polysaccharide was measured on a Metrohm 761 compact ion chromatograph equipped with a Metrosep A Supp 5 column and suppressed conductivity detection with 0.5 M $H_2SO_4$. Ions were separated by an isocratic flow of carbonate buffer (3.2 mM $Na_2CO_3$, 1 mM $NaHCO_3$) and the duration of each run was 20 min, with sulfate eluting at 16 min. Chromatograms were analysed with the instrument's software MagIC Net v.3.2.

## Elemental analysis

Elemental analysis was performed using an Elementar modern elemental analyser. Lyophilised sample (0.1–1.0 mg) was transferred to a dry and pre-weighed tin boat, and a small amount of tungsten oxide was added. For the calibration curve, sulfanilamide (0.1–1 mg) was prepared in the same way. Before each test, the samples were degassed by compressing the tin boats.

## ELISA

Enzyme-linked immunosorbent assay was used to assess the binding of four fucoidan-specific rat mAbs, namely BAM1, BAM2, BAM3 and BAM4 (ref. 23) from SeaProbes (https://www.sb-roscoff.fr/en/seaprobes), to *Glossomastix* fucoidan purified by AEX. The purified fucoidan was dissolved in water and underwent dilution in PBS to 200 µg ml⁻¹, followed by five 2-fold dilutions in PBS, resulting in a total of 6 concentrations. Each antibody–fucoidan combination was tested in triplicate. For the ELISA, 100 µl of the fucoidan solution were added to wells of a 96-well plate (NUNC MaxiSorp, Thermo Fisher). After overnight incubation at 4 °C, wells were washed six times with tap water, and unbound sites were blocked with 200 µl PBS buffer (137 mM NaCl, 2.7 mM KCl, 10 mM $Na_2HPO_4$, 1.7 mM $KH_2PO_4$, pH 7.4) containing 5% (w/v) low-fat milk powder (5% MPBS) for 2 h at r.t. After washing nine times with tap water, 100 µl of the mAbs diluted 1:10 in 5% MPBS were added to each well and incubated for 1.5 h at r.t. After washing nine times with tap water, 100 µl of anti-rat secondary antibody conjugated to horseradish peroxidase (A9037, Sigma-Aldrich) diluted 1:1,000 in 5% MPBS were added to each well and incubated for 1.5 h at r.t. Wells were washed with tap water nine times. The plate was developed by adding 100 µl ELISA tetramethylbenzidine (TMB) substrate per well. The enzyme reaction was stopped after 5–10 min by addition of 100 µl 1 M HCl to each well. Absorbance at 450 nm (mAb binding intensities) was measured with a SPECTROstar Nano absorbance plate reader using the MARS software (BMG Labtech). Fucoidan from *Laminaria* (Glycomix, PSa13) was used as a positive control. Appropriate negative controls were run on every plate.

## Media and monitoring of growth

Unless otherwise stated, bacteria were cultivated in marine minimal Tris-HCl medium[73] supplemented with vitamins[74] and iron. The base minimal medium (MMT) contained 2.3% (w/v) sea salts (S9883, Sigma-Aldrich), 9 mM $NH_4Cl$, 26 mg l⁻¹ ammonium ferric citrate (F5879, Sigma-Aldrich), 50 mM Tris-HCl (pH 7.8) and 1× vitamin mix (111,000× vitamin mix: 10 mg biotin, 10 mg lipoate, 50 mg Ca-D-pantothenate, 50 mg vitamin B12, 100 mg nicotinate, 100 mg pyridoxamine dihydrochloride, 100 mg thiamine hydrochloride, 40 mg aminobenzoate and 30 mg folate). MMT was further supplemented with 0.03% (w/v) yeast extract (MMT-YE), 0.03% (w/v) casamino acids (MMT-CA) or 50–100 µM $KH_2PO_4$ (MMT-KDP). At a concentration of 0.03% (w/v) Bacto casamino acids, the medium already contains ~48.9 µM Pi[75]. All media were sterilized by 0.22 µm filtration. Unless otherwise stated, bacteria were grown at r.t. (400 r.p.m.) in 24-well plates (Sarstedt, 83.3922.500), and growth was monitored as $OD_{600}$ using a SpectraMax iD3 plate reader (Molecular Devices).

## Bacteria and growth conditions

'*Lentimonas*' sp. CC4 was obtained previously[9] and *Wenyingzhuangia fucoidanilytica* CZ1127[T] (DSM 100787) was purchased from DSMZ (German Collection of Microorganisms and Cell Cultures). Bacteria were isolated from non-axenic *Glossomastix* by plating 1:10[4] diluted microalgal culture onto Difco Marine Agar 2216. Single colonies were obtained from plates after incubation at 15 °C for 12 days. All bacteria were cultured in MMT-CA medium containing 0.05% (w/v) glucose for 5 days, and then inoculated 1:1,000 (v/v) into MMT-CA medium containing 0.05% (w/v) *Glossomastix* fucoidan. $OD_{600}$ values of the cultures were measured on day 6 and ability to degrade fucoidan was assessed by carbohydrate polyacrylamide gel electrophoresis (C-PAGE) as described below.

## Isolation of fucoidan-degrading bacterium V_227

Sediment-associated bacteria were sampled on 27 February 2022 at low tide from the North Beach of Helgoland, Germany, and used to inoculate 13 ml polypropylene tubes (Sarstedt, 62.515.006) containing 3 ml MMT-YE medium supplemented with 0.2% (w/v) fucoidan (glycans eluted in AEX using 0.5 M NaCl). Enrichment cultures were incubated at 17 °C at 115 r.p.m. for 2 weeks, and then diluted 1:100 (v/v) into fresh MMT-YE medium with fucoidan (AEX 5 M NaCl fraction) and cultivated for an additional 7 days. Cultures showing an increase in optical density were diluted 1:10[5] with fresh MMT medium and plated on solid MMT-YE with 0.05% (w/v) fucoidan and 1% (w/v) agarose. After incubating at r.t. for 5–6 days, the putative fucoidan-degrading isolates appearing as colonies on the plates were re-inoculated into fresh MMT-YE medium to confirm growth and degradation of fucoidan.

## Detection of fucoidan degradation

Bacterial degradation of fucoidan in the culture medium was analysed qualitatively by the BSA–acetate method[76,77] or C-PAGE[78,79], and quantitatively by a toluidine blue (TB) assay[80].

**BSA–acetate method.** For the BSA–acetate assay, 20 µl culture supernatant was mixed in a 96-well plate with 180 µl acid albumin solution (per litre: 3.26 g of sodium acetate, 4.56 ml of glacial acetic acid and 1 g of bovine serum albumin, pH adjusted to 3.72 to 3.78). Degradation of fucoidan was assessed by observing the formation of cloudy white precipitates against a black background; the degree of turbidity correlates positively with the concentration of acidic polysaccharides and thereby, a clear transparent solution indicated fucoidan degradation.

**C-PAGE.** Bacterial cultures with fucoidan were centrifuged at 12,000 r.p.m. (13,850 × *g*) (Thermo Scientific Fresco 21 microcentrifuge) for 10 min, and 24 µl of supernatant was mixed with 6 µl 5× phenol red loading dye before loading onto an acrylamide gel (25% resolving/5% stacking). Electrophoresis was performed for 30 min at 100 V, followed by 1 h at 200 V in native running buffer (1 l: 3 g Tris, 15 g glycine). The gel was stained with 0.005% (w/v) Stains-all in 30% ethanol overnight, and de-stained with UPW until the background of the gel was clear.

**TB method.** For the TB assay, 10 µl culture supernatant was mixed with 990 µl TB solution (0.03 mM toluidine blue in 20 mM citrate buffer, pH 3.0, 0.22-µm filtered). Then, 100 µl solution was transferred to a 96-well plate and absorbance measured at 632 nm in a SpectraMax iD3 plate reader (Molecular Devices). Absorbance values were converted to concentration via standard curves constructed on *Glossomastix* fucoidan (0–1 mg ml⁻¹), where sulfated fucoidan concentration is inversely proportional to $OD_{632\,nm}$ (Supplementary Table 11).

## Growth physiology of V_227

**Carbohydrate assimilation experiments.** Growth assays with mono- and polysaccharides were performed in multiwell culture plates (0.5–1 ml medium) or 13 ml culture tubes (4 ml medium) depending on the availability of carbohydrate substrate and the volume needed for glycan detection. For growth assays with *Glossomastix* fucoidan, a 4-day-old pre-culture of V_227 grown in MMT-YE was inoculated 1:100 (v/v) into 4 ml fresh MMT-YE medium (20 °C, 110 r.p.m.). Negative control means that no carbohydrates were added. This growth experiment was performed in 13 ml polypropylene tubes (Sarstedt, 62.515.006). Sampling was performed at intervals of 6–14 h and the $OD_{600}$ was measured in 10 mm cuvettes using a BioSpectrometer (Eppendorf AG). Supernatants were centrifuged at 12,000 r.p.m. (~13,850 × *g*) (Thermo Scientific Fresco 21 microcentrifuge) for 10 min and collected for fucoidan detection (all time points) and monosaccharides detection (some time points) after acid hydrolysis via HPAEC-PAD. This growth experiment was performed in test tubes. In subsequent growth experiments, pre-cultures were inoculated into fresh medium at a ratio of 1:1,000.

Monosaccharides (Supplementary Table 10) were used in the growth assay at a final concentration of 0.05% (w/v). For the fucose, rhamnose, galacturonate and glucuronate, V_227 was incubated for 3 weeks (MMT-CA, 24-well plate or test tubes, 20 °C, 110 r.p.m.). For glucose and galactose, V_227 was incubated for 1 week (MMT-KDP, 24-well plate, r.t., 400 r.p.m.).

Growth experiments with *Fucus vesiculosus* fucoidan and *Glossomastix* fucoidan as carbohydrates were carried out in MMT-CA medium at a final concentration of 0.05% (w/v), and $OD_{600}$ was measured at 6–24 h intervals (24-well plate, r.t., 400 r.p.m.). Negative control means that no carbohydrates were added. We simultaneously tested the growth of V_227 with the addition of different polysaccharides in MMT-KDP medium (100 µM KH₂PO₄) (Supplementary Table 10), and the cultivation was performed in 48-well plates (Sarstedt, 83.3923.500) (r.t., 400 r.p.m.). $OD_{600}$ was measured via BioSpectrometer (Eppendorf) after 1 week incubation.

**Phosphate-limitation experiments.** V_227 was cultivated in MMT-KDP medium (1–50 µM KH₂PO₄ final concentration; for pectin, 1–30 µM) with 0.05% (w/v) polysaccharide as the sole carbon source: *Glossomastix* fucoidan (5,592.98 µM glycan carbon, 24-well plate, r.t., 400 r.p.m.), *Eisenia bicyclis* laminarin (14,911.39 µM glycan carbon, 24-well plate, r.t., 400 r.p.m.), corncob xylan (15,381.25 µM glycan carbon, 24-well plate, r.t., 450 r.p.m.) and sugar beet pectin (12,294.61 µM glycan carbon, 24-well plate, r.t., 450 r.p.m.) (Supplementary Table 10). Growth was monitored during 144 h of incubation, after which the cultures were centrifuged at 12,000 r.p.m. (~13,850 × *g*) (Thermo Scientific Fresco 21 microcentrifuge) for 10 min. Supernatants were subjected to monosaccharide composition analysis via HPAEC-PAD and total carbohydrate quantification as described above. The converted glycan carbon is the sum of the initial molar concentrations of all monosaccharides minus the sum of the molar concentrations of all monosaccharides remaining in the supernatant at 144 h, and then the value obtained was multiplied by 6 for these hexoses having 6 carbon atoms.

## Calibration of OD against cell counts

V_227 was grown in MMT-CA medium with 0.05% (w/v) fucoidan. When V_227 was in the logarithmic growth phase ($OD_{600}$ = 0.262), cultures were diluted into groups with different cell densities. The $OD_{600}$ of the diluted cultures was measured in 24-well plates using a SpectraMax iD3 plate reader. Each serial dilution was then further diluted 10,000 times, and 100 µl was spread on MMT-CA plates (0.05% fucoidan and 1% agar) and incubated for 9 days. Linear correlation analysis was performed between the number of colonies counted and the $OD_{600}$ value (Supplementary Table 12).

## Super-resolution microscopy of selfish fucoidan uptake

Fucoidan was fluorescently labelled with fluoresceinamine isomer II (Sigma, 51649-83-3) as previously described[81]. Pre-cultures of V_227

were diluted 1:1,000 in 1 ml MMT-KDP medium (50 μM KH₂PO4) containing 0.4% (w/v) fluorescently labelled fucoidan, and triplicate cultures were grown in a 24-well plate (r.t., 400 r.p.m.). At several time points over 7 days, 100-μl cultures were fixed with 2% (v/v) formaldehyde and diluted 1:10 in 1× PBS buffer. The fixed cells were collected on polycarbonate filters (0.2 μm, Ø47 mm) before staining with 4′,6-diamidino-2-phenylindole (DAPI), and then mounted onto glass slides using a Citifluor/VectaShield (4:1) solution. Likewise, 100 μl V_227 pre-culture with 100 μl MMT-KDP medium was fixed, DAPI-stained and applied to glass slides as controls. Stained V_227 cells (day 3) were visualized by epifluorescence microscopy on an AxioImager.Z2 microscopic stand (Carl Zeiss) equipped with automated imaging and light-emitting diodes[82,83]. Images were acquired at ×63 magnification. For super-resolution microscopy, the cells were visualized on a Zeiss ELYRA PS.1 (Carl Zeiss) using 488 and 405-nm lasers, and BP 502–538 and BP 420–480 + LP 750 optical filters. Z-stack images were taken with a Plan-Apochromat ×63/1.4 oil objective and processed with the ZEN2011 software (Carl Zeiss).

### Genome sequencing and assembly
Genomic DNA was extracted with the Gentra Puregene Yeast/Bact kit (Qiagen) using a 5-ml culture of V_227 that was grown in MMT-YE medium (20 °C, 110 r.p.m.) for 4 days, and the purified genomic DNA was then sequenced on a PacBio Sequel II platform at the Max Planck Genome Centre Cologne. Assembly of the V_227 genome was carried out using HiCanu (v.2.2)[84] based on PacBio hifi reads, and assembly quality was evaluated using checkM (v.1.2.2)[85].

### Relative abundance in TARA and Helgoland spring bloom water
We performed read mapping against the TARA read dataset using Bowtie2 (v.2.3.5.1)[86]. Samtools (v.1.7)[87] was used to convert the SAM files to BAM, which were then sorted. The trimmed_mean values across all reads were obtained from the sorted BAM files via CoverM v.0.6.1 (https://github.com/wwood/CoverM). The trimmed mean was normalized using the genome equivalence of the *RpoB* gene as calculated from the TARA dataset. This was then repeated using the Helgoland spring bloom read dataset (mid-March – mid-May of 2010, 2011, 2012, 2016, 2018 and 2020)[88,89], except that the trimmed mean was then normalized using the genome equivalence of the *RpoB* gene as calculated from the Helgoland dataset. The genome equivalence of the *RpoB* gene was obtained by mapping all reads in the TARA dataset against a collection of reference *RpoB* sequences[29,90] using the same tools detailed above. The genome equivalence for each sample was calculated as follows: (total number of hits to the *RpoB* references × average read length) ÷ the average length of the *RpoB* references. This analysis was then repeated using the Helgoland dataset.

### Phylogenetic tree of fucoidan-degrading Verrucomicrobiaceae sp. isolate
Genomes and MAGs belonging to the Akkermansiaceae family were obtained from GTDB (v.214.1)[91]. Genomes used were selected on the basis of their quality (completion 5×, contamination ≥50) as determined using checkM (v.1.2.2)[85] (lineage_wf) and then dereplicated using a 99% ANI threshold for the secondary clustering step in dRep (v.3.4.3)[92]. A total of 246 selected genomes (Supplementary Table 7) were further used for phylogenetic reconstruction based on 120 conserved genes determined using GTDB-tk (v.2.3.2)[93] and the v.214.1 GTDB release. A maximum likelihood tree was determined using IQ-TREE v.2.2.2.6 (-m MFP)[63] and visualized using the interactive Tree of Life (iTol)[94].

### Genome annotation
Preliminary annotation of the V_227 genome was performed with Prokka (v.1.14.5)[95]. The protein domain families were annotated using Pfam-A HMMs[96] via HMMER (v.3.3.2, http://hmmer.org) with a 'cut_tc' thresholding, and the hit with the highest score value for

each sequence was extracted. CAZymes were predicted using a combination of HMMER ($E$-value $< 1 \times 10^{-15}$, query coverage >35%) against the dbCAN v.11 (https://bcb.unl.edu/dbCAN2/download/Databases/V11/) HMM database and Diamond blastp[97] (v.2.0.14.152, $E$-value $< 1 \times 10^{-20}$, identity >40% and query coverage >50%) against the 2022 CAZy database[98]. Only results with consistent dbCAN and CAZy annotations were considered reliable. Sulfatases were annotated on the basis of Diamond blastp (v.2.0.14.152, $E$-value $< 1 \times 10^{-20}$, identity >40% and query coverage >50%) searches against the SulfAtlas database (v.1.3)[99]. Transporters were annotated using the Transporter Automatic Annotation Pipeline (TransAAP)[100]. Alpha-L-fucosidases were annotated using reference HMM models (PF01120) via HMMER. The MetaCyc database[101–103] and BlastKO-ALA[104] were used for metabolic pathway reconstruction and confirmed by InterPro[105] or CDD[106,107]. Prodigal (v.2.6.3)[108] was used to obtain amino acid sequences of proteins encoded in 245 bacterial genomes selected from the Akkermansiaceae family. Proteins in the phosphate transport high-affinity system (PstA, PstB, PstC and PstS) were identified in the 245 genomes via HMMER using the individual HMM modules: PstA (TIGR00974.1, $E$-value $< 1 \times 10^{-15}$, query coverage >50%), PstB (TIGR00972.1, $E$-value $< 1 \times 10^{-50}$, query coverage >50%), PstC (TIGR02138.1, $E$-value $< 1 \times 10^{-15}$, query coverage >50%) and PstS (TIGR00975.1, $E$-value $< 1 \times 10^{-05}$, query coverage >35%). A bacterium was considered to encode Pst when at least three homologues were identified in its genome. The AAI matrix was determined using the aai.rb script from the enveomics collection[109]. CAZymes, sulfatases and transporters encoded by the other SW10 genomes were annotated using the same approach detailed above, but a length filter was added to select matches with at least 50% of alignment coverage to the subject.

### Data visualization and statistics
Most data visualizations and statistical analyses were generated using Python with Matplotlib[110], Pandas[111], NumPy[112], SciPy[113] and Statsmodels[114] packages. Growth rate was obtained with R using the Growthcurver[115] package. The synteny of the regions encoding the GH141 genes was plotted in R using gggenes[116]. Figure 3f and Extended Data Figs. 3a and 4d were created with BioRender.com. Extended Data Fig. 4a was created with Google Maps.

### Sustainability and inclusion statement
The here-conducted experiments, methods, instruments and organisms are broadly accessible, hence this work is inclusive for many scientists around the globe. The most advanced instrument was a nuclear magnetic resonance machine. The next advanced instrument was an HPAEC-PAD machine for detection of glycans with pulsed amperometric detection. HPLC machines with other modes of detection can also work for the analysis of glycans. Other commonly accessible instruments include a spectrophotometer. Only one bacterial genome was sequenced for this study, limiting the amount of data that require long term storage and sustainability.

### Reporting summary
Further information on research design is available in the Nature Portfolio Reporting Summary linked to this article.

## Data availability
All relevant data supporting the results of this study are available in the paper and its Supplementary Information. The 18S rRNA gene sequence of *Glossomastix* sp. PLY432 has been deposited in NCBI under accession number PP265255. The genomic data of V_227 have been deposited in NCBI under accession number PRJNA1070871. All experimental data used to generate plots are available in the Source Data file. Please note that tables have been submitted as supplementary files. Source data are provided with this paper.

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

## Acknowledgements

We thank J. Parnami and K. Imhoff from Max Planck Institute for Marine Microbiology (MPIMM) for sulphate quantification; A. Bolte, G. Klockgether and K. Föll from MPIMM for conducting the HPAEC-PAD and elemental measurements; T. Horstmann from MPIMM for help in pre-treatment of microalgal cultures and *Glossomastix* genomic DNA extraction; M. Oftebro from the Norwegian University of Science and Technology for desulfating the glycan before NMR characterization, and B. Huettel from Max Planck Genome Centre Cologne for genome sequencing; I. Probert from Roscoff Culture Collection for providing *Glossomastix* sp. PLY432, and T. Song and G. Huang from Sichuan University for their advice on experimental design. We acknowledge the institutional support from the University of Bremen, MARUM and MPIMM. Y.X. was supported by a scholarship granted by the China Scholarship Council (No. 201906240194). L.J.K. and F.L.A. were funded by the Research Council of Norway through projects of the Norwegian seaweed Biorefinery Platform (Grant 294946) and Norwegian NMR Platform (Grant 226244). G.R. received funding from the German Research Foundation (Grant 496342779). The work was funded by the European Research Council, ERC grant, C-Quest (Grant 101044738) to J.-H.H. and by the Simons Foundation through the Principles of Microbial Ecosystems (PriME) collaboration (Grant 970824) to J.-H.H. M.S.-J. received funding from the BMBF SNAP BlueBio project (Grant 161B0943). J.-H.H. was funded by the German Research Foundation through the Heisenberg program Grant (HE 7217/5-1). We also acknowledge funding from the Max Planck Society.

## Author contributions

Y.X., D.Q. and J.-H.H. conceived and planned the study. Y.X., Y. Cao and J.-H.H. initiated the research. Y.X. and M.S.-J. isolated bacteria. Y.X. and Y. Chen performed bacterial growth experiments. B.G. performed semi-continuous experiments, phosphate and deoxy-sugar quantifications. Y.X. and S.-V.M. performed ELISA. L.J.K. and F.L.A. obtained and interpreted NMR spectroscopy data. I.W., L.H.O. and Y.X. did bioinformatic analyses. M.F. and G.R. did fluorescence labelling and microscopy. Y.X., B.G., H.Y. and J.-H.H. analysed the data and designed illustrations. Y.X., B.G., H.Y., M.S.-J. and J.-H.H. wrote the manuscript with input from all authors.

## Funding

## Competing interests

The authors declare no competing interests.

## Additional information

**Extended data** is available for this paper at https://doi.org/10.1038/s41564-025-02240-z.

**Correspondence and requests for materials** should be addressed to Yi Cao or Jan-Hendrik Hehemann.

a

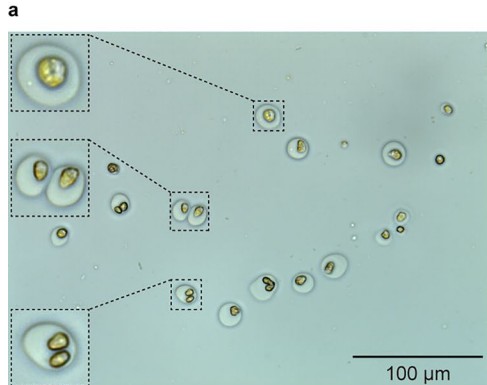

b

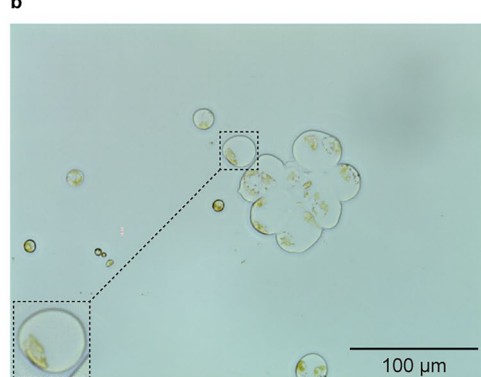

**Extended Data Fig. 1 | *Glossomastix* cells were coated with mucus.**
**a**, Microscopic imaging for less viscous *Glossomastix* culture. Some algal cells have finished their bipartition period and are detaching from each other.

**b**, Microscopic imaging for highly viscous *Glossomastix* culture. Cell fragments were enveloped in mucus and formed aggregates. Similar results were observed in different fields of view across three independent replicates.

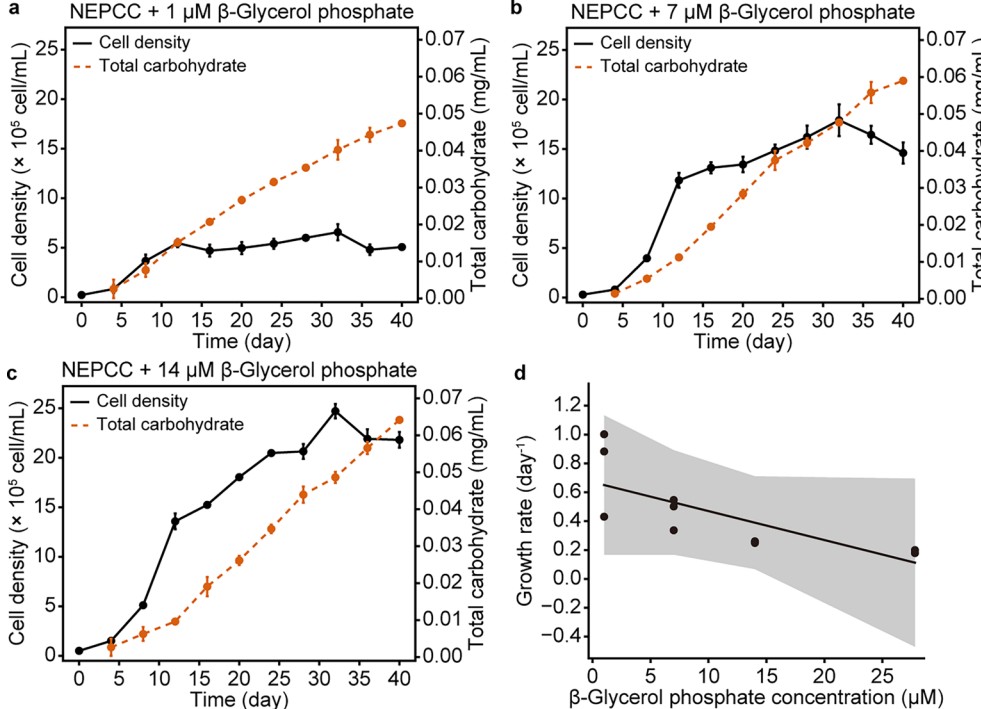

**Extended Data Fig. 2 | Growth rate of *Glossomastix* was not significantly affected by the decrease of organic phosphate. a–c**, Dynamic monitoring of total carbohydrate content and cell density in *Glossomastix* cultures with different concentrations (1, 7, and 14 µM) of β-Glycerol phosphate as the phosphorus source. These experiments were performed in independent triplicate (n = 3), and error bars are the standard deviation of the mean. **d**, Correlation analysis between the growth rate of *Glossomastix* and phosphate concentration (*r* = -0.89, *P* = 0.1131; two-sided Pearson-correlation test). The shade indicates the 95% confidence interval around the linear regression line.

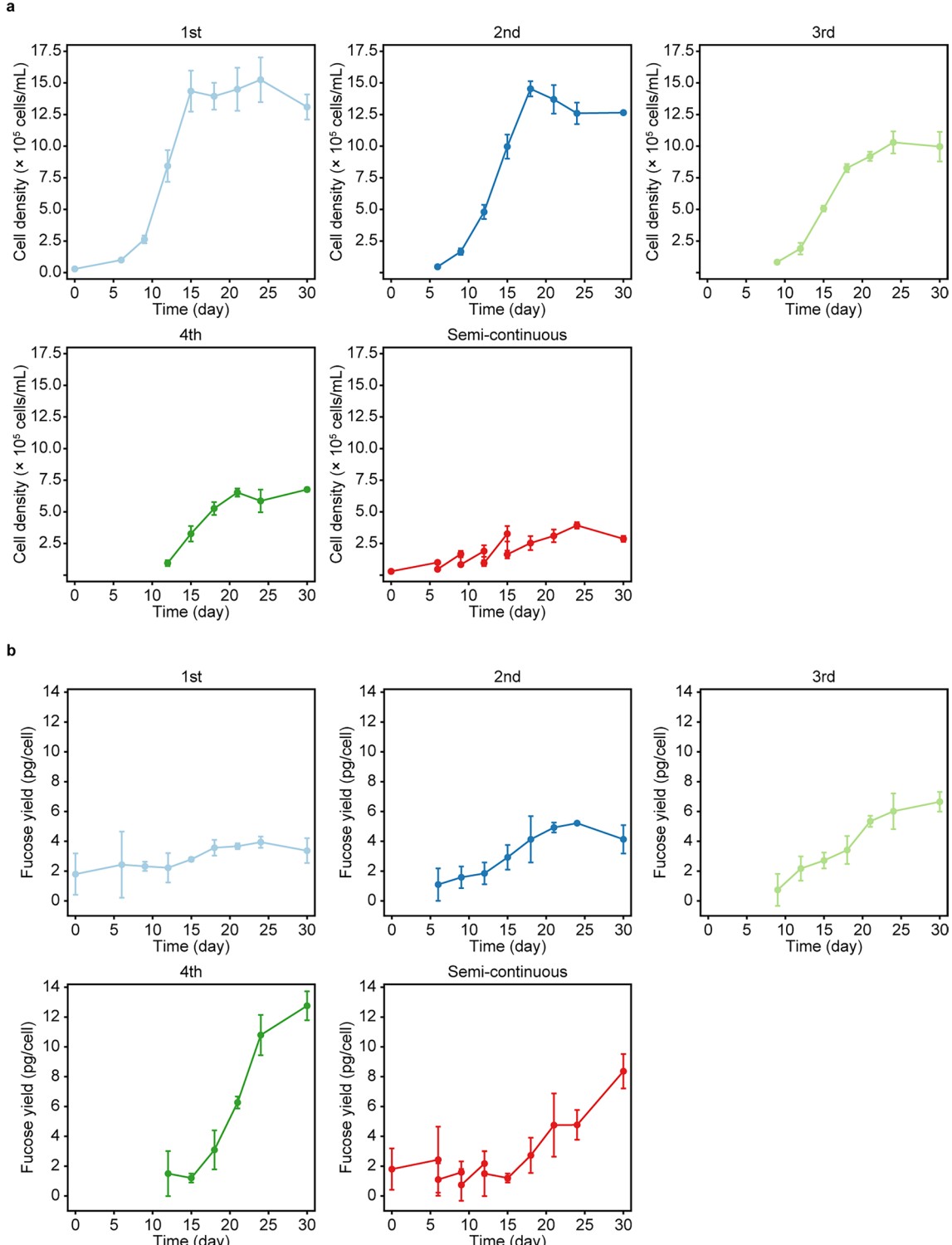

**Extended Data Fig. 3 | Cell growth and fucoidan secretion of *Glossomastix* in the semi-continuous culture. a**, Cell density of *Glossomastix* in a semi-continuous growth and batch mode. **b**, Fucose yield of in the semi-continuous growth and batch mode. In the 1st batch culture, the starting inorganic phosphate concentration was 14 μM. In the 2nd, 3rd, 4th batch cultures and the semi-continuous growth, the actual phosphate concentrations were lower than the theoretical ones (7 μM, 3.5 μM, 1.75 μM and 0.875 μM respectively). The experiment was performed in independent triplicate (n = 3), error bars are the standard deviation of the mean.

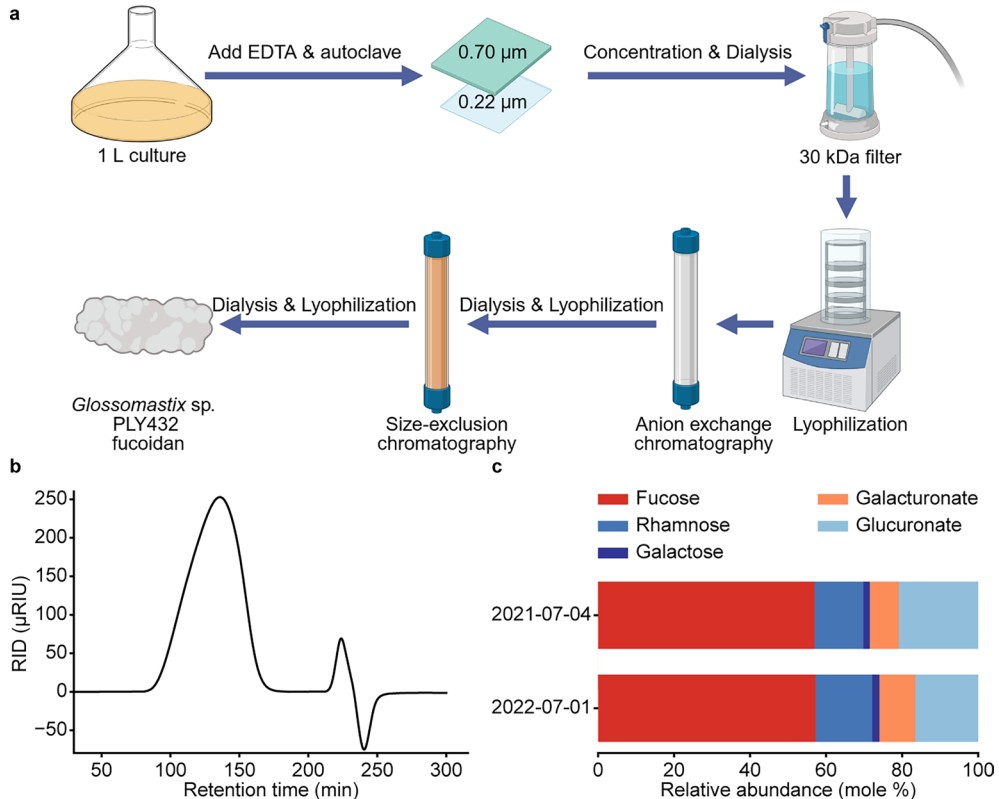

**Extended Data Fig. 4 | Purification of *Glossomastix* fucoidan. a**, Schematic representation of the *Glossomastix* fucoidan purification workflow. The panel was created with BioRender.com. **b**, Size exclusion chromatography for AEX-purified fucoidan. **c**, Monosaccharide composition analysis of different, one year apart batches of AEX-purified fucoidan. The *Glossomastix* cells were continuously cultivated.

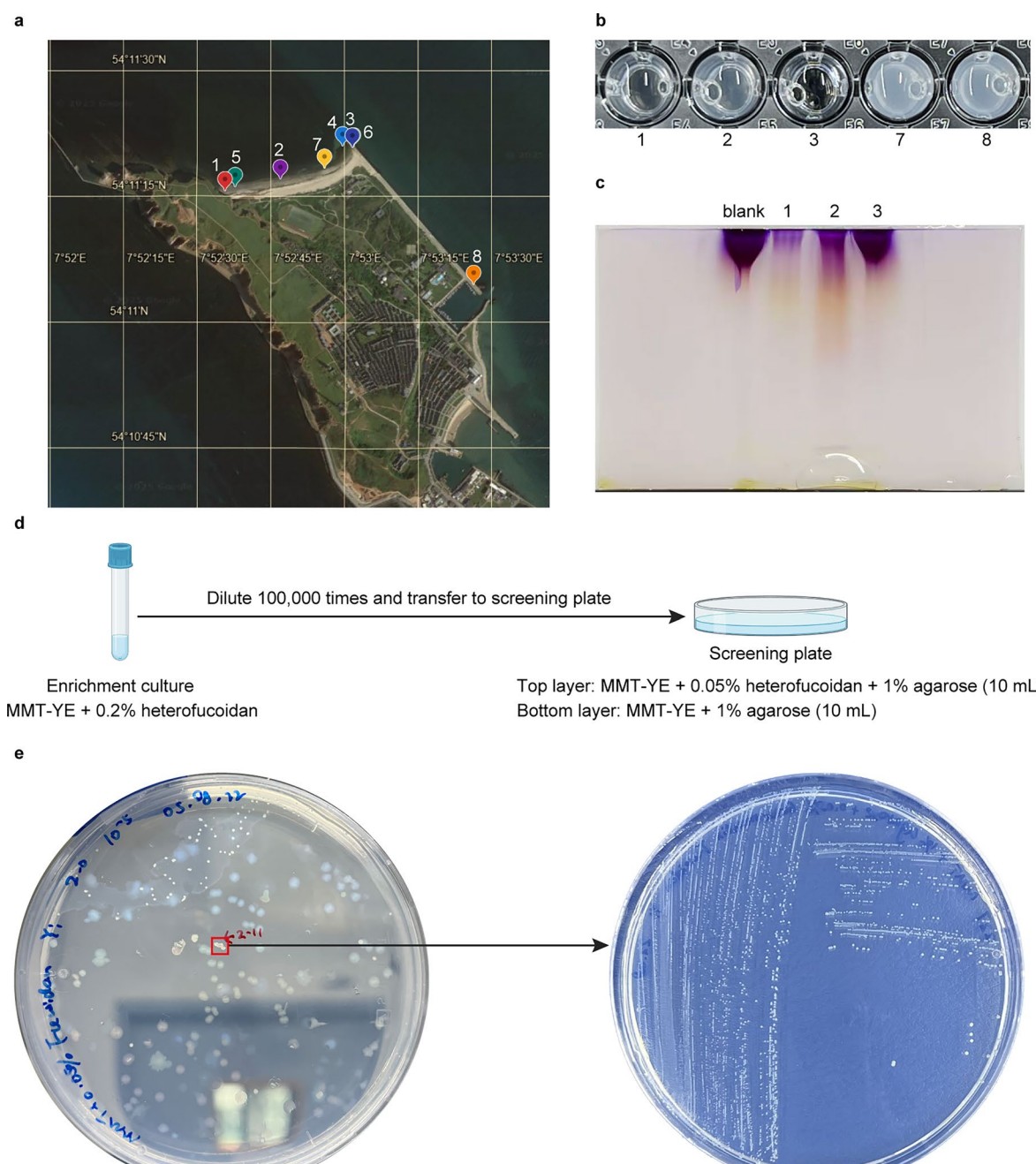

**Extended Data Fig. 5 | Process of isolating fucoidan degraders V_227.**
**a**, Sampling locations in Helgoland. Map source: Google Earth imagery
(© Google) **b**, Detection of fucoidan degradation via BSA acetate method.
As the clarity of the solution increases, there is a proportional escalation in the
degradation of fucoidan. **c**, Carbohydrate polyacrylamide gel electrophoresis
(C-PAGE) for the consumption of fucoidan. A suitable size marker does not yet

exist for this fucoidan. The samples were obtained after 14 days of cultivation.
Note that the enrichment 3 in the BSA assay shows degradation of fucoidan, but
after transferring into fresh medium the degradation ability was lost as indicated
by the high molecular weight band of fucoidan on the gel. **d**, **e**, Isolation steps and
the physiological morphology of bacteria on MMT-YE + 0.05% fucoidan agarose
plates. Panel **d** was created with BioRender.com.

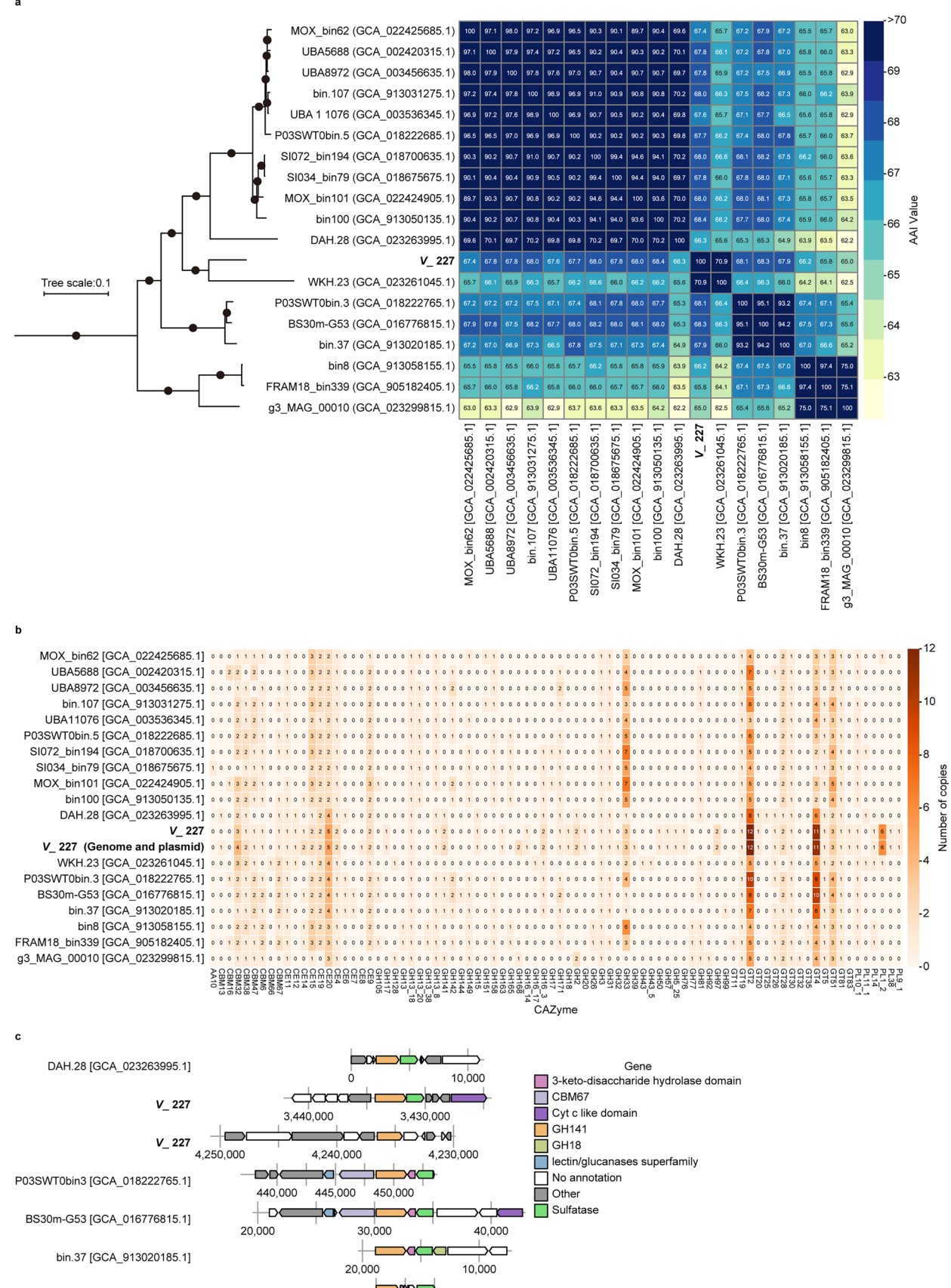

**Extended Data Fig. 6 | Genomic and functional characteristics of the SW10 genus. a**, Average amino acid (AAI) values between all members of the SW10 genus. AAI values range from 62% (pale yellow) to over 70% (blue). The genomes are ordered according to the phylogenetic clustering determined in this work. The diagonal line represents the comparison of genomes against themselves. **b**, Summary of CAZy enzymes detected in all representative species of the SW10 genus. **c**, Genetic contexts for GH141 genes in members of the SW10 genus.

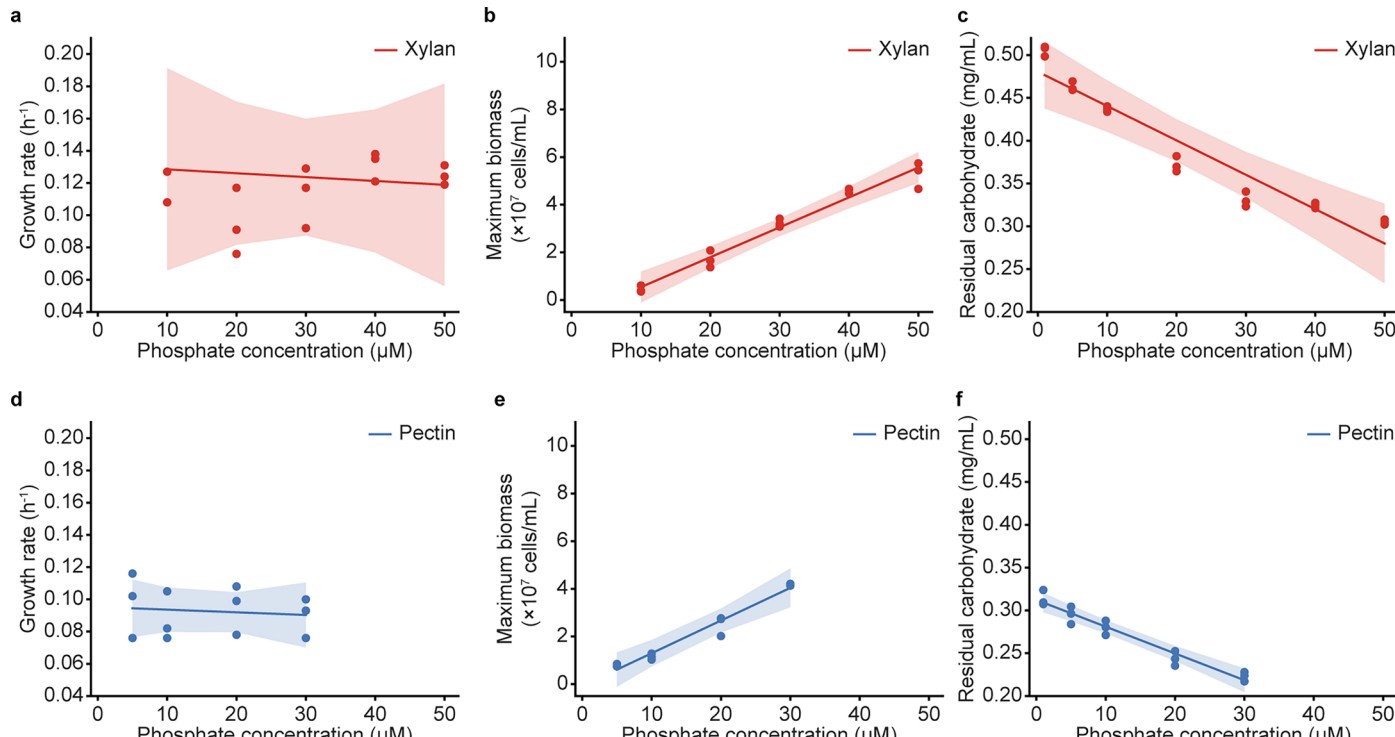

**Extended Data Fig. 7 | Phosphate limits the utilization of complex carbohydrates by V_227. a**, Correlation analysis between growth rate and phosphate concentration on xylan ($r = -0.17$, $P = 0.7852$). **b**, Correlation analysis between maximum biomass and phosphate concentration on xylan ($r = 0.99$, $P = 0.0006301$) **c**, Correlation analysis between residual carbohydrate in the culture supernatant and phosphate concentration on xylan (slope = -0.004, $r = -0.96$, $P = 0.0007908$). **d**, Correlation analysis between growth rate and phosphate concentration on pectin ($r = -0.39$, $P = 0.6128$). **e**, Correlation analysis between maximum biomass and phosphate concentration on pectin ($r = 0.99$, $P = 0.006697$) **f**, Correlation analysis between residual carbohydrate in the culture supernatant and phosphate concentration on pectin (slope = -0.003, $r = -0.99$, $P = 0.0007424$). The absent data points of 0 μM phosphate concentration in panels **a**, **b**, **d** and **e** was because the logistic growth model did not fit at low phosphate concentrations. The absent data points of 40 and 50 μM phosphate concentrations in panels **d**, **e** and **f** was because a large amount of precipitation appeared at the bottom of the 24-well plate and a growth curve could not be obtained. The fitting was performed using the mean of independent triplicate (n = 3). $P$-value was derived from two-sided Pearson correlation test. Shade indicates the 95% confidence interval around the linear regression line.

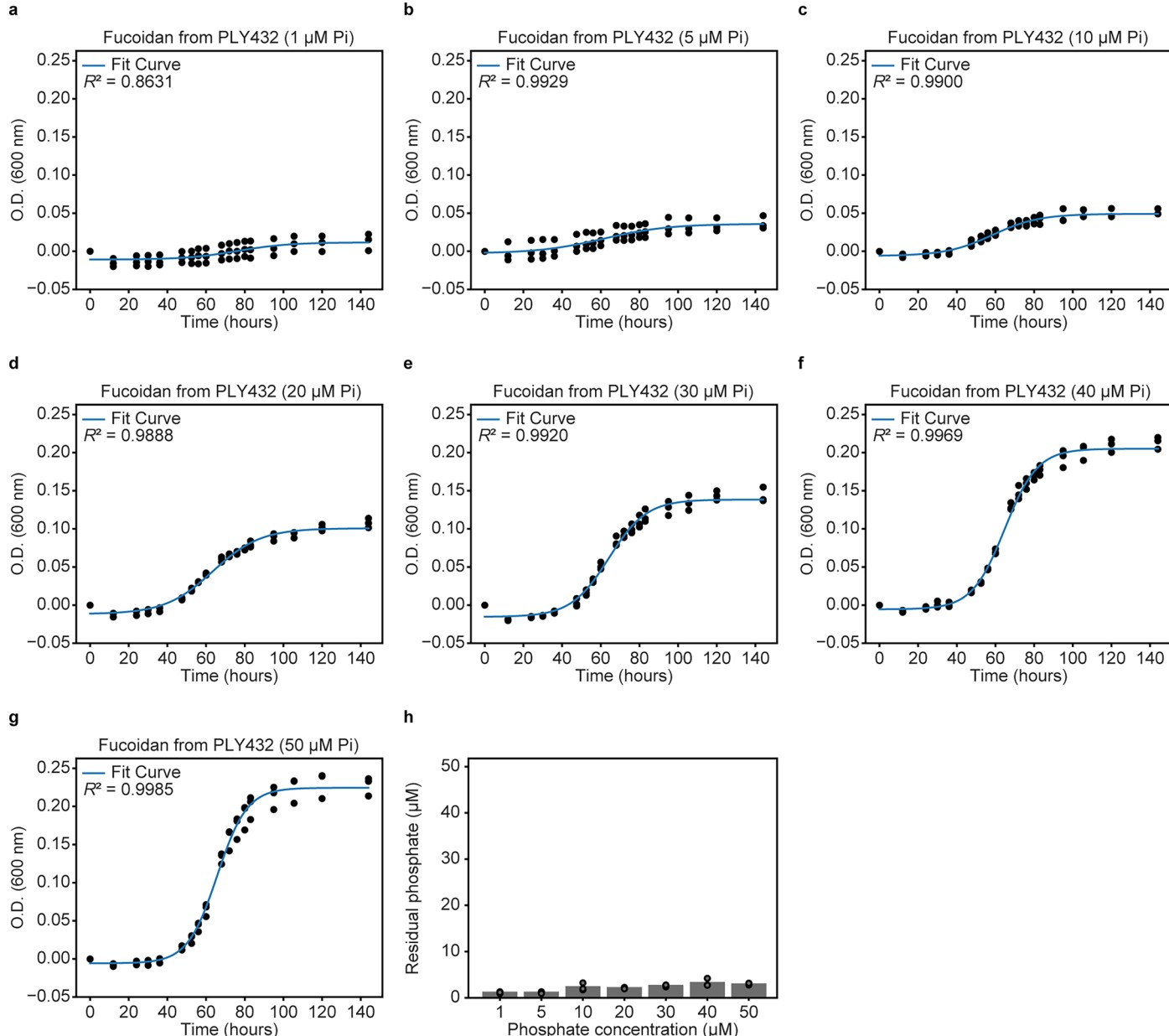

**Extended Data Fig. 8 | Growth of V_227 in MMT-KDP medium containing** *Glossomastix* **fucoidan as the carbon source with varying concentrations of phosphate. a–g**, Growth curve. The experiment was performed in independent triplicate (n = 3). The fitting was performed using the mean of triplicate.

**h**, Residual phosphate in each culture by the end of bacterial growth. Experiments were performed as independent triplicates (n = 3) and bars represent the mean vale of triplicates.

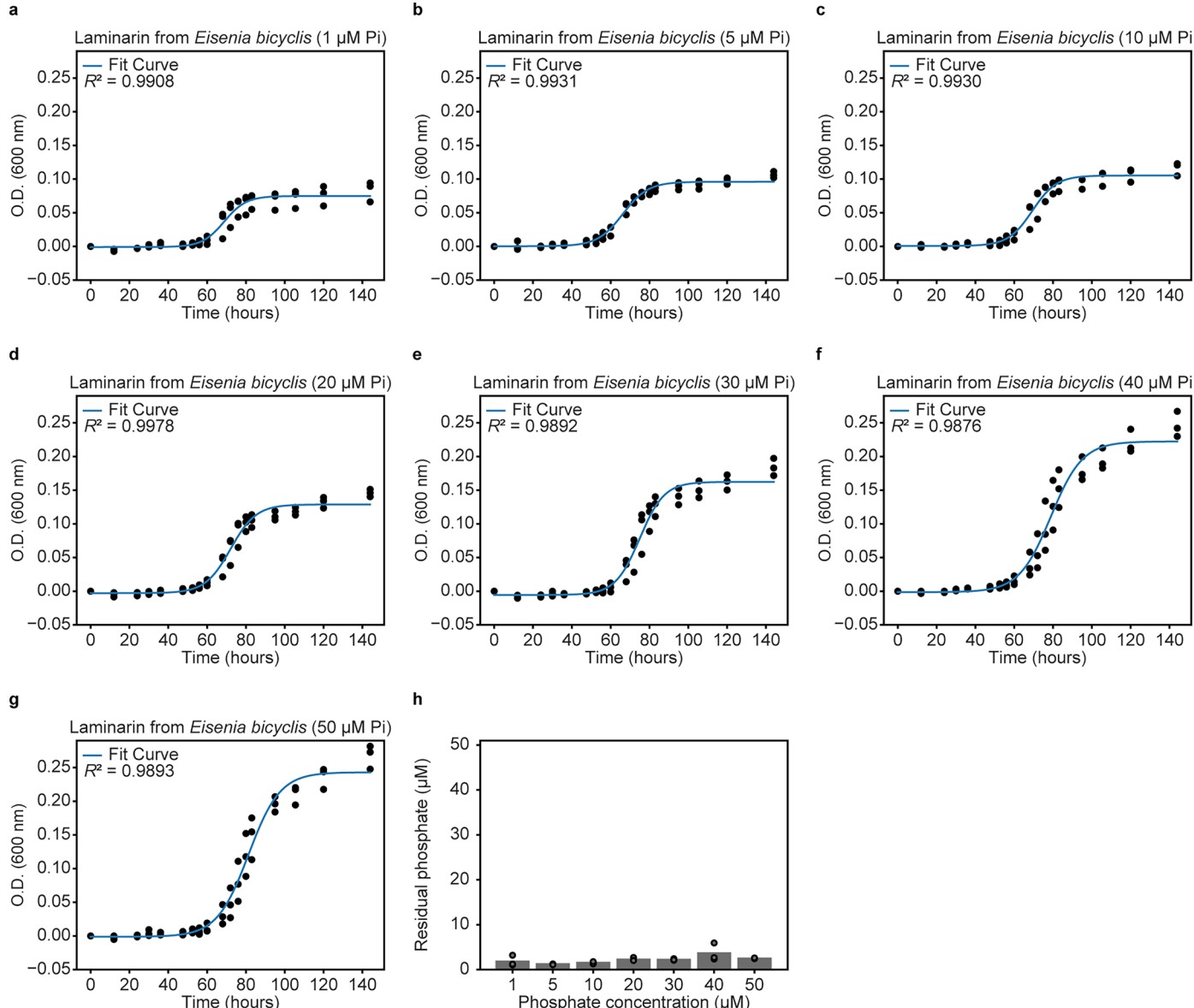

**Extended Data Fig. 9 | Growth of V_227 in MMT-KDP medium containing Eisenia bicyclis laminarin as the carbon source with varying concentrations of phosphate. a–g,** Growth curve. The experiment was performed in independent triplicate (n = 3). The fitting was performed using the mean of triplicate. **h,** Residual phosphate in each culture by the end of bacterial growth. Experiments were performed as independent triplicates (n = 3) and bars represent the mean vale of triplicates.

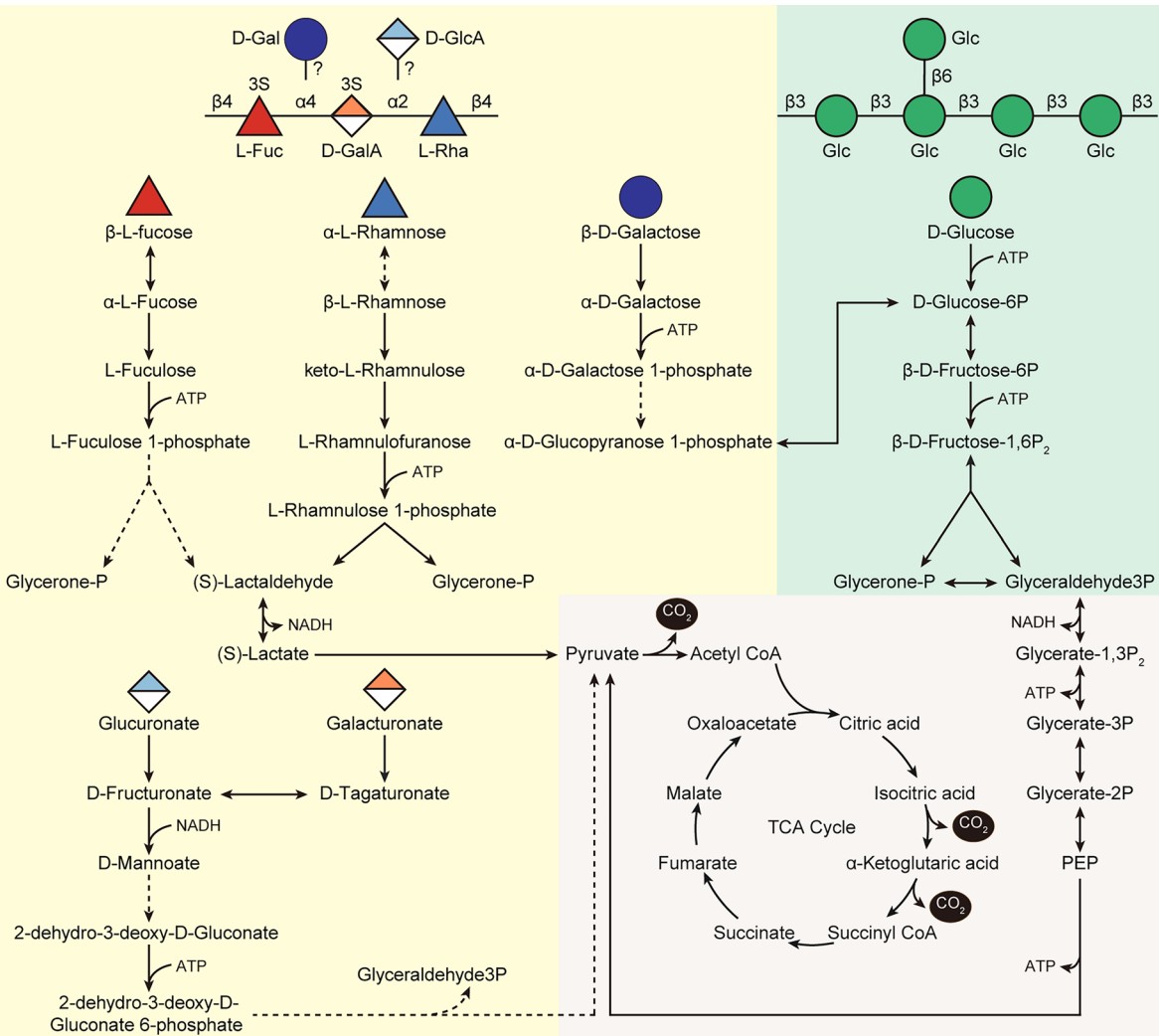

**Extended Data Fig. 10 | Reconstructed metabolic pathways for six monosaccharides in V_227.** Solid lines represent homologs found in the genome or this reaction is spontaneous, dashed lines represent homologs not found. The structure of *Eisenia bicyclis* Laminarin and the partial structure of *Glossomastix* fucoidan are on the top.

# Reporting Summary

## Statistics

For all statistical analyses, confirm that the following items are present in the figure legend, table legend, main text, or Methods section.

| n/a | Confirmed | |
|---|---|---|
| ☐ | ☒ | The exact sample size (*n*) for each experimental group/condition, given as a discrete number and unit of measurement |
| ☐ | ☒ | A statement on whether measurements were taken from distinct samples or whether the same sample was measured repeatedly |
| ☐ | ☒ | The statistical test(s) used AND whether they are one- or two-sided *Only common tests should be described solely by name; describe more complex techniques in the Methods section.* |
| ☒ | ☐ | A description of all covariates tested |
| ☐ | ☒ | A description of any assumptions or corrections, such as tests of normality and adjustment for multiple comparisons |
| ☐ | ☒ | A full description of the statistical parameters including central tendency (e.g. means) or other basic estimates (e.g. regression coefficient) AND variation (e.g. standard deviation) or associated estimates of uncertainty (e.g. confidence intervals) |
| ☐ | ☒ | For null hypothesis testing, the test statistic (e.g. *F*, *t*, *r*) with confidence intervals, effect sizes, degrees of freedom and *P* value noted *Give P values as exact values whenever suitable.* |
| ☒ | ☐ | For Bayesian analysis, information on the choice of priors and Markov chain Monte Carlo settings |
| ☒ | ☐ | For hierarchical and complex designs, identification of the appropriate level for tests and full reporting of outcomes |
| ☐ | ☒ | Estimates of effect sizes (e.g. Cohen's *d*, Pearson's *r*), indicating how they were calculated |

*Our web collection on statistics for biologists contains articles on many of the points above.*

## Software and code

Policy information about availability of computer code

| | |
|---|---|
| Data collection | MagIC Net v3.2, TopSpin v3.5 and v4.0.1, Chromeleon v7.2, SMRT Portal software v2.3.0, ZEN2011 (Carl Zeiss, Germany), SoftMax® Pro Software 7, MARS (SPECTROstar® Nano absorbance plate reader) and PurityChrom v5.09.069. |
| Data analysis | MUSCLE v3.8.31, IQ-TREE v2.2.2.6, HiCanu v2.2, checkM v1.2.2, Bowtie2 v2.3.5.1, Samtools v1.7, CoverM v0.6.1, GTDB v214.1, GTDB-tk v2.3.2, dRep v3.4.3, Prokka v1.14.5, HMMER v3.3.2, dbCAN v11, Diamond blastp v2.0.14.152, SulfAtlas database v1.3, Transporter Automatic Annotation Pipeline, MetaCyc database, BlastKOALA, Prodigal v.2.6.3, Python 3.9, R v4.3.1, MPI Bioinformatics Toolkit, TVBOT, Tree of Life (iTol) and Growthcurver. |

For manuscripts utilizing custom algorithms or software that are central to the research but not yet described in published literature, software must be made available to editors and reviewers. We strongly encourage code deposition in a community repository (e.g. GitHub). See the Nature Portfolio guidelines for submitting code & software for further information.

## Data

Policy information about availability of data

All manuscripts must include a data availability statement. This statement should provide the following information, where applicable:
- Accession codes, unique identifiers, or web links for publicly available datasets
- A description of any restrictions on data availability
- For clinical datasets or third party data, please ensure that the statement adheres to our policy

All relevant data supporting the results of this study are available in the paper and its supplementary information. The 18S rRNA gene sequence of Glossomastix sp. PLY432 has been deposited in NCBI under accession number PP265255. The genomic data of V_227 has been deposited in NCBI under accession number PRJNA1070871. The raw data of HPAEC-PAD plus the resulting data tables used for calculation are currently being deposited on the Pangaea data repository https://www.pangaea.de/. A digital object identifier for these data will follow. Please note the tables have been also submitted as supplementary files.

## Research involving human participants, their data, or biological material

Policy information about studies with human participants or human data. See also policy information about sex, gender (identity/presentation), and sexual orientation and race, ethnicity and racism.

| | |
|---|---|
| Reporting on sex and gender | NA |
| Reporting on race, ethnicity, or other socially relevant groupings | NA |
| Population characteristics | NA |
| Recruitment | NA |
| Ethics oversight | NA |

Note that full information on the approval of the study protocol must also be provided in the manuscript.

# Field-specific reporting

Please select the one below that is the best fit for your research. If you are not sure, read the appropriate sections before making your selection.

☒ Life sciences   ☐ Behavioural & social sciences   ☐ Ecological, evolutionary & environmental sciences

For a reference copy of the document with all sections, see nature.com/documents/nr-reporting-summary-flat.pdf

# Life sciences study design

All studies must disclose on these points even when the disclosure is negative.

| | |
|---|---|
| Sample size | No sample size calculations were performed. |
| Data exclusions | No data was excluded from the analyses. |
| Replication | To ensure the reproducibility of the experiments, growth experiments for both microalga and bacteria were performed in independent biological replicates. Elemental analysis, monosaccharide composition analysis, sulfate group analysis and ELISA analysis were performed in three technical replicates for purified fucan. |
| Randomization | Randomization was not relevant as only in vitro experiments were performed. |
| Blinding | Blinding was not relevant as only in vitro experiments were performed. |

# Reporting for specific materials, systems and methods

We require information from authors about some types of materials, experimental systems and methods used in many studies. Here, indicate whether each material, system or method listed is relevant to your study. If you are not sure if a list item applies to your research, read the appropriate section before selecting a response.

## Materials & experimental systems

| n/a | Involved in the study |
|-----|----------------------|
| ☐ | ☒ Antibodies |
| ☒ | ☐ Eukaryotic cell lines |
| ☒ | ☐ Palaeontology and archaeology |
| ☒ | ☐ Animals and other organisms |
| ☒ | ☐ Clinical data |
| ☒ | ☐ Dual use research of concern |
| ☒ | ☐ Plants |

## Methods

| n/a | Involved in the study |
|-----|----------------------|
| ☒ | ☐ ChIP-seq |
| ☒ | ☐ Flow cytometry |
| ☒ | ☐ MRI-based neuroimaging |

## Antibodies

| | |
|---|---|
| Antibodies used | The following commercial antibodies were used. Product and lot numbers are specified when available.<br><br>From PlantProbes (Leeds, UK) and SeaProbes (Roscoff, France).<br>1. BAM1*<br>2. BAM2*<br>3. BAM3*<br>4. BAM4*<br>*PlantProbes has ceased activities since 2021, but BAM mAbs are still commercially available via SeaProbes.<br><br>From Sigma-Aldrich (MO, USA).<br>1. Anti-Rat IgG HR-Peroxidase secondary antibody (A9037, batch #SLCH8291) |
| Validation | - Fucoidan specificity of BAM mAbs demonstrated in Torode et al. (2015), PlosONE 10, e0118366.<br>- Validation information of Anti-Rat IgG A9037: www.sigmaaldrich.com/product/sigma/a9037 |

## Plants

| | |
|---|---|
| Seed stocks | NA |
| Novel plant genotypes | NA |
| Authentication | NA |

