## [Peer Review File · Nature Microbiology]

Phosphate deprivation restricts bacterial degradation of the marine polysaccharide fucoidan

Corresponding Author: Professor Jan-Hendrik Hehemann

Version 0:

Reviewer comments:

Reviewer #1

(Remarks to the Author)

The authors present a fascinating insight into the role of phosphate dependencies across both fucan synthesis by a marine macroalgae *Glossomastix* and fucan degradation by an isoalted Akkermansiaceae bacterium. *Glossomastix*-production of fucan was characterized in detail as to the structure of the polysaccharide. In concert efforts were made to recover bacteria that could degrade and consume fucan as a substrate. Isolation efforts produced one bacterium that was characterized in detail, whereby it was deduced that phosphate starvation restricted growth, which is in contrast to *Glossomastix*-production of fucan which occurs under nutrient limiting conditions. My background is in microbial structure and function, hence my comments below are reflective of that and do not cover elements I am not an expert in (such as polysaccharide structural characterisation. Overall, the paper is well written and presented, I have several comments for consideration.

- The phenotypic data observed for V_227 was very interesting, though its culture-centric characterisation makes it difficult to judge the ecological impact of this organism in a wider context. Such insight could be drawn from its isolation source and enrichment culture that led to its recovery. Given the authors observations that bacterial capabilities to digest fucan are rare (L209) and that this organism was seemingly quite specific to *Glossomastix*-fucan and unable to degrade other macroalgae fucans (L203-209), the question then arises if there is any co-occurring pattern between V_227 and *Glossomastix* at a larger ecological level? Were these macroalgae present at the sampling sites or why where these sites chosen, this is not clearly presented in the paper.
- The interspecies dynamics at play in the original enrichment may well have provided information as to why certain isolates “lost” their ability to degrade fucan or the strict requirements of additional “additional nutrients, trace elements and vitamins” (L180). Where any efforts made to characterize the enrichment cultures? Was V_227 the dominant fucan degrader in this enrichment? Were efforts made to genomically characterise the other fucan degraders that lost the phenotype (what taxonomy where they? Did they encode fucan degrading capabilities). Additional insight into the complex enrichments may well have assisted in understanding the seemingly lack of other bacteria callable of fucan degradation?
- L184 “enough nutrients” what does this mean?
- The screening of public data for organisms related to V_227 is a good idea, but not surprising that limited matches occurred when using DNA-DNA based searches. Could searches using AAI, CAZy families, CAZy/gene synteny help identify other distantly related organisms that still fulfill the same functional niche? Were the CAZyme determinants for fucan degradation identified from V_227 and could they be used to identify other fucan degrading populations?
- L253. This statement needs to be slightly modified, or more info provided. determined using which bacterium? this is phrased as though its been determined for V_227?
- L255. From which organism and are these data from samples where V_227 is present? This is not clear how this is connected to V_227

Reviewer #2

(Remarks to the Author)

Remarks to the Author:

This manuscript presents important findings about the interaction between the microalgal production of sulfated fucan and the bacterial degradation of these glycans under low-phosphorus (P) conditions. Fucoidans are a family of polysaccharides which contain a large amount of fucose and sulfate ester groups. They are produced by diatoms and brown algae and secreted during growth and after death as an extracellular matrix. This fucan matrix is of note because, contrary to other polysaccharides, it is resistant to bacterial degradation, allowing particulate fucan to persist in the ocean and be exported to depth, providing a mechanism for carbon sequestration. In this paper, the authors present evidence that low phosphate concentrations can slow the rate of bacterial fucan degradation, without impacting algal fucan production, leading to a stabilization of fucan, which in turn, may help algal cells compete with heterotrophic bacteria for nutrients, and may provide a pathway for the long-term sequestration of fucan-carbon.

The manuscript is well written and clearly guides the reader through the arguments being presented. The data are of high quality. I was particularly impressed with the characterization of the fucan structure. The methodologies employed and the analyses performed were of exceptional, and they are well described in the manuscript. This paper also presents novel evidence of nutrient limitation on the uptake of a specific carbon substrate by marine heterotrophs. Previous works demonstrate that nutrient limitation can affect carbon metabolism by enhancing the uptake of specific substrates¹, or by slowing carbon metabolism as a whole.² But this study is, to my knowledge, the first demonstration of nutrient limitation leading to the suppression of uptake of a specific carbon substrate. As such, I support the publication of this work in some form, although there are some concerns that I would like to see addressed before publication in this journal.

General Concerns:

One concern I have with this work is the difficulty in untangling the relationship between P concentration and cell response. As it is presented, each experimental treatment is denoted based on its initial concentration of P, but no data are presented showing the changes in dissolved P over time. Thus, it is unclear how the differential responses observed between treatments directly relate to P concentration. The initial P concentration in the “phosphate limited” treatment is 2-5 times higher than what is typically observed in the English Channel (200-400 nM), where PLY432 was isolated from.³ As the concentration of P throughout the growth experiments is not presented, and the growth rates of PLY432 showed no significant difference between P treatments, it is unclear at what point PLY432 begins experiencing P limitation. To fully support the conclusion that PLY432 fucan production is unaffected by P limitation, I believe these data, or additional evidence of P stress by PLY432 such as biomarkers for P stress, are needed.

In the discussion on quantitative glycan carbon accounting (beginning at line 301), it is unclear if the data are normalized by the number of cells in each treatment. The per cell uptake of the polysaccharides would be a useful visualization that would remove the effects of growth from the effects of reduced C uptake. Additionally, the data presented in this section appear to be at odds with the data shown in Fig. 6C. The body text states that, at 1 μM P, more Laminarin-C is converted than Fucan-C. Fig. 6C and the data in Supplementary Table 14 show that more Fucan-C is converted.

Finally, at times, the readability of this manuscript is lacking, resulting from confusing sentence structure and turns of phrase. The work would benefit from editing by someone with full professional proficiency in English.

Specific Comments:

Ln 23-25 Later in the text, the authors state that phosphate concentration didn't affect the maximum synthesis of fucan. Thus, the statement that phosphate limitation enhanced fucan synthesis is not clear supported.

Ln 53 The work should state that the strain of *Glossomastix* was isolated from the English Channel instead of the Atlantic Ocean. This provides more context for the work.

Ln 86: The terminology in this sentence should be clarified. Does the phrase “not growing” mean that the culture has reached stationary or senescence/death phase?

Ln 90: The list of sugars which do and don't increase in concentration are contradictory; galactose is listed twice. I believe the first list of sugars is supposed to include galacturonate.

Ln 105: This conclusion is difficult to see from the data presented. An analysis of the growth curves to determine when the acceleration of growth becomes zero would provide a clearer metric of the change in growth patterns with changing P concentrations. See Navarro-Pérez et al.⁴ as an example.

Ln 288: Without measuring dissolved phosphate at the end of the experiment, I'm not sure how you can justify this assumption.

Ln 303 and 461: Was a recovery analysis of the HPAEC-PAD method performed? Hydrolysis of polysaccharides to yield monosaccharides can result in poor recoveries, especially with partially degraded polysaccharides.

Ln 365: “of” is repeated in this line.

Ln 630: Authors should provide the polysaccharide-C concentration given in each treatment as those data would make interpretation of the data, particularly figure 6C more facile.

Ln 720: Consider depositing your NMR data in a repository such as NMRxiv.

Ln 983: “*classis nova*” should be italicized.

Ln 991: “Biogeochemistry of marine dissolved organic matter” should be italicized, not in quotations.

Ln 1002: The number “1” should be superscripted in “¹H-NMR”

Ln 1016: “Closterium” and “Nutrient Limitation” should not be capitalized.

Ln 1019: “Closterium” should not be capitalized.

Ln 1057: “can” should be capitalized.

Ln 1066: “cholerae” should be italicized.

Ln 1141: The isotope mass numbers in this title should be superscripted

References cited in this review:

1. Liang, Z., Letscher, R. T. & Knapp, A. N. Dissolved organic phosphorus concentrations in the surface ocean controlled by both phosphate and iron stress. *Nat. Geosci.* 15, 651–657 (2022).
2. Fourquez, M. et al. Effects of iron limitation on growth and carbon metabolism in oceanic and coastal heterotrophic bacteria. *Limnology and Oceanography* 59, 349–360 (2014).
3. Jordan, M. B. & Joint, I. Seasonal variation in nitrate:phosphate ratios in the English Channel 1923–1987. *Estuarine, Coastal and Shelf Science* 46, 157–164 (1998).
4. Navarro-Pérez, M. L., Fernández-Calderón, M. C. & Vadillo-Rodríguez, V. Decomposition of growth curves into growth rate and acceleration: A novel procedure to monitor bacterial growth and the time-dependent effect of antimicrobials. *Applied and Environmental Microbiology* 88, e01849-21 (2022)

Reviewer #3

(Remarks to the Author)

Key results

The study examined fucan synthesis by the marine microalgae *Glossomastix* (non-axenic) as well as fucan digestion by a bacterium in the Akkermansiaceae family, aiming to understand how limitation by organic phosphate affected both processes. *Glossomastix* kept in long term culture (~10 years) was found to exude fucan extracellularly. In the stationary phase, under P

limitation, fucan synthesis increased. The structure of the algae-generated fucan was characterized. Bacteria were isolated from the *Glossomastix* co-culture, seawater and porewater. With the exception of one bacterium, V_227, the isolated bacteria did not digest fucan. V_227 imported fucan into its cells and digested low concentrations of fucan if given additional trace elements and vitamins. V-227 was sequenced and used to look for its presence in the ocean (it was rare). Genome annotation showed genes consistent with glycan digestion and high affinity P uptake. Following limitation experiments (down to 1 μ M), glycan carbon accounting found fucan more stable with limited phosphate for this system. The main conclusions I drew from reading the manuscript were: organic phosphate limitation enhances fucan production by *Glossomastix* and fucan digestion is rare and/or very specific to particular organisms. Additional (in my assessment, unwarranted) conclusions were made by the authors, including that fucan plays a key role in algae-bacteria interactions, allowing for algal persistence and CO₂ concentrations in the ocean.

Validity

The conclusions of this paper were a stretch based on the data collected. I would suggest re-interpretation and reconsideration of conclusions. Three major concerns are as follows:

(1) This work relies solely on batch culture experiments to talk about the effects of phosphate limitation on fucan synthesis and/or digestion. There is a major flaw with this approach in that carrying capacity can be reached due to factors other than phosphate limitation, especially at the time scales of this experiment. What other resources were measured? How do we not know other factors influenced fucan dynamics? Moreover, cells are not in a stable state physiologically. To be convinced of the conclusions, I would need to see data from a chemostat experiment (or even semi-continuous growth experiment) that shows the same results.

(2) This study focuses on only one form of phosphate, B-glycerol-phosphate, but conclusions talk about phosphate limitation generally. No experiments were done on inorganic phosphate. How do conclusions about the ocean change if only organic phosphate limitation affects fucan synthesis/digestion? Either experiments should be added or the conclusions tamped down.

(3) The *Glossomastix* used was not axenic. How do we know that the production of fucan was driven by the algae and not algae-bacterial interactions within the culture? I didn't see taxonomy data of all bacteria (not just culturable bacteria) associated with the algae. How do these taxa affect the performance of the algae? Traits?

(4) At times, the framing of the paper calls into question credibility of the authors. For example, line 344: "So how do algae persist?" Because they use sunlight and CO₂. Complementary niches. I am not sure what to do with this sentence. Is it a joke?

Other conclusions were drawn that seemed far-reaching and need more support. These are outlined below.

a) Lines 32-33. "We conclude phosphate starvation constrains the ability of bacteria to digest fucan, which evolves to maintain stability around algal cells and consequentially also to keep carbon dioxide in the ocean." I don't know where the evidence is for this conclusion. The study did not look at evolved traits, algae-bacteria interactions in the phycosphere, or resources in the phycosphere. In other places, a conclusion is drawn that algae are given more time/space to resources. But I do not see data that support this. Are there resource data? Time resolved data?

b) Line 103 and afterwards. "Algae continued fucan exudation despite phosphate limitation.... Phosphate concentration neither affected the maximum, synthesis rate per cell and day ($r = -0.42$, $P > 108$ 0.5) nor the growth rate ($r = -0.89$, $P > 0.05$) (Fig. 2d and Extended Data Fig. 2d)." But looking at the supplementary figures, changing the phosphate concentration changed the carrying capacity. It also looks like the maximum specific growth rate changes. I question the conclusion here based on these data as well as experimental design. Also, how do the concentrations in the experiment to concentrations in the ocean? It seems that the concentrations are high. Is it possible that phosphate was simply not limiting enough to see effects on fucan synthesis? I would also like to see these concentrations in context of other papers. For example, XX et al. considered P-limited 200 nM and P replete 45 μ M. What about cell densities? Globally, are the experiments relevant to what cells would see in the ocean?

c) Lines 180-182. "These results indicate bacteria can digest the fucan but only when the environment provides enough additional resources and when the fucan is not too concentrated." This study focused on a single bacterium and the authors have demonstrated that there aren't a lot of cells that can digest the fucan. Maybe this one cell was just not happy in the culture conditions present. Alternatively, thinking about the broader bacterial community and the central story here that P limitation changes how fucan digestion is regulated -- maybe there are just not a lot of fucan digesting bacteria (authors present evidence for this) and P-limitation shifts the community to strains that cannot digest fucan. This is a storyline that crops in at times in the paper, but I am not sure which story the authors are trying to tell.

Significance

The paper has some significance for understanding C cycling and algal-bacteria interactions in the ocean, but I am not sure how much given how rare fucan digesting bacteria are in the ocean.

Data and methodology

On approach, I question the relevance of the *Glossomastix* that had been in culture for ~10 years to ocean dynamics. It is a lab-rat. Is there a way to look at other cultures that have been isolated more recently?

Lines 275-280 and Extended Data Fig. 7: there are missing data. in the third panel (residual carbohydrates), phosphate concentrations go down to 0, but in the a, b, d, f, it only goes to 5 or 10 - where are the other data? It looks like they have been removed and this is a big red flag to me.

Lines 284-285 and Extended Figures: I don't see a difference in these slopes in 6b. Moreover in the curves, there are also decreases in the laminarian. The difference is in the shape of the curve at 1 and 5 μM P for glycan. This requires further exploration. I might standardize the growth relative to the maximum and then look at differences.

Analytical approach

Stats in parentheses are lacking (just naked p-values)- what was the test? where are other metrics?

Parameters used on CheckM should be listed.

Why is coverage for HMMER 35%?

Suggested improvements

I would add a set of chemostat experiments or even semi-continuous growth experiments that show results consistent with the ones presented, so that P limitation can be directly linked to processes described. Here, I would also take P limitation to lower concentrations.

There seem to be two storylines in the paper. The first is the bacteria can digest fucan and P limitation changes these dynamics. The second is that P limitation changes the community (i.e., distribution of fucan digesting bacteria, line 233). These hypotheses need to be better spelled out in the beginning and sifted through.

For a paper where so much rests on P limitation, Extended Data Fig 2 should be a main figure.

I would like to see some discussion of B-glycerol phosphate as a phosphate source for this algae strain.

Lines 295-299: this is well articulated, I might use it in other sections, to provide background/context/thread throughout.

There are a number of places in the text that don't read well or have tense issues (e.g., line 86, "Not growing algae continue synthesis of fucan under nutrient limiting conditions.", line 101, "The data shows a sustained population of algal cells sustains the fucan sequestration pathway."). The manuscript deserves another sentence level read for clarity.

Clarity and context

The section between 253 and 263 needs more clarity and context. The sentence 253-254 is cited, but the citation is about the first part of the sentence, not anything to do with algal blooms. All we know is that Pst ABC system has an affinity constant 200 - 400 nm. Experiments were done at 1 μM , which is more than 2X this affinity constant. Lines 255-257 -- These all seem to be different papers. I also don't understand how disparate references can be linked to make a conclusion here. If that is not true, more context is needed in the text. Lines 260-263, I am missing this link. What I know here is that these bacteria experience P limitation, they have evolved transporters to get very low amounts of P, how is that related to glycan here?

Decision Letter:

29th May 2024

Dear Professor Hehemann,

Thank you for your patience while your manuscript "Phosphate starvation stops bacteria digesting algal fucan that sequesters carbon" was under peer-review at Nature Microbiology. It has now been seen by 3 referees, whose expertise and comments you will find at the end of this email. Although they find your work of some potential interest, they have raised a number of concerns that will need to be addressed before we can consider publication of the work in Nature Microbiology.

In particular, from the reports it is clear thought some additional experimental work is needed (P concentration tests, as well as chemostat or semi-continuous cultures to better assess growth dynamics). Furthermore, the referees point out the need for additional analyses (CAZy analysis, and more robust stats), as well as additional analyses to assess the environmental relevance of these findings (is this happening in the field, with other organisms?)--we agree that these revisions would strengthen the paper.

Should further experimental data allow you to address these criticisms, we would be happy to look at a revised manuscript.

Please include a data availability statement as a separate section after Methods but before references, under the heading "Data Availability". This section should inform readers about the availability of the data used to support the conclusions of your study. This information includes accession codes to public repositories (data banks for protein, DNA or RNA sequences, microarray,

proteomics data etc...), references to source data published alongside the paper, unique identifiers such as URLs to data repository entries, or data set DOIs, and any other statement about data availability. At a minimum, you should include the following statement: "The data that support the findings of this study are available from the corresponding author upon request", mentioning any restrictions on availability. If DOIs are provided, we also strongly encourage including these in the Reference list (authors, title, publisher (repository name), identifier, year). For more guidance on how to write this section please see: <http://www.nature.com/authors/policies/data/data-availability-statements-data-citations.pdf>

* If you have not done so already we suggest that you begin to revise your manuscript so that it conforms to our Article format instructions at <http://www.nature.com/nmicrobiol/info/final-submission>. Refer also to any guidelines provided in this letter.

When submitting the revised version of your manuscript, please pay close attention to our [href="https://www.nature.com/nature-portfolio/editorial-policies/image-integrity">Digital Image Integrity Guidelines](https://www.nature.com/nature-portfolio/editorial-policies/image-integrity) and to the following points below:

Link Redacted

Note: This url links to your confidential homepage and associated information about manuscripts you may have submitted or be reviewing for us. If you wish to forward this e-mail to co-authors, please delete this link to your homepage first.

Nature Microbiology is committed to improving transparency in authorship. As part of our efforts in this direction, we are now requesting that all authors identified as 'corresponding author' on published papers create and link their Open Researcher and Contributor Identifier (ORCID) with their account on the Manuscript Tracking System (MTS), prior to acceptance. This applies to primary research papers only. ORCID helps the scientific community achieve unambiguous attribution of all scholarly contributions. You can create and link your ORCID from the home page of the MTS by clicking on 'Modify my Springer Nature account'. For more information please visit [please visit www.springernature.com/orcid](http://www.springernature.com/orcid).

If you wish to submit a suitably revised manuscript we would hope to receive it within 6 months. If you cannot send it within this time, please let us know. We will be happy to consider your revision, even if a similar study has been accepted for publication at Nature Microbiology or published elsewhere (up to a maximum of 6 months).

Yours sincerely,

Reviewer Expertise:

Referee #1: microbial ecology, carbon metabolism, 'omics

Referee #2: geo- and biogeochemistry, glycobiology, marine microbiology

Referee #3: marine microbial ecology, phytoplankton polysaccharide metabolism

Reviewer Comments:

Reviewer #1 (Remarks to the Author):

The authors present a fascinating insight into the role of phosphate dependencies across both fucan synthesis by a marine macroalgae *Glossomastix* and fucan degradation by an isoalted Akkermansiaceae bacterium. *Glossomastix*-production of fucan was characterized in detail as to the structure of the polysaccharide. In concert efforts were made to recover bacteria that could

degrade and consume fucan as a substrate. Isolation efforts produced one bacterium that was characterized in detail, whereby it was deduced that phosphate starvation restricted growth, which is in contrast to *Glossomastix*-production of fucan which occurs under nutrient limiting conditions. My background is in microbial structure and function, hence my comments below are reflective of that and do not cover elements I am not an expert in (such as polysaccharide structural characterisation). Overall, the paper is well written and presented, I have several comments for consideration.

- The phenotypic data observed for V_227 was very interesting, though its culture-centric characterisation makes it difficult to judge the ecological impact of this organism in a wider context. Such insight could be drawn from its isolation source and enrichment culture that led to its recovery. Given the authors observations that bacterial capabilities to digest fucan are rare (L209) and that this organism was seemingly quite specific to *Glossomastix*-fucan and unable to degrade other macroalgae fucans (L203-209), the question then arises if there is any co-occurring pattern between V_227 and *Glossomastix* at a larger ecological level? Were these macroalgae present at the sampling sites or why were these sites chosen, this is not clearly presented in the paper.
- The interspecies dynamics at play in the original enrichment may well have provided information as to why certain isolates “lost” their ability to degrade fucan or the strict requirements of additional “additional nutrients, trace elements and vitamins” (L180). Where any efforts made to characterize the enrichment cultures? Was V_227 the dominant fucan degrader in this enrichment? Were efforts made to genomically characterise the other fucan degraders that lost the phenotype (what taxonomy were they? Did they encode fucan degrading capabilities). Additional insight into the complex enrichments may well have assisted in understanding the seemingly lack of other bacteria capable of fucan degradation?
- L184 “enough nutrients” what does this mean?
- The screening of public data for organisms related to V_227 is a good idea, but not surprising that limited matches occurred when using DNA-DNA based searches. Could searches using AAI, CAZy families, CAZy/gene synteny help identify other distantly related organisms that still fulfill the same functional niche? Were the CAZyme determinants for fucan degradation identified from V_227 and could they be used to identify other fucan degrading populations?
- L253. This statement needs to be slightly modified, or more info provided. determined using which bacterium? this is phrased as though its been determined for V_227?
- L255. From which organism and are these data from samples where V_227 is present? This is not clear how this is connected to V_227

Reviewer #2 (Remarks to the Author):

Remarks to the Author:

This manuscript presents important findings about the interaction between the microalgal production of sulfated fucan and the bacterial degradation of these glycans under low-phosphorus (P) conditions. Fucoidans are a family of polysaccharides which contain a large amount of fucose and sulfate ester groups. They are produced by diatoms and brown algae and secreted during growth and after death as an extracellular matrix. This fucan matrix is of note because, contrary to other polysaccharides, it is resistant to bacterial degradation, allowing particulate fucan to persist in the ocean and be exported to depth, providing a mechanism for carbon sequestration. In this paper, the authors present evidence that low phosphate concentrations can slow the rate of bacterial fucan degradation, without impacting algal fucan production, leading to a stabilization of fucan, which in turn, may help algal cells compete with heterotrophic bacteria for nutrients, and may provide a pathway for the long-term sequestration of fucan-carbon.

The manuscript is well written and clearly guides the reader through the arguments being presented. The data are of high quality. I was particularly impressed with the characterization of the fucan structure. The methodologies employed and the analyses performed were of exceptional, and they are well described in the manuscript. This paper also presents novel evidence of nutrient limitation on the uptake of a specific carbon substrate by marine heterotrophs. Previous works demonstrate that nutrient limitation can affect carbon metabolism by enhancing the uptake of specific substrates¹, or by slowing carbon metabolism as a whole.² But this study is, to my knowledge, the first demonstration of nutrient limitation leading to the suppression of uptake of a specific carbon substrate. As such, I support the publication of this work in some form, although there are some concerns that I would like to see addressed before publication in this journal.

General Concerns:

One concern I have with this work is the difficulty in untangling the relationship between P concentration and cell response. As it is presented, each experimental treatment is denoted based on its initial concentration of P, but no data are presented showing the changes in dissolved P over time. Thus, it is unclear how the differential responses observed between treatments directly relate to P concentration. The initial P concentration in the “phosphate limited” treatment is 2-5 times higher than what is typically observed in the English Channel (200-400 nM), where PLY432 was isolated from.³ As the concentration of P throughout the growth experiments is not presented, and the growth rates of PLY432 showed no significant difference between P treatments, it is unclear at what point PLY432 begins experiencing P limitation. To fully support the conclusion that PLY432 fucan production is unaffected by P limitation, I believe these data, or additional evidence of P stress by PLY432 such as biomarkers for P stress, are needed.

In the discussion on quantitative glycan carbon accounting (beginning at line 301), it is unclear if the data are normalized by the number of cells in each treatment. The per cell uptake of the polysaccharides would be a useful visualization that would remove the effects of growth from the effects of reduced C uptake. Additionally, the data presented in this section appear to be at odds with the data shown in Fig. 6C. The body text states that, at 1 μM P, more Laminarin-C is converted than Fucan-C. Fig. 6C and the data in Supplementary Table 14 show that more Fucan-C is converted.

Finally, at times, the readability of this manuscript is lacking, resulting from confusing sentence structure and turns of phrase. The work would benefit from editing by someone with full professional proficiency in English.

Specific Comments:

Ln 23-25 Later in the text, the authors state that phosphate concentration didn't affect the maximum synthesis of fucan. Thus, the statement that phosphate limitation enhanced fucan synthesis is not clearly supported.

Ln 53 The work should state that the strain of *Glossomastix* was isolated from the English Channel instead of the Atlantic Ocean. This provides more context for the work.

Ln 86: The terminology in this sentence should be clarified. Does the phrase “not growing” mean that the culture has reached stationary or senescence/death phase?

Ln 90: The list of sugars which do and don't increase in concentration are contradictory; galactose is listed twice. I believe the first list of sugars is supposed to include galacturonate.

Ln 105: This conclusion is difficult to see from the data presented. An analysis of the growth curves to determine when the acceleration of growth becomes zero would provide a clearer metric of the change in growth patterns with changing P concentrations. See Navarro-Pérez et al.4 as an example.

Ln 288: Without measuring dissolved phosphate at the end of the experiment, I'm not sure how you can justify this assumption.

Ln 303 and 461: Was a recovery analysis of the HPAEC-PAD method performed? Hydrolysis of polysaccharides to yield monosaccharides can result in poor recoveries, especially with partially degraded polysaccharides.

Ln 365: “of” is repeated in this line.

Ln 630: Authors should provide the polysaccharide-C concentration given in each treatment as those data would make interpretation of the data, particularly figure 6C more facile.

Ln 720: Consider depositing your NMR data in a repository such as NMRxiv.

Ln 983: “classis nova” should be italicized.

Ln 991: “Biogeochemistry of marine dissolved organic matter” should be italicized, not in quotations.

Ln 1002: The number “1” should be superscripted in “¹H-NMR”

Ln 1016: “Closterium” and “Nutrient Limitation” should not be capitalized.

Ln 1019: “Closterium” should not be capitalized.

Ln 1057: “can” should be capitalized.

Ln 1066: “cholerae” should be italicized.

Ln 1141: The isotope mass numbers in this title should be superscripted

References cited in this review:

1. Liang, Z., Letscher, R. T. & Knapp, A. N. Dissolved organic phosphorus concentrations in the surface ocean controlled by both phosphate and iron stress. *Nat. Geosci.* 15, 651–657 (2022).
2. Fourquez, M. et al. Effects of iron limitation on growth and carbon metabolism in oceanic and coastal heterotrophic bacteria. *Limnology and Oceanography* 59, 349–360 (2014).
3. Jordan, M. B. & Joint, I. Seasonal variation in nitrate:phosphate ratios in the english channel 1923–1987. *Estuarine, Coastal and Shelf Science* 46, 157–164 (1998).
4. Navaro-Pérez, M. L., Fernández-Calderón, M. C. & Vadillo-Rodríguez, V. Decomposition of growth curves into growth rate and acceleration: A novel procedure to monitor bacterial growth and the time-dependent effect of antimicrobials. *Applied and Environmental Microbiology* 88, e01849-21 (2022)

Reviewer #3 (Remarks to the Author):

Key results

The study examined fucan synthesis by the marine microalgae *Glossomastix* (non-axenic) as well as fucan digestion by a bacterium in the Akkermansiaceae family, aiming to understand how limitation by organic phosphate affected both processes. *Glossomastix* kept in long term culture (~10 years) was found to exude fucan extracellularly. In the stationary phase, under P limitation, fucan synthesis increased. The structure of the algae-generated fucan was characterized. Bacteria were isolated from the *Glossomastix* co-culture, seawater and porewater. With the exception of one bacterium, V_227, the isolated bacteria did not digest fucan. V_227 imported fucan into its cells and digested low concentrations of fucan if given additional trace elements and vitamins. V-227 was sequenced and used to look for its presence in the ocean (it was rare). Genome annotation showed genes consistent with glycan digestion and high affinity P uptake. Following limitation experiments (down to 1 μ M), glycan carbon accounting found fucan more stable with limited phosphate for this system. The main conclusions I drew from reading the manuscript were: organic phosphate limitation enhances fucan production by *Glossomastix* and fucan digestion is rare and/or very specific to particular organisms. Additional (in my assessment, unwarranted) conclusions were made by the authors, including that fucan plays a key role in algae-bacteria interactions, allowing for algal persistence and CO₂ concentrations in the ocean.

Validity

The conclusions of this paper were a stretch based on the data collected. I would suggest re-interpretation and reconsideration of conclusions. Three major concerns are as follows:

(1) This work relies solely on batch culture experiments to talk about the effects of phosphate limitation on fucan synthesis and/or digestion. There is a major flaw with this approach in that carrying capacity can be reached due to factors other than phosphate limitation, especially at the time scales of this experiment. What other resources were measured? How do we not know other factors influenced fucan dynamics? Moreover, cells are not in a stable state physiologically. To be convinced of the conclusions, I would need to see data from a chemostat experiment (or even semi-continuous growth experiment) that shows the same results.

(2) This study focuses on only one form of phosphate, B-glycerol-phosphate, but conclusions talk about phosphate limitation generally. No experiments were done on inorganic phosphate. How do conclusions about the ocean change if only organic phosphate limitation affects fucan synthesis/digestion? Either experiments should be added or the conclusions tamped down.

(3) The *Glossomastix* used was not axenic. How do we know that the production of fucan was driven by the algae and not algae-bacterial interactions within the culture? I didn't see taxonomy data of all bacteria (not just culturable bacteria) associated with

the algae. How do these taxa affect the performance of the algae? Traits?

(4) At times, the framing of the paper calls into question credibility of the authors. For example, line 344: "So how do algae persist?" Because they use sunlight and CO₂. Complementary niches. I am not sure what to do with this sentence. Is it a joke?

Other conclusions were drawn that seemed far-reaching and need more support. These are outlined below.

a) Lines 32-33. "We conclude phosphate starvation constrains the ability of bacteria to digest fucan, which evolves to maintain stability around algal cells and consequentially also to keep carbon dioxide in the ocean." I don't know where the evidence is for this conclusion. The study did not look at evolved traits, algae-bacteria interactions in the phycosphere, or resources in the phycosphere. In other places, a conclusion is drawn that algae are given more time/space to resources. But I do not see data that support this. Are there resource data? Time resolved data?

b) Line 103 and afterwards. "Algae continued fucan exudation despite phosphate limitation.... Phosphate concentration neither affected the maximum, synthesis rate per cell and day ($r = -0.42$, $P > 108\ 0.5$) nor the growth rate ($r = -0.89$, $P > 0.05$) (Fig. 2d and Extended Data Fig. 2d)." But looking at the supplementary figures, changing the phosphate concentration changed the carrying capacity. It also looks like the maximum specific growth rate changes. I question the conclusion here based on these data as well as experimental design. Also, how do the concentrations in the experiment to concentrations in the ocean? It seems that the concentrations are high. Is it possible that phosphate was simply not limiting enough to see effects on fucan synthesis? I would also like to see these concentrations in context of other papers. For example, XX et al. considered P-limited 200 nM and P replete 45 μ M. What about cell densities? Globally, are the experiments relevant to what cells would see in the ocean?

c) Lines 180-182. "These results indicate bacteria can digest the fucan but only when the environment provides enough additional resources and when the fucan is not too concentrated." This study focused on a single bacterium and the authors have demonstrated that there aren't a lot of cells that can digest the fucan. Maybe this one cell was just not happy in the culture conditions present. Alternatively, thinking about the broader bacterial community and the central story here that P limitation changes how fucan digestion is regulated -- maybe there are just not a lot of fucan digesting bacteria (authors present evidence for this) and P-limitation shifts the community to strains that cannot digest fucan. This is a storyline that crops in at times in the paper, but I am not sure which story the authors are trying to tell.

Significance

The paper has some significance for understanding C cycling and algal-bacteria interactions in the ocean, but I am not sure how much given how rare fucan digesting bacteria are in the ocean.

Data and methodology

On approach, I question the relevance of the *Glossomastix* that had been in culture for ~10 years to ocean dynamics. It is a lab-rat. Is there a way to look at other cultures that have been isolated more recently?

Lines 275-280 and Extended Data Fig. 7: there are missing data. in the third panel (residual carbohydrates), phosphate concentrations go down to 0, but in the a, b, d, f, it only goes to 5 or 10 - where are the other data? It looks like they have been removed and this is a big red flag to me.

Lines 284-285 and Extended Figures: I don't see a difference in these slopes in 6b. Moreover in the curves, there are also decreases in the laminarian. The difference is in the shape of the curve at 1 and 5 μ M P for glycan. This requires further exploration. I might standardize the growth relative to the maximum and then look at differences.

Analytical approach

Stats in parentheses are lacking (just naked p-values)- what was the test? where are other metrics?

Parameters used on CheckM should be listed.

Why is coverage for HMMER 35%?

Suggested improvements

I would add a set of chemostat experiments or even semi-continuous growth experiments that show results consistent with the ones presented, so that P limitation can be directly linked to processes described. Here, I would also take P limitation to lower concentrations.

There seem to be two storylines in the paper. The first is the bacteria can digest fucan and P limitation changes these dynamics. The second is that P limitation changes the community (i.e., distribution of fucan digesting bacteria, line 233). These hypotheses need to be better spelled out in the beginning and sifted through.

For a paper where so much rests on P limitation, Extended Data Fig 2 should be a main figure.

I would like to see some discussion of B-glycerol phosphate as a phosphate source for this algae strain.

Lines 295-299: this is well articulated, I might use it in other sections, to provide background/context/thread throughout.

There are a number of places in the text that don't read well or have tense issues (e.g., line 86, "Not growing algae continue synthesis of fucan under nutrient limiting conditions.", line 101, "The data shows a sustained population of algal cells sustains

the fucan sequestration pathway.”). The manuscript deserves another sentence level read for clarity.

Clarity and context

The section between 253 and 263 needs more clarity and context. The sentence 253-254 is cited, but the citation is about the first part of the sentence, not anything to do with algal blooms. All we know is that Pst ABC system has an affinity constant 200 - 400 nM. Experiments were done at 1 μ M, which is more than 2X this affinity constant. Lines 255-257 -- These all seem to be different papers. I also don't understand how disparate references can be linked to make a conclusion here. If that is not true, more context is needed in the text. Lines 260-263, I am missing this link. What I know here is that these bacteria experience P limitation, they have evolved transporters to get very low amounts of P, how is that related to glycan here?

Version 1:

Reviewer comments:

Reviewer #1

(Remarks to the Author)

Authors have addressed all concerns. Congratulations on a great piece of work!

Reviewer #2

(Remarks to the Author)

Remarks to the Author:

The authors have done a good job addressing the major concerns that I had in the first submitted version of this paper. The addition of the semi-continuous growth experiment was particularly important in strengthening the manuscript. While there are still some minor revisions that are needed to improve the clarity of this work, I feel that authors have substantially improved this manuscript and I am happy to recommend it for publication after these minor errors are addressed.

General Remarks:

I was surprised that the V_227 bacteria only drew down phosphate to $\sim 1 \mu$ M in the growth yield experiments (Extended figures 10-11), as this level of phosphate is still considered replete and would not typically be low enough to trigger the phosphate regulons that control high affinity ABC phosphate transporters. It might be worth clarifying if this is a result of experimental methodology and design, or if V_227 does not draw down phosphate below $\sim 1 \mu$ M. If more significant drawdown did occur, consider changing the y-axis scale to make it easier to interpret the figure.

Specific Comments:

Ln 141: There should be a comma after diatoms

Ln 193: "Enrichments" should not be plural

Ln 209: there should be an "and" between rumen and ocean

The paragraph beginning at Ln 212: This paragraph was hard to follow. I suggest reviewing it for clarity and restructuring it so it is clear to the readers which "fucoïdan" is being discussed.

Ln 241 & Ln 382: Consider also citing Granzow and Repeta (2024) ES&T, which also found acylpolysaccharides make up $\sim 20 \mu$ M of DOC in the surface ocean.

Ln 246: While P is often limiting in the Mediterranean and Sargasso sea, many other oligotrophic regions of the ocean are primarily N limited and only secondarily limited by P (See Moore et al 2013 Nat. Geosci). I think you can keep this sentence largely unchanged, but I would temper the statement to be more precise.

Ln 247-251: The idea about Verrucomicrobiota becoming more abundant on sinking particles due to increased dissolved phosphate is interesting but not well described here. Consider rewriting or removing, as the dynamics between dissolved nutrient concentrations and microbial succession on sinking particles is complicated and deserves more nuance than described here.

Ln 277: Consider changing to "may be important for and may influence" to improve readability

Ln 293: What does "low" mean in this sentence?

Ln 313: What linear model are you referring to? It's not clear in the text.

Ln 345-346: I'm not sure what this sentence is saying. Please revise

Ln 347: What do you mean by "Without knowing what we lost?" is this in reference to monosaccharide content?

Ln 352: You use the term "stability index" which would typically imply a formula that can be used to calculate the stability of a glycan, but no formula is given. Consider changing to a different term, such as relationship or characteristic.

Ln 355-356: I am not sure what you mean by this sentence. Do you mean that algae have evolved to increase their glycan structural complexity as a survival strategy? Or do you mean that complex glycans are synthesized on the outer membrane of algae?

Ln 382-383: The 2nd clause in this sentence makes it difficult to follow. Consider rephrasing.

Ln 391: The term "carbon energy compounds" is confusing. Consider removing "energy" from the sentence.

Ln 392: Metabolites are not inherently labile or easy to consume. Fucoïdan is a metabolite. Picrotoxins are metabolites. You should be more specific about which metabolites you are talking about in this sentence.

Ln 398: "Contributes" should be "contribute".

Ln 413-414: There should be commas after "show" and "taxa".

Reviewer #3

(Remarks to the Author)

I approve the publication of this work as is.

Decision Letter:

Our ref: NMICROBIOL-24041026A

6th November 2025

Dear Dr. Hehemann,

Thank you for submitting your revised manuscript "Phosphate deprivation restricts the degradation of fucoidan by a *Verrucomicrobium*" (NMICROBIOL-24041026A). It has now been seen by the original referees and their comments are below. The reviewers find that the paper has improved in revision, and therefore we'll be happy in principle to publish it in *Nature Microbiology*, pending minor revisions to satisfy the referees' final requests and to comply with our editorial and formatting guidelines.

Thank you again for your interest in *Nature Microbiology*. Please do not hesitate to contact me if you have any questions.

Sincerely,

Reviewer #1 (Remarks to the Author):

Authors have addressed all concerns. Congratulations on a great piece of work!

Reviewer #2 (Remarks to the Author):

Remarks to the Author:

The authors have done a good job addressing the major concerns that I had in the first submitted version of this paper. The addition of the semi-continuous growth experiment was particularly important in strengthening the manuscript. While there are still some minor revisions that are needed to improve the clarity of this work, I feel that authors have substantially improved this manuscript and I am happy to recommend it for publication after these minor errors are addressed.

General Remarks:

I was surprised that the V_227 bacteria only drew down phosphate to ~1 μM in the growth yield experiments (Extended figures 10-11), as this level of phosphate is still considered replete and would not typically be low enough to trigger the pho regulons that control high affinity ABC phosphate transporters. It might be worth clarifying if this is a result of experimental methodology and design, or if V_227 does not draw down phosphate below ~1 μM . If more significant drawdown did occur, consider changing the y-axis scale to make it easier to interpret the figure.

Specific Comments:

Ln 141: There should be a comma after diatoms

Ln 193: "Enrichments" should not be plural

Ln 209: there should be an "and" between rumen and ocean

The paragraph beginning at Ln 212: This paragraph was hard to follow. I suggest reviewing it for clarity and restructuring it so it is clear to the readers which "fucoidan" is being discussed.

Ln 241 & Ln 382: Consider also citing Granzow and Repeta (2024) *ES&T*, which also found acylpolysaccharides make up ~ 20 μM of DOC in the surface ocean.

Ln 246: While P is often limiting in the Mediterranean and Sargasso sea, many other oligotrophic regions of the ocean are primarily N limited and only secondarily limited by P (See Moore et al 2013 *Nat. Geosci*). I think you can keep this sentence largely unchanged, but I would temper the statement to be more precise.

Ln 247-251: The idea about *Verrucomicrobiota* becoming more abundant on sinking particles due to increased dissolved phosphate is interesting but not well described here. Consider rewriting or removing, as the dynamics between dissolved nutrient concentrations and microbial succession on sinking particles is complicated and deserves more nuance than described here.

Ln 277: Consider changing to "may be important for and may influence" to improve readability

Ln 293: What does "low" mean in this sentence?

Ln 313: What linear model are you referring to? It's not clear in the text.

Ln 345-346: I'm not sure what this sentence is saying. Please revise

Ln 347: What do you mean by "Without knowing what we lost?" is this in reference to monosaccharide content?

Ln 352: You use the term “stability index” which would typically imply a formula that can be used to calculate the stability of a glycan, but no formula is given. Consider changing to a different term, such as relationship or characteristic.

Ln 355-356: I am not sure what you mean by this sentence. Do you mean that algae have evolved to increase their glycan structural complexity as a survival strategy? Or do you mean that complex glycans are synthesized on the outer membrane of algae?

Ln 382-383: The 2nd clause in this sentence makes it difficult to follow. Consider rephrasing.

Ln 391: The term “carbon energy compounds” is confusing. Consider removing “energy” from the sentence.

Ln 392: Metabolites are not inherently labile or easy to consume. Fucoidan is a metabolite. Picrotoxins are metabolites. You should be more specific about which metabolites you are talking about in this sentence.

Ln 398: “Contributes” should be “contribute”.

Ln 413-414: There should be commas after “show” and “taxa”.

Reviewer #3 (Remarks to the Author):

I approve the publication of this work as is.

Version 2:

Decision Letter:

10th December 2025

Dear Jan-Hendrik,

I am pleased to accept your Article "Phosphate deprivation restricts bacterial degradation of the marine polysaccharide fucoidan" for publication in Nature Microbiology. Thank you for having chosen to submit your work to us and many congratulations.

You may wish to make your media relations office aware of your accepted publication, in case they consider it appropriate to organize some internal or external publicity. Once your paper has been scheduled you will receive an email confirming the publication details. This is normally 3-4 working days in advance of publication. If you need additional notice of the date and time of publication, please let the production team know when you receive the proof of your article to ensure there is sufficient time to coordinate. Further information on our embargo policies can be found here:

<https://www.nature.com/authors/policies/embargo.html>

Authors may need to take specific actions to achieve compliance with funder and institutional open access mandates. If your research is supported by a funder that requires immediate open access (e.g. according to [a href="https://www.springernature.com/gp/open-science/plan-s-compliance"> Plan S principles](https://www.springernature.com/gp/open-science/plan-s-compliance) or the [a href="https://www.springernature.com/gp/open-science/us-federal-agency-compliance"> NIH public access policy](https://www.springernature.com/gp/open-science/us-federal-agency-compliance)) then you should select the gold OA route, and we will direct you to the compliant route where possible. Because authors warrant under our subscription licensing terms that they haven't committed to licensing any version of their article under a licence inconsistent with the terms of our agreement – including the applicable embargo period – publication under the subscription model isn't suitable for authors whose funders require no embargo.

With kind regards,

P.S. Click on the following link if you would like to recommend Nature Microbiology to your librarian
<http://www.nature.com/subscriptions/recommend.html#forms>

** Visit the Springer Nature Editorial and Publishing website at http://editorial-jobs.springernature.com?utm_source=ejP_NMicro_email&utm_medium=ejP_NMicro_email&utm_campaign=ejp_NMicro for more information about our career opportunities. If you have any questions please click [here](mailto:editorial.publishing.jobs@springernature.com).

Open Access This Peer Review File is licensed under a Creative Commons Attribution 4.0 International License, which permits use, sharing, adaptation, distribution and reproduction in any medium or format, as long as you give appropriate credit to the original author(s) and the source, provide a link to the Creative Commons license, and indicate if changes were made. In cases where reviewers are anonymous, credit should be given to 'Anonymous Referee' and the source. The images or other third party material in this Peer Review File are included in the article's Creative Commons license, unless indicated otherwise in a credit line to the material. If material is not included in the article's Creative Commons license and your intended use is not permitted by statutory regulation or exceeds the permitted use, you will need to obtain permission directly

from the copyright holder.

**Point by point response to the comments of referees in relation to the manuscript**
**NMICROBIOL-24041026: “Phosphate starvation stops bacteria digesting algal**
**fucan that sequesters carbon”**

**Summary of the reviewers’ comments and our response**

We have received three extensive reviews. The reviewers appreciated the novel aspect
of this study that bacterial degradation of fucoidan is constrained by phosphate and that
this might contribute to the potential of fucoidan to become a carbon dioxide sink in the
ocean. For example, **Reviewer # 1** remarked: “*The authors present a fascinating insight*
*into the role of phosphate dependencies across both fucan synthesis by a marine*
*macrolalgae *Glossomastix* and fucan degradation by an isolated *Akkermansiaceae**
*bacterium. Moreover, we are glad that the reviewers recognized the tremendous effort*
*required to isolate and characterize unknown algal polysaccharides.” **Reviewer #2:** “*The*
*data are of high quality. I was particularly impressed with the characterization of the fucan*
*structure. The methodologies employed and the analyses performed were of exceptional,*
*and they are well described in the manuscript. This paper also presents novel evidence*
*of nutrient limitation on the uptake of a specific carbon substrate by marine heterotrophs.*
*But this study is, to my knowledge, the first demonstration of nutrient limitation leading to*
*the suppression of uptake of a specific carbon substrate.” **Reviewer #3:** “*The paper has*
*some significance for understanding C cycling and algal-bacteria interactions in the*
*oceans.”***

However, we also recognized the identified crucial aspects to improve the manuscript.
For example, **Reviewer #2** remarked, “*As such, I support the publication of this work in*
*some form, although there are some concerns that I would like to see addressed before*
*publication in this journal.” **Reviewer #2** requests: “*One concern I have with this work is*
*the difficulty in untangling the relationship between P concentration and cell response. As*
*it is presented, each experimental treatment is denoted based on its initial concentration*
*of P, but no data are presented showing the changes in dissolved P over time. Thus, it is**

*unclear how the differential responses observed between treatments directly relate to P*
*concentration.” Hence, we have included phosphate quantification in the revised*
**manuscript.** Moreover, **Reviewer #2** but also **Reviewer #3** have requested more
evidence for fucoidan synthesis by the algae in phosphate limiting conditions and
proposed semi-continuous growth experiments. **To address this request, we have**
**included a series of additional semi-continuous growth experiments of algal for**
**one month.** Moreover, **Reviewer #3** noted: “*Additional (in my assessment, unwarranted)*
*conclusions were made by the authors, including that fucan plays a key role in algae-*
*bacteria interactions, allowing for algal persistence and CO₂ concentrations in the ocean.”*
**To clarify that this work is a model system for our previous finding that fucoidan**
**represents a carbon dioxide sink in the ocean, we have rewritten the introduction**
**to introduce the evidence to the reader that may have been overlooked in the**
**previous version. Moreover, we understood the request for an editorial revision**
**and so we have largely revised the entire manuscript.**

We are thankful for the time and effort Editors and Reviewers devoted to this manuscript.
We are grateful for their constructive comments, which we have carefully addressed in
our revisions. The suggested improvements have been incorporated into the manuscript,
and we believe these changes have significantly enhanced its overall quality.

-----

**Responses to Reviewer #1**

-----

Reviewer #1 (Remarks to the Author):

**Reviewer #1:** The authors present a fascinating insight into the role of phosphate
dependencies across both fucan synthesis by a marine macroalgae *Glossomastix* and
fucan degradation by an isolated *Akkermansiaceae* bacterium. *Glossomastix*-production
of fucan was characterized in detail as to the structure of the polysaccharide. In concert
efforts were made to recover bacteria that could degrade and consume fucan as a
substrate. Isolation efforts produced one bacterium that was characterized in detail,
whereby it was deduced that phosphate starvation restricted growth, which is in contrast
to *Glossomastix*-production of fucan which occurs under nutrient limiting conditions. My
background is in microbial structure and function, hence my comments below are
reflective of that and do not cover elements I am not an expert in (such as polysaccharide
structural characterisation. Overall, the paper is well written and presented, I have several
comments for consideration.

**Authors:** We thank the reviewer for this positive evaluation.

-----

**Reviewer #1:** The phenotypic data observed for *V_227* was very interesting, though its
culture-centric characterisation makes it difficult to judge the ecological impact of this
organism in a wider context. Such insight could be drawn from its isolation source and
enrichment culture that led to its recovery. Given the authors observations that bacterial
capabilities to digest fucan are rare (L209) and that this organism was seemingly quite
specific to *Glossomastix*-fucan and unable to degrade other macroalgae fucans (L203-
209), the question then arises if there is any co-occurring pattern between *V_227* and
*Glossomastix* at a larger ecological level?

**Authors:** We thank the reviewer for this comment. Ecological co-occurring pattern
between *V_227* and *Glossomastix* is still unknown and would be an interesting project. In
this study, we focus on assembling a model system for (fucoidan) carbon-cycling

containing a fucoidan-producer and –degrader to reveal how phosphate limitation affects
the fucoidan exudation and degradation. Hence, our initial sampling strategy aimed to get
fucoidan-degrading bacteria across a range of environments and was not specifically
designed to test for ecological co-occurrence patterns between V_227 and *Glossomastix*.
To make it more clear to the reader, we re-wrote the corresponding introduction part on
Line 56-74 as:

*“Previous research showed transporters for phosphate acquisition and enzymes for*
*laminarin degradation are highly expressed by bacteria during algal blooms¹⁰. Hence,*
*bacteria may require phosphate to degrade polysaccharides including laminarin and*
*fucoidan. It has been suggested that the accumulation of algae derived mucilage*
*polysaccharides occurs during algae blooms when phosphate becomes limiting¹¹. Under*
*these conditions the algae show reduced or absent growth (cell divisions) yet continue*
*carbon dioxide fixation into released polysaccharides. However, this accumulation also*
*requires that bacteria fail to degrade the polysaccharides that are released by algae. This*
*bacterial inability to degrade polysaccharides in response to phosphate limitation has to*
*the best of our knowledge not been tested. To investigate the role of phosphate for*
*fucoidan degradation at the molecular level we assembled a model system. As a fucoidan*
*producer we chose the microalgae *Glossomastix* sp. PLY432 isolated from the English*
*Channel¹². We purified and structurally characterized this previously unknown fucoidan*
*from *Glossomastix* sp. PLY432, the first fucoidan structure from a microalgae. We used*
*the purified fucoidan as sole carbon source to isolate a previously unknown marine*
*Verrucomicrobiaceae bacterium that was specialized on this fucoidan. Physiological*
*experiments showed that phosphate deprivation promotes fucoidan synthesis by algae*
*and inhibits fucoidan degradation by bacteria. Our results offer a new theory in the ocean*
*that phosphate deprivation of microbes could promote the fixation and sequestration of*
*carbon dioxide in the form of fucoidan.”*

-----

**Reviewer #1:** Were these macroalgae present at the sampling sites or why where these
sites chosen, this is not clearly presented in the paper.

**Authors:** We selected these sampling sites due to the frequent occurrence of algal
blooms and fucoidan accumulation in the surrounding waters of Hegoland, which
increases the possibility of isolating fucoidan-degrading bacteria. In addition, the sites are
in close geographic proximity to our laboratory, facilitating repeated sampling efforts. In
the revised manuscript, we make it clear to readers that why we chose these sites to
isolate fucoidan degrader by adding the following text on Line 190-193:

*“To isolate a fucoidan degrading bacterium we used seawater and intertidal pore water*
*from multiple sites along a beach on the island of Helgoland (Extended Data Fig. 4a, b,*
*c), where algal blooms and fucoidan accumulation were reported²”.*

-----
**Reviewer #1:** The interspecies dynamics at play in the original enrichment may well have
provided information as to why certain isolates “lost” their ability to degrade fucan or the
strict requirements of additional “additional nutrients, trace elements and vitamins” (L180).
Were any efforts made to characterize the enrichment cultures? Was V_227 the dominant
fucan degrader in this enrichment? Were efforts made to genomically characterise the
other fucan degraders that lost the phenotype (what taxonomy where they? Did they
encode fucan degrading capabilities). Additional insight into the complex enrichments
may well have assisted in understanding the seemingly lack of other bacteria callable of
fucan degradation?

**Authors:** We found that the degradation of *Glossomastix* fucan by community 2 took
about 14 days (Extended Data Fig. 5c), whereas V_227 isolated from community 2 was
able to mostly degrade *Glossomastix* fucan on day 3 (Fig. 5a), and by day 6 it was able
to completely degrade *Glossomastix* fucan (Extended Data Fig. 6). This indicates that
V_227 is in nutrient competition with other bacteria. We did Pac-Bio sequencing of DNA
extracted from community 2 (data not shown), and 16s rRNA matching showed that the
community contained a variety of bacteria from the Proteobacterium and Flavobacterium,
which are much more abundant in the ocean than V_227, and which we termed as the
dominant community in the ocean, and which apparently prey on nutrients to a greater

extent than V_227. Although genes encoding fucan degradation from other bacteria were
annotated in community 2, only V_227 contained the GH168 for endo-fucanase activity.

-----

**Reviewer #1:** L184 “enough nutrients” what does this mean?

**Authors:** We revised this sentence to ‘*When we added nutrients including phosphate but*
*also trace elements and vitamins, V_227 degraded 80% of the fucoidan (Fig. 4a)*’ on Line
196-197 to make it clear to readers.

-----

**Reviewer #1:** The screening of public data for organisms related to V_227 is a good idea,
but not surprising that limited matches occurred when using DNA-DNA based searches.
Could searches using AAI, CAZy families, CAZy/gene synteny help identify other distantly
related organisms that still fulfill the same functional niche? Were the CAZyme
determinants for fucan degradation identified from V_227 and could they be used to
identify other fucan degrading populations?

**Authors:** This is a good point, and we have added further clarifications: Our original
search was focused on finding closely related organisms in marine metagenomes (that’s
why we used a nucleotide search). However, we realized that, so far, no close relatives
have recovered. Then, we used the Genome Taxonomic Database to generate a
phylogenetic tree, which included all available genomes for the family *Akkermansiaceae*.
At the average amino acid level (AAI), it is also clear that V_227 is a novel species of the
SW10 genus. We have added an AAI matrix comparing all known SW10 genomes
(Extended Data Fig. 7a). To follow up on your suggestions, we have also extended the
search to compare all CAZy families, emphasizing those related to fucan degradation
(Supplementary Table 4). From this analysis, it is clear that the genetic potential for fucan
degradation is particular to V_227. Within the genus SW10, this isolate is the only known
species with GHs with fucose and endo fucanase GH141 and GH168 potential
degradation activity, respectively (Supplementary Table 4). The latter is present with two
copies and within a genetic context that includes other genes for fucoidan degradation
only detected in V_227. These results suggest its unique ability to degrade fucoidan within

members of SW10 genus. Of course, this analysis is limited to the known diversity so far
and the fact that a copy of the GH168 is also found in the g3_MAG_00010
(GCA_023299815.1), but given the genomic fragmentation of this MAG, its role in
fucoidan degradation is not conclusive. This is because the GH168 gene is located on a
small contig, and the neighboring genes could not be assigned a function. For GH141
genes, we detected similar genetic contexts for five species. Although these are not the
canonical PULs commonly found in *Bacteroidota*, we observe a clustering of genes likely
involved in the degradation of fucose-containing sulfated polysaccharides. This
comparison of genetic contexts is part of Extended Data Fig. 7. We have now added a
paragraph on Line 222-235 to account for these added results.

*“Deoxyribonucleic acid, DNA sequencing of V_227 gave two contigs, one chromosome*
*with 6,34 Mbp, and a megaplasmid with 0,42 Mbp. Phylogeny showed V_227 belongs to*
*the Akkermansiaceae family³⁴. This family also includes the human commensal*
*Akkermansia muciniphila that digests mucin, a fucoidan homolog, which is secreted by*
*human, mammalian epithelial cells³⁵. Within the family V_227 belongs to the SW10 clade*
*clades containing so far only sequences. V_227 shares 73.54% average nucleotide*
*identity (ANI) with a metagenome assembled genome (WKH.23) from a river in Southern*
*China³⁶ and is the first cultivated member strain within this genus (Fig. 5). Average amino*
*acid (AAI) values were also low when comparing V_227 to known members of the SW10*
*genus (Extended Data Fig. 8a), reinforcing its novelty at the species level. V_227 is the*
*only species carrying two copies of an endo-fucoidanase (GH168), thus likely*
*representing a unique niche among members of the SW10 genus. Also, GH141 (alpha-*
*fucosidase) was detected for other members of the SW10 genus with overall similar*
*genetic contexts, including CBMs and sulfatases in the same genetic contexts (Extended*
*Data Fig. 8b-c).”*

-----
**Reviewer #1:** L253. This statement needs to be slightly modified, or more info provided.
determined using which bacterium? this is phrased as though its been determined for
V_227?

**Authors:** The affinity constant of pstSABC system was reported for *E. coli*. Combined
with the comments from Review #3 and to make this paragraph clearer to the reader, we
have rewritten this paragraph on Line 273-286 as:

*“In other ecosystems, bacteria such as Bacteroides thetaiotaomicron⁴⁷ and Clostridium*
*perfringens⁴⁸ that degrade recalcitrant mucilage polysaccharides use high affinity*
*phosphate transporters of the ABC-type (Supplementary Table 8). This high affinity*
*required to bind phosphate indicates that phosphate is an important but limited resource*
*for which bacteria must compete. Hence, phosphate may be important for and influence*
*the degradation of recalcitrant polysaccharides. During algal blooms in the ocean, we*
*previously found that bacteria express the ABC transporter and other proteins used for*
*phosphate and phosphonate uptake¹⁰. They also express enzymes for the degradation*
*and metabolism of laminarin such as GH16 endo-beta-1,3-glucanases. However, the*
*fucoidan degrading bacteria showed downregulation or absence of fucoidanase*
*expression². This absence of degradation was further supported by the concentration of*
*fucoidan increasing over several month². Meanwhile, laminarin was effectively consumed*
*by bacteria despite the limited phosphate concentration^{2,23}. Conclusively, the availability*
*of phosphate might especially constrain the ability of bacteria specialized in the*
*degradation of complex glycans such as fucoidan.”*

-----
**Reviewer #1:** L255. From which organism and are these data from samples where V_227
is present? This is not clear how this is connected to V_227

**Authors:** These data are from an algae bloom in North Sea revealed by metagenomics,
which has been reported by Vidal-Melgosa et al (2021)
(<https://www.nature.com/articles/s41467-021-21009-6>). Combined with the comments
from Review #3 and to make this paragraph more logic and clearer to the reader, we have
rewritten this paragraph on Line 273-286 as:

*“In other ecosystems, bacteria such as Bacteroides thetaiotaomicron⁴⁷ and Clostridium*
*perfringens⁴⁸ that degrade recalcitrant mucilage polysaccharides use high affinity*

*phosphate transporters of the ABC-type (Supplementary Table 8). This high affinity*
*required to bind phosphate indicates that phosphate is an important but limited resource*
*for which bacteria must compete. Hence, phosphate may be important for and influence*
*the degradation of recalcitrant polysaccharides. During algal blooms in the ocean, we*
*previously found that bacteria express the ABC transporter and other proteins used for*
*phosphate and phosphonate uptake¹⁰. They also express enzymes for the degradation*
*and metabolism of laminarin such as GH16 endo-beta-1,3 glucanases. However, the*
*fucoidan degrading bacteria showed downregulation or absence of fucoidanase*
*expression². This absence of degradation was further supported by the concentration of*
*fucoidan increasing over several month². Meanwhile, laminarin was effectively consumed*
*by bacteria despite the limited phosphate concentration^{2,23}. Conclusively, the availability*
*of phosphate might especially constrain the ability of bacteria specialized in the*
*degradation of complex glycans such as fucoidan.”*

-----

**Responses to Reviewer #2**

-----

Reviewer #2 (Remarks to the Author):

**Reviewer #2:** This manuscript presents important findings about the interaction between
the microalgal production of sulfated fucan and the bacterial degradation of these glycans
under low-phosphorus (P) conditions. Fucoidans are a family of polysaccharides which
contain a large amount of fucose and sulfate ester groups. They are produced by diatoms
and brown algae and secreted during growth and after death as an extracellular matrix.
This fucan matrix is of note because, contrary to other polysaccharides, it is resistant to
bacterial degradation, allowing particulate fucan to persist in the ocean and be exported
to depth, providing a mechanism for carbon sequestration. In this paper, the authors
present evidence that low phosphate concentrations can slow the rate of bacterial fucan
degradation, without impacting algal fucan production, leading to a stabilization of fucan,
which in turn, may help algal cells compete with heterotrophic bacteria for nutrients, and
may provide a pathway for the long-term sequestration of fucan-carbon.

The manuscript is well written and clearly guides the reader through the arguments being
presented. The data are of high quality. I was particularly impressed with the
characterization of the fucan structure. The methodologies employed and the analyses
performed were of exceptional, and they are well described in the manuscript. This paper
also presents novel evidence of nutrient limitation on the uptake of a specific carbon
substrate by marine heterotrophs. Previous works demonstrate that nutrient limitation can
affect carbon metabolism by enhancing the uptake of specific substrates¹, or by slowing
carbon metabolism as a whole.² But this study is, to my knowledge, the first
demonstration of nutrient limitation leading to the suppression of uptake of a specific
carbon substrate. As such, I support the publication of this work in some form, although
there are some concerns that I would like to see addressed before publication in this
journal.

**Authors:** We thank the reviewer for this very positive and encouraging evaluation.

-----

General Concerns:

**Reviewer #2:** One concern I have with this work is the difficulty in untangling the
relationship between P concentration and cell response. As it is presented, each
experimental treatment is denoted based on its initial concentration of P, but no data are
presented showing the changes in dissolved P over time. Thus, it is unclear how the
differential responses observed between treatments directly relate to P concentration.
The initial P concentration in the “phosphate limited” treatment is 2-5 times higher than
what is typically observed in the English Channel (200-400 nM), where PLY432 was
isolated from.³ As the concentration of P throughout the growth experiments is not
presented, and the growth rates of PLY432 showed no significant difference between P
treatments, it is unclear at what point PLY432 begins experiencing P limitation. To fully
support the conclusion that PLY432 fucan production is unaffected by P limitation, I
believe these data, or additional evidence of P stress by PLY432 such as biomarkers for
P stress, are needed.

**Authors:** We understand the reviewer's concern. To convince the reader that secretions
of algal polysaccharides are enhanced by phosphate deprivation, we not only added
related citations (e.g. <https://royalsocietypublishing.org/doi/10.1098/rstb.2016.0406>;
<https://www.jstor.org/stable/44635010>) on Line 113, but also added an experiment of
semi-continuous culture for the algae. The results can be found in Figure 2f and Extended
Figure 2. The corresponding text can be found between Lines 120 and 132. We propose
to write the paragraph about this experiment in this way:

*“It has been previously shown that phosphate deprivation can enhance the amount of*
*polysaccharide release by microalgae in laboratory¹⁸⁻²² and environmental settings^{2,23}.*
*For example in the Adriatic Sea the accumulation of mucilage polysaccharides increases*
*when phosphate becomes growth limiting¹¹. Hence, we tested the influence of phosphate*
*concentration on fucoidan release by *Glossomastix*. We conducted semi continuous*
*growth experiments where we increasingly decreased the phosphate concentration while*
*keeping other essential nutrients at a constant concentration. The pre culture was grow*

*in medium with an inorganic phosphate source and then inoculated in the same medium*
 *but without phosphate. When the algae entered the exponential growth phase half of the*
 *culture was inoculated into fresh medium again without phosphate. The remainder was*
 *kept as batch experiment. During the semi-continuous experiment, which included four*
 *transfers, we recorded cell numbers, phosphate concentrations and the fucoidan*
 *exudation rate. In the semi-continuous growth and batch mode, after the cultivation cycle*
 *ended, the fucoidan yield per cell was higher in cultures with lower phosphate*
 *concentrations (Fig. 2b and Extended Data Fig. 2). The semi-continuous growth*
 *experiment showed that when the culture stopped being diluted on day 15, the fucoidan*
 *concentration began to increase (Extended Data Fig. 2b)."*

**Fig. 2f. Fucose yield of *Glossomastix* in a semi-continuous growth mode.** In the 1st
 batch culture, the starting phosphate concentration was 14 μM . In the 2nd, 3rd, 4th batch
 cultures and the semi-continuous growth, the actual phosphate concentrations were lower
 than the theoretical ones (7 μM , 3.5 μM , 1.75 μM and 0.875 μM respectively).

-----

**Reviewer #2:** In the discussion on quantitative glycan carbon accounting (beginning at
 line 301), it is unclear if the data are normalized by the number of cells in each treatment.
 The per cell uptake of the polysaccharides would be a useful visualization that would
 remove the effects of growth from the effects of reduced C uptake. Additionally, the data
 presented in this section appear to be at odds with the data shown in Fig. 6C. The body
 text states that, at 1 μM P, more Laminarin-C is converted than Fucan-C. Fig. 6C and the
 data in Supplementary Table 14 show that more Fucan-C is converted.

**Authors:** We think it makes more sense to test it on a macro level as in this way effects
such as cell volume can be ignored. The data presented in Fig. 6c and Supplementary
Table 14 are consistent. In Fig. 6c we highlight the slope, the ratio of glycan-carbon
conversions per molecule of phosphate. To make it clear, we have modified this
description in the text (Line 335-336) as:

*“For laminarin carbon, the conversion ratio is C:P = 195.46:1 (r = 1, P < 0.0001) (Fig. 6c).*
*In case of the fucoïdan, the conversion ratio is C:P = 84.43:1 (r = 1, P < 0.0001)”.*

-----
**Reviewer #2:** Finally, at times, the readability of this manuscript is lacking, resulting from
confusing sentence structure and turns of phrase. The work would benefit from editing by
someone with full professional proficiency in English.

**Authors:** We have revised the language thoroughly.

-----
Specific Comments:

**Reviewer #2:** Ln 23-25 Later in the text, the authors state that phosphate concentration
didn't affect the maximum synthesis of fucan. Thus, the statement that phosphate
limitation enhanced fucan synthesis is not clear supported.

**Authors:** We added a semi-continuous culture for the algae, and the result showed that
PLY432 fucan production per cell is not limited but enhanced by phosphate limitation. To
express more rigorously and accurately, we have rewritten the abstract on Line 29-30 as:

*“The fixation of carbon dioxide into fucoïdan by the microalgae, Glossomastix sp. PLY432,*
*occurred independent of the phosphate concentration.”*

-----
**Reviewer #2:** Ln 53 The work should state that the strain of Glossomastix was isolated
from the English Channel instead of the Atlantic Ocean. This provides more context for
the work.

Authors: We have revised it on Line 67.

Reviewer #2: Ln 86: The terminology in this sentence should be clarified. Does the phrase “not growing” mean that the culture has reached stationary or senescence/death phase?

Authors: It was meant in the stationary phase. Considering suggestions from all reviewers, we rewrite this paragraph on Line 96-110 as:

“Phosphate deprivation promotes fucoidan exudation

*During two months of growth, the total glycan carbon content present in the secreted carbohydrates increased 36.42-fold, from 4.14 mg/L to 150.72 mg/L. The concentration of fucoidan, composed of fucose, rhamnose and galacturonate, increased during the stationary phase (Fig. 2f). Galactose, glucose and glucuronate did not follow this trend (Fig. 2i-j and Supplementary Fig. 1). The carbon accumulation rate was 0.52 pg/cell/day, for fucose 14 μ M/day ($r = 1.00$, $P < 0.001$), for rhamnose 4 μ M ($r = 1$, $P < 0.0001$) and for galacturonate 1 μ M ($r = 1$, $P < 0.0001$) (Fig. 2j), or ~ 0.68 pg of fucose, ~ 0.19 pg of rhamnose, and ~ 0.06 pg of galacturonate. Based on the volume and carbon content of diverse microalgae¹⁵⁻¹⁷, we estimate a *Glossomastix* cell contains 24.49 pg carbon (Supplementary Table 2). Using this figure, we calculated one cell releases 2.12% of its fixed carbon dioxide in form of fucoidan. This figure is consistent with the fucoidan excretion rates of the brown macroalgae *Fucus vesiculosus*⁴.”*

Reviewer #2: Ln 90: The list of sugars which do and don’t increase in concentration are contradictory; galactose is listed twice. I believe the first list of sugars is supposed to include galacturonate.

Authors: We thank the reviewer for pointing out this error. We have made the correction.

**Reviewer #2:** Ln 105: This conclusion is difficult to see from the data presented. An
analysis of the growth curves to determine when the acceleration of growth becomes zero
would provide a clearer metric of the change in growth patterns with changing P
concentrations. See Navarro-Pérez et al.⁴ as an example.

**Authors:** We added a semi-continuous experiment of algae culture with phosphate as
the limiting nutrient to show the phosphate limitation enhanced the fucoidan production
408 per cell. We rewrite this paragraph on Line 112-132 as:

*“It has been previously shown that phosphate deprivation can enhance the amount of*
*polysaccharide release by microalgae in laboratory¹⁸⁻²² and environmental settings^{2,23}.*
*For example in the Adriatic Sea the accumulation of mucilage polysaccharides increases*
*when phosphate becomes growth limiting¹¹. Hence, we tested the influence of phosphate*
*concentration on glycans/fucoidan exudation by *Glossomastix*. We restricted access to*
*organic phosphate and found the algae entered the stationary phase faster (**Fig. 2a and***
***Extended Data Fig. 2a-c**). Neither the maximum, synthesis rate of glycans per cell and*
*day ($r = -0.42$, $P > 0.5$) nor the growth rate ($r = -0.89$, $P > 0.5$) were significantly affected*
*by the decrease of organic phosphate concentration (**Fig. 2e and Extended Data Fig.***
***2d**). Moreover, we conducted semi-continuous growth experiments where we*
*increasingly decreased the inorganic phosphate concentration while keeping other*
*essential nutrients at a constant concentration. The pre-culture was grown in medium with*
*inorganic phosphate and then inoculated in the same medium but without phosphate.*
*When the algae entered the exponential growth phase, half of the culture was inoculated*
*into fresh medium again without phosphate. The remainder was kept as batch experiment.*
*During the semi-continuous experiment, which included four transfers, we recorded cell*
*numbers, phosphate concentrations and the fucoidan exudation rate. In the semi-*
*continuous growth and batch mode, after the cultivation cycle ended, the fucoidan yield*
*per cell was higher in cultures with lower phosphate concentrations (**Fig. 2f and***
***Extended Data Fig. 3**). The semi-continuous growth experiment showed that when the*
*culture stopped being diluted on day 15, the fucoidan concentration began to increase*
*(**Extended Data Fig. 3**).*”

-----
**Reviewer #2:** Ln 288: Without measuring dissolved phosphate at the end of the
experiment, I'm not sure how you can justify this assumption.

**Authors:** We added data of phosphate concentrations in the culture supernatant of
bacterial growth experiment to Extended Data Fig. 9–10. It shows that most phosphate
was consumed up by the end of the culture.

-----
**Reviewer #2:** Ln 303 and 461: Was a recovery analysis of the HPAEC-PAD method
performed? Hydrolysis of polysaccharides to yield monosaccharides can result in poor
recoveries, especially with partially degraded polysaccharides.

**Authors:** Acid hydrolysis could result in poor recoveries. We did not do recovery analysis
since it had been reported by Engel & Händel (2011)
(<https://www.sciencedirect.com/science/article/pii/S0304420311001010>) that highest
yields of different sugars were achieved at different HCl concentrations while most of the
monosaccharide achieved the highest yields at a HCl concentration of 0.8-1 M. We cited
this paper on Line 561.

-----
**Reviewer #2:** Ln 365: "of" is repeated in this line.

**Authors:** The repetition has been removed.

-----
**Reviewer #2:** Ln 630: Authors should provide the polysaccharide-C concentration given
in each treatment as those data would make interpretation of the data, particularly figure
6C more facile.

**Authors:** We calculated the initial glycan-carbon concentration for each treatment group
using the HPAEC-PAD data and integrated it into the method (Line 740-742).

-----
**Reviewer #2:** Ln 720: Consider depositing your NMR data in a repository such as NMRxiv.

Authors: Yes, we will also upload the NMR data to Pangaea.

Reviewer #2: Below is the citation format, which will be revised at the time of submission

Authors: Sorry for the format mistakes. We have revised all mentioned here and double checked the format of all citations.

Reviewer #2: Ln 983: “classis nova” should be italicized.

Authors: Done.

Reviewer #2: Ln 991: “Biogeochemistry of marine dissolved organic matter” should be italicized, not in quotations.

Authors: Done.

Reviewer #2: Ln 1002: The number “1” should be superscripted in “¹H-NMR”

Authors: Done.

Reviewer #2: Ln 1016: “Closterium” and “Nutrient Limitation” should not be capitalized.

Authors: Done.

Reviewer #2: Ln 1019: “Closterium” should not be capitalized.

Authors: Done.

Reviewer #2: Ln 1057: “can” should be capitalized.

**Authors:** Done.

-----

**Reviewer #2:** Ln 1066: “cholerae” should be italicized.

**Authors:** Done.

-----

**Reviewer #2:** Ln 1141: The isotope mass numbers in this title should be superscripted

**Authors:** Done.

-----

**Reviewer #2:** References cited in this review:

1. Liang, Z., Letscher, R. T. & Knapp, A. N. Dissolved organic phosphorus concentrations
in the surface ocean controlled by both phosphate and iron stress. *Nat. Geosci.* 15, 651–
657 (2022).

2. Fourquez, M. et al. Effects of iron limitation on growth and carbon metabolism in
oceanic and coastal heterotrophic bacteria. *Limnology and Oceanography* 59, 349–360
(2014).

3. Jordan, M. B. & Joint, I. Seasonal variation in nitrate:phosphate ratios in the english
channel 1923–1987. *Estuarine, Coastal and Shelf Science* 46, 157–164 (1998).

4. Navarro-Pérez, M. L., Fernández-Calderón, M. C. & Vadillo-Rodríguez, V.
Decomposition of growth curves into growth rate and acceleration: A novel procedure to
monitor bacterial growth and the time-dependent effect of antimicrobials. *Applied and*
*Environmental Microbiology* 88, e01849-21 (2022)

**Authors:** We have cited related papers in the corresponding text. We cited Liang et al.
and Jordan & Joint on Line 246, Fourquez et al on Line 412, respectively.

-----

**Responses to Reviewer #3**

-----

Reviewer #3 (Remarks to the Author):

Key results

**Reviewer #3:** The study examined fucan synthesis by the marine microalgae
*Glossomastix* (non-axenic) as well as fucan digestion by a bacterium in the
*Akkermansiaceae* family, aiming to understand how limitation by organic phosphate
affected both processes. *Glossomastix* kept in long term culture (~10 years) was found
to exude fucan extracellularly. In the stationary phase, under P limitation, fucan synthesis
increased. The structure of the algae-generated fucan was characterized. Bacteria were
isolated from the *Glossomastix* co-culture, seawater and porewater. With the exception
of one bacterium, V_227, the isolated bacteria did not digest fucan. V_227 imported fucan
into its cells and digested low concentrations of fucan if given additional trace elements
and vitamins. V-227 was sequenced and used to look for its presence in the ocean (it was
rare). Genome annotation showed genes consistent with glycan digestion and high affinity
P uptake. Following limitation experiments (down to 1 uM), glycan carbon accounting
found fucan more stable with limited phosphate for this system. The main conclusions I
drew from reading the manuscript were: organic phosphate limitation enhances fucan
production by *Glossomastix* and fucan digestion is rare and/or very specific to particular
organisms. Additional (in my assessment, unwarranted) conclusions were made by the
authors, including that fucan plays a key role in algae-bacteria interactions, allowing for
algal persistence and CO₂ concentrations in the ocean.

**Authors:** We appreciate the critical, rigorous and extensive evaluation of this manuscript.
We have thoroughly considered this reviewer's suggestions and propose.

-----

Validity

The conclusions of this paper were a stretch based on the data collected. I would suggest
re-interpretation and reconsideration of conclusions. **Three major concerns are as**

**follows:**

**Reviewer #3:** This work relies solely on batch culture experiments to talk about the effects
of phosphate limitation on fucan synthesis and/or digestion. There is a major flaw with
this approach in that carrying capacity can be reached due to factors other than
phosphate limitation, especially at the time scales of this experiment. What other
resources were measured?

**Authors:** We agree that many factors can contribute to the synthesis and degradation of
fucoidan by algae and bacteria. Hence there are many other variables one could test. We
chose phosphate as Liebig's limiting factor because it has been previously shown that
phosphate deprivation can have an inverse proportional effect on the polysaccharide
exudation rate by algae. However, it has to the best of our knowledge not been tested
before for bacterial degradation of complicated polysaccharides such as fucoidan. So, the
hypothesis was that algae can synthesize it based on these papers but we did not know
how bacteria would be affected. To clarify this previous lack of knowledge better to the
reader we added the following sentences on Line 112-116:

*"It has been previously shown that phosphate deprivation can enhance the amount of*
*polysaccharide release by microalgae in laboratory¹⁸⁻²² and environmental settings^{2,23}.*
*For example in the Adriatic Sea the accumulation of mucilage polysaccharides increases*
*when phosphate becomes growth limiting¹¹. Hence, we tested the influence of phosphate*
*concentration on glycans/fucoidan exudation by *Glossomastix*".*

We also appreciate the experiment suggested by the reviewer to convince readers of the
reported results/conclusions. We propose to make the hypothesis of phosphate limitation
restricting fucoidan degradation by bacteria more clearly by rewriting the paragraph on
Line 273-286 as:

*"In other ecosystems, bacteria such as *Bacteroides thetaiotaomicron*⁴⁷ and *Clostridium**
*perfringens⁴⁸ that degrade recalcitrant mucilage polysaccharides use high affinity*
*phosphate transporters of the ABC-type (Supplementary Table 8). This high affinity*

*required to bind phosphate indicates that phosphate is an important but limited resource*
*for which bacteria must compete. Hence, phosphate may be important for and influence*
*the degradation of recalcitrant polysaccharides. During algal blooms in the ocean, we*
*previously found that bacteria express the ABC transporter and other proteins used for*
*phosphate and phosphonate uptake¹⁰. They also express enzymes for the degradation*
*and metabolism of laminarin such as GH16 endo-beta-1,3 glucanases. However, the*
*fucoidan degrading bacteria showed downregulation or absence of fucoidanase*
*expression². This absence of degradation was further supported by the concentration of*
*fucoidan increasing over several month². Meanwhile, laminarin was effectively consumed*
*by bacteria despite the limited phosphate concentration^{2,23}. Conclusively, the availability*
*of phosphate might especially constrain the ability of bacteria specialized in the*
*degradation of complex glycans such as fucoidan.”*

In response to this reviewer, we now also conducted experiments with inorganic
phosphate. These results are described in the following sections of the paper on Line
120-132:

*“Moreover, we conducted semi-continuous growth experiments where we increasingly*
*decreased the inorganic phosphate concentration while keeping other essential nutrients*
*at a constant concentration. The pre-culture was grown in medium with inorganic*
*phosphate and then inoculated in the same medium but without phosphate. When the*
*algae entered the exponential growth phase, half of the culture was inoculated into fresh*
*medium again without phosphate. The remainder was kept as batch experiment. During*
*the semi-continuous experiment, which included four transfers, we recorded cell numbers,*
*phosphate concentrations and the fucoidan exudation rate. In the semi-continuous growth*
*and batch mode, after the cultivation cycle ended, the fucoidan yield per cell was higher*
*in cultures with lower phosphate concentrations (Fig. 2f and Extended Data Fig. 3). The*
*semi-continuous growth experiment showed that when the culture stopped being diluted*
*on day 15, the fucoidan concentration began to increase (Extended Data Fig. 3)”.*

This study was focused on macronutrients and we have tested fucoidan as the carbon
source. We would have liked to test the other crucial macronutrient, nitrogen.
Unfortunately, the fucoidan contains some tightly associated, unknown nitrogen
compounds. It is known that fucoidan binds strongly to proteins. We have not been able
to separate these nitrogen containing molecules despite extensive and many purification
steps developed and applied for the first time in this paper. Because the fucoidan
contained a nitrogen source, we could not test nitrogen as an independent variable, which
would be required to measure the effects of nitrogen in form of a nutrient. Hence, we
could not disentangle the effect of nitrogen limitation as we did with phosphate.

However, to make this concern of the reviewer clear to readers of the paper, we propose
amend a limitation section that clarifies that many other variables, nutrients can contribute
to the synthesis of fucoidan by algae and its degradation by bacteria and should be
considered in future experiments. We propose to write it like this between lines 410 and
413 of the revised manuscript:

*“Besides, not only phosphate, other variables, such as nitrogen and trace metals could*
*also affect fucoidan exudation and degradation, and these should be considered in future*
*research”.*

-----

**Reviewer #3:** How do we not know other factors influenced fucan dynamics?

**Authors:** To clarify that other potential factors can influence fucoidan dynamics, we now
propose to write the limitation that other nutrient factors could also influence the fucoidan
dynamics and they should be considered in future experiments. The revised limitation can
be found on Line 410-413 as:

*“Besides, not only phosphate, other variables, such as algal surrounding microbiome,*
*nitrogen and trace metals could also affect fucoidan exudation and degradation, and*
*these should be considered in future research”.*

-----

**Reviewer #3:** Moreover, cells are not in a stable state physiologically. To be convinced of
the conclusions, I would need to see data from a chemostat experiment (or even semi-
continuous growth experiment) that shows the same results.

**Authors:** As requested by Reviewer 3, We conducted the semi-continuous growth
experiment. The results can be found in Figure 2f and Extended Figure 3. The
corresponding changed text can be found between Lines 96 and 131. We propose to
write the paragraph about this experiment on Line 120-132 in this way:

*“Moreover, we conducted semi-continuous growth experiments where we increasingly*
*decreased the inorganic phosphate concentration while keeping other essential nutrients*
*at a constant concentration. The pre-culture was grown in medium with inorganic*
*phosphate and then inoculated in the same medium but without phosphate. When the*
*algae entered the exponential growth phase, half of the culture was inoculated into fresh*
*medium again without phosphate. The remainder was kept as batch experiment. During*
*the semi-continuous experiment, which included four transfers, we recorded cell numbers,*
*phosphate concentrations and the fucoidan exudation rate. In the semi-continuous growth*
*and batch mode, after the cultivation cycle ended, the fucoidan yield per cell was higher*
*in cultures with lower phosphate concentrations (Fig. 2f and Extended Data Fig. 3). The*
*semi-continuous growth experiment showed that when the culture stopped being diluted*
*on day 15, the fucoidan concentration began to increase (Extended Data Fig. 3).”*

**Figure 2f. Fucose yield of *Glossomastix* in a semi-continuous growth mode.** In the
1st batch culture, the starting phosphate concentration was 14 μ M. In the 2nd, 3rd, 4th

batch cultures and the semi-continuous growth, the actual phosphate concentrations
were lower than the theoretical ones (7 μM , 3.5 μM , 1.75 μM and 0.875 μM respectively).

-----

**Reviewer #3:** This study focuses on only one form of phosphate, B-glycerol-phosphate,
but conclusions talk about phosphate limitation generally. No experiments were done on
inorganic phosphate. How do conclusions about the ocean change if only organic
phosphate limitation affects fucan synthesis/digestion? Either experiments should be
added or the conclusions tamped down.

**Authors:** To address these concerns, we have conducted a series of additional
experiments with inorganic phosphate. In brief, we did batch culture and semi-continuous
culture of algae in inorganic phosphate. The batch culture of algae in inorganic phosphate
showed the similar result that algae started to exudate large amount of fucoidan in the
stationary phase where phosphate was consumed up (see the figure below). This new
figure can be found in Figure 2d of the new version manuscript. The corresponding text
is on the Line 108-110 as:

*“Glossomastix also continued exuding substantial quantities of fucoidan during stationary*
*phase when they were growing in the medium with inorganic phosphate.”*

Figure 2d. Monitoring of growth by cell counts, phosphate concentration and fucose
content in *Glossomastix* cultures. 7 μM KH_2PO_4 was used as the phosphate source.
Experiments were performed as independent triplicates ($n = 3$) and error bars represent
standard deviation of the mean.

The semi-continuous culture of algae in Figure 2f show that lower phosphate
concentration, higher fucose yield per algal cell. The results and description of these new
experiments can be found in the revised manuscript between Lines 120 and 132 as:

*“Moreover, we conducted semi continuous growth experiments where we increasingly*
*decreased the inorganic phosphate concentration while keeping other essential nutrients*
*at a constant concentration. The pre-culture was grown in medium with inorganic*
*phosphate and then inoculated in the same medium but without phosphate. When the*
*algae entered the exponential growth phase, half of the culture was inoculated into fresh*
*medium again without phosphate. The remainder was kept as batch experiment. During*
*the semi-continuous experiment, which included four transfers, we recorded cell numbers,*
*phosphate concentrations and the fucoidan exudation rate. In the semi-continuous growth*
*and batch mode, after the cultivation cycle ended, the fucoidan yield per cell was higher*
*in cultures with lower phosphate concentrations (Fig. 2f and Extended Data Fig. 3). The*
*semi-continuous growth experiment showed that when the culture stopped being diluted*
*on day 15, the fucoidan concentration began to increase (Extended Data Fig. 3).”*

**Figure 2f. Fucose yield of *Glossomastix* in a semi-continuous growth mode.** In the
1st batch culture, the starting phosphate concentration was 14 μM . In the 2nd, 3rd, 4th
batch cultures and the semi-continuous growth, the actual phosphate concentrations
were lower than the theoretical ones (7 μM , 3.5 μM , 1.75 μM and 0.875 μM respectively).

-----

**Reviewer #3:** The *Glossomastix* used was not axenic. How do we know that the
production of fucan was driven by the algae and not algae-bacterial interactions within

the culture? I didn't see taxonomy data of all bacteria (not just culturable bacteria)
associated with the algae. How do these taxa affect the performance of the algae?

**Authors:** It is possible that the microbiome affects the glycan synthesis exudation by
algae as previously shown for diatoms (Bartolek et al., 2022;
<https://academic.oup.com/ismej/article/16/12/2741/7474086>). Unfortunately, the extreme
sliminess, fucoidan content of the *Glossomastix* culture prohibits making these axenic,
and the slime potentially protects bacteria against antibiotics. The slime also prohibits
DNA extraction and community profiling. Hence, we cannot be certain, easily test the
influence of natural or mock communities associated by *Glossomastix* due to these
mentioned issues.

To address this concern, we propose to add this point as a new limitation point where we
clarify that a limitation of this study is the lack of tested microbiome influence on the slime
production. Corresponding text can be found on Line 410-412 as: "*Besides, not only*
*phosphate, other variables, such as algal surrounding microbiome, nitrogen and trace*
*metals could also affect fucoidan exudation or degradation, and these should be*
*considered in future research*".

We also now cite in the revised paper a previous study showing bacteria can influence
glycan synthesis in microalgae (diatoms) on Line 411. However, we would also like to
point out that it has been previously shown that phosphate and microbiome can influence
glycan synthesis in microalgae (now cited on Line 113) and that the major point of this
manuscript is the finding of the bacteria being negatively effects by phosphate
concentration.

-----
**Reviewer #3:** At times, the framing of the paper calls into question credibility of the
authors. For example, line 344: "So how do algae persist?" Because they use sunlight
and CO2. Complementary niches. I am not sure what to do with this sentence. Is it a joke?

**Authors:** We have clarified the question by writing on Line 380-387:

*“So how do algae persist and compete for nutrients given that heterotrophic bacteria such*
*as SAR11 are more abundant resulting in a large surface area covered by nanomole*
*affinity transporters for the uptake of inorganic nutrients⁵⁶? Earlier studies of quantitative*
*glycan accounting found algae derived, fucose-containing glycans of unknown structure*
*are stable for years in the global surface ocean. These glycans hold 20%, ~ 20 μM of the*
*dissolved organic carbon in this for the largest part phosphate deprived (0-300 nM)*
*system³⁷⁻³⁹. This persistence is in line with nutrients such as phosphate being required*
*for the degradation of such complex glycans^{47,48,60}.”*

-----

Other conclusions were drawn that seemed far-reaching and need more support. These
are outlined below.

**Reviewer #3:** Lines 32-33. “We conclude phosphate starvation constrains the ability of
bacteria to digest fucan, which evolves to maintain stability around algal cells and
consequentially also to keep carbon dioxide in the ocean.” I don’t know where the
evidence is for this conclusion. The study did not look at evolved traits, algae-bacteria
interactions in the phycosphere, or resources in the phycosphere. In other places, a
conclusion is drawn that algae are given more time/space to resources. But I do not see
data that support this. Are there resource data? Time resolved data?

**Authors:** To make it clearer and more rigorous, we rewrote the abstract on Line 22-35
as:

*“Brown algae and diatoms convert a substantial fraction of fixed carbon dioxide into the*
*sulfated polysaccharide fucoïdan, which sequesters carbon in the ocean. The stability of*
*fucoïdan remains intriguing considering marine bacteria with fucoïdanase genes are*
*globally present around algae. Bacteria that degrade complex polysaccharides share high*
*affinity transporters for phosphate uptake. Phosphate could be an important nutrient for*
*bacteria that degrade fucoïdan and reduce carbon sequestration. To test this hypothesis,*
*we assembled a microbial carbon cycle model consisting of a microalgae that produces*

*and a bacterium that degrades fucoïdan. The fixation of carbon dioxide into fucoïdan by*
*the microalgae, Glossomastix sp. PLY432, occurred independent of the phosphate*
*concentration. In contrast, the fucoïdan degrading Verrucomicrobium was inhibited by a*
*lack of phosphate. Degradation of the structurally simpler polysaccharide laminarin was*
*unaffected by the phosphate concentration. Phosphate deprivation enabled the fixation*
*of carbon dioxide in fucoïdan and disabled its degradation. Hence, phosphate deprivation*
*could be a potential strategy used to promote the fixation and sequestration of carbon*
*dioxide in form of fucoïdan.”*

-----

**Reviewer #3:** Line 103 and afterwards. “Algae continued fucan exudation despite
phosphate limitation.... Phosphate concentration neither affected the maximum,
synthesis rate per cell and day ($r = -0.42$, $P > 108$ 0.5). the maximum specific growth rate
($r = -0.89$, $P > 0.05$) (Fig. 2d and Extended Data Fig. 2d).” But looking at the
supplementary figures, changing the phosphate concentration changed the carrying
capacity. It also looks like the maximum specific growth rate changes. I question the
conclusion here based on these data as well as experimental design.

**Authors:** Phosphate concentration does affect the carrying capacity. We added the semi-
continuous culture of algae. The pre culture was grown in medium with an inorganic
phosphate source and then inoculated in the same medium but without phosphate. At the
beginning of the resulting exponential phase half of the culture was inoculated into fresh
medium again without phosphate to maintain the cells in the exponential phase. The
remainder was kept as batch experiment. We conducted four transfers and recorded cell
numbers, phosphate concentrations, fucoïdan exudation rate over the course of the semi
continuous experiment. Generally, the lower phosphate concentrations, the lower algal
abundance. However, the fucose yield per cell was higher in the lower-phosphate culture
(Figure 2f and Extended Figure 3). We have rewritten this part on Line 96-132 as:

**“Phosphate deprivation promotes fucoïdan exudation**

*During two months of growth in the medium with organic phosphate, the total glycan*
*carbon content present in the secreted carbohydrates increased 36.42-fold, from 4.14*

814 mg/L to 150.72 mg/L (Fig. 2a). The concentration of fucoïdan, composed of fucose,
rhamnose and galacturonate, increased during the stationary phase (Fig. 2b). Galactose,
glucose and glucuronate did not follow this trend (Supplementary Fig. 1). The carbon
accumulation rate was 0.52 pg/cell/day, for fucose 14 µM/day ($r = 1.00$, $P < 0.001$), for
rhamnose 4 µM ($r = 1$, $P < 0.0001$) and for galacturonate 1 µM ($r = 1$, $P < 0.0001$) (Fig.
2c), or ~ 0.68 pg of fucose, ~ 0.19 pg of rhamnose, and ~ 0.06 pg of galacturonate. Based
on the volume and carbon content of diverse microalgae¹⁵⁻¹⁷, we estimate a *Glossomastix*
cell contains 24.49 pg carbon (Supplementary Table 2). Using this figure, we calculated
one cell releases 2.12% of its fixed carbon dioxide in form of fucoïdan. This figure is
consistent with the fucoïdan excretion rates of the brown macroalgae *Fucus vesiculosus*⁴.
*Glossomastix* also continued exuding substantial quantities of fucoïdan during stationary
phase when they were growing in the same medium with inorganic phosphate (Fig. 2d).

It has been previously shown that phosphate deprivation can enhance the amount of
polysaccharide release by microalgae in laboratory¹⁸⁻²² and environmental settings^{2,23}.
For example in the Adriatic Sea the accumulation of mucilage polysaccharides increases
when phosphate becomes growth limiting¹¹. Hence, we tested the influence of phosphate
concentration on glycans/fucoïdan exudation by *Glossomastix*. We restricted access to
organic phosphate and found the algae entered the stationary phase faster (Fig. 2a and
Extended Data Fig. 2a-c). Neither the maximum, synthesis rate of glycans per cell and
834 day ($r = -0.42$, $P > 0.5$) nor the growth rate ($r = -0.89$, $P > 0.5$) were significantly affected
by the decrease of organic phosphate concentration (Fig. 2e and Extended Data Fig. 2d).
Moreover, we conducted semi-continuous growth experiments where we increasingly
decreased the inorganic phosphate concentration while keeping other essential nutrients
at a constant concentration. The pre-culture was grown in medium with inorganic
phosphate and then inoculated in the same medium but without phosphate. When the
algae entered the exponential growth phase, half of the culture was inoculated into fresh
medium again without phosphate. The remainder was kept as batch experiment. During
the semi-continuous experiment, which included four transfers, we recorded cell numbers,
phosphate concentrations and the fucoïdan exudation rate. In the semi-continuous growth
and batch mode, after the cultivation cycle ended, the fucoïdan yield per cell was higher

*in cultures with lower phosphate concentrations (Fig. 2f and Extended Data Fig. 3). The*
 *semi-continuous growth experiment showed that when the culture stopped being diluted*
 *on day 15, the fucoidan concentration began to increase (Extended Data Fig. 3).”*

**Extended Data Fig. 3: Cell growth and fucoidan secretion of *Glossomastix* in the**
**semi-continuous culture. a,** Cell density of *Glossomastix* in a semi-continuous growth
and batch mode. **b,** Fucose yield of in the semi-continuous growth and batch mode. In
the 1st batch culture, the starting inorganic phosphate concentration was 14 μM . In the
2nd, 3rd, 4th batch cultures and the semi-continuous growth, the actual phosphate
concentrations were lower than the theoretical ones (7 μM , 3.5 μM , 1.75 μM and 0.875
μM respectively). The experiment was performed in independent triplicate ($n = 3$), error
bars are the standard deviation of the mean.

-----
**Reviewer #3:** Also, how do the concentrations in the experiment to concentrations in the
ocean?

**Authors:** To inform the readers about phosphate concentration in the ocean in relation to
what we used in the experiments, we propose to reference on Line 245-247 of the paper
and values of phosphate concentration in oligotrophic regions of the surface ocean. This
new sentence is:

*“Especially in oligotrophic regions of the surface ocean with a phosphate concentration*
*of 0-300 nM³⁷⁻³⁹, but also during later phases of coastal algal blooms, phosphate rather*
*than the dissolved organic carbon content restricts the growth of bacteria⁴⁰.”*

-----
**Reviewer #3:** It seems that the concentrations are high. Is it possible that phosphate was
simply not limiting enough to see effects on fucan synthesis?

**Authors:** It is possible. Thus, we did semi-continuous culture of algae to confirm that
phosphate limitation has an effect on fucan synthesis. In the semi-continuous growth and
batch mode, after the cultivation cycle ends, the yield of fucose per cell in cultures with
lower phosphate content were higher than those in cultures with higher phosphate
concentration (Figure 2f).

 **Figure 2f. Fucose yield of *Glossomastix* in a semi-continuous growth mode.** In the
 1st batch culture, the starting phosphate concentration was 14 μM . In the 2nd, 3rd, 4th
 batch cultures and the semi-continuous growth, the actual phosphate concentrations
 were lower than the theoretical ones (7 μM , 3.5 μM , 1.75 μM and 0.875 μM respectively).

 The corresponding text can be found on Line 120-132.

-----
 **Reviewer #3:** I would also like to see these concentrations in context of other papers. For
 example, XX et al. considered P-limited 200 nM and P replete 45 μM .

 **Authors:** We added the semi continuous experiment. The pre culture was grow in
 medium with an inorganic phosphate source and then inoculated in medium without
 phosphate. At the beginning of the resulting exponential phase, half of the culture was
 inoculated into fresh medium again without phosphate to maintain the cells in the
 exponential phase. The remainder was kept as batch experiment. Phosphate
 concentrations in the batch two, three, four and five were below the detection limit (~ 1
 μM) of the phosphate quantification assay. Hence, we include nM-level phosphate in our
 experiment now. The key result is that, the yield of fucose per cell in cultures with lower
 phosphate content were higher than those in cultures with higher phosphate
 concentration (Figure 2f).

-----
 **Reviewer #3:** What about cell densities? Globally, are the experiments relevant to what
 cells would see in the ocean?

**Authors:** As far as we know, cell density of *Glossomastix* in the environment is not
recorded probably because it is still challenging to quantify it. In this study, the abundance
of *Glossomastix* is 10^5 - 10^6 cell mL⁻¹, in the range of phytoplankton abundance (10^4 - 10^6
cells mL⁻¹) in coastal oceans (<https://academic.oup.com/plankt/article/32/1/1/1496461>;
<https://www.sciencedirect.com/science/article/pii/0022098187901699>).

-----

**Reviewer #3:** Lines 180-182. "These results indicate bacteria can digest the fucan but
only when the environment provides enough additional resources and when the fucan is
not too concentrated." This study focused on a single bacterium and the authors have
demonstrated that there aren't a lot of cells that can digest the fucan. Maybe this one cell
was just not happy in the culture conditions present. Alternatively, thinking about the
broader bacterial community and the central story here that P limitation changes how
fucan digestion is regulated -- maybe there are just not a lot of fucan digesting bacteria
(authors present evidence for this) and P-limitation shifts the community to strains that
cannot digest fucan. This is a storyline that crops in at times in the paper, but I am not
sure which story the authors are trying to tell.

**Authors:** We thank the reviewer for the insightful feedback. We've revised the manuscript
to clarify our central storyline regarding phosphate limitation's impact on fucan digestion
on 273-286. While our study focuses on a single bacterium, *V_227*, as a model, we
emphasize that it represents a broader group of fucan-degrading bacteria. We
acknowledge the potential concern that the observed limited digestion could be due to
culture conditions, but our data consistently show that phosphate limitation reduces the
growth and activity of fucan-digesting bacteria. This revision better integrates our findings
on the single bacterium with the broader community dynamics, reinforcing the conclusion
that phosphate limitation stabilizes fucan and enhances carbon sequestration.

-----

Significance

**Reviewer #3:** The paper has some significance for understanding C cycling and algal-
bacteria interactions in the ocean, but I am not sure how much given how rare fucan
digesting bacteria are in the ocean.

**Authors:** Our manuscript presents significant insights into the marine carbon cycle,
particularly focusing on the role of phosphate limitation in enhancing algal fucan synthesis
and suppressing bacterial digestion of fucan, thereby stabilizing the carbon sequestration
pathway. The findings offer a biochemical explanation for the stability of fucan in marine
environments and have global implications for understanding the balance between algal
carbon fixation and bacterial degradation. The research not only advances our
comprehension of marine biogeochemical processes but also provides valuable
information for strategies aimed at increasing carbon dioxide removal and storage in the
ocean, which is crucial for addressing climate change.

-----

Data and methodology

**Reviewer #3:** On approach, I question the relevance of the *Glossomastix* that had been
in culture for ~10 years to ocean dynamics. It is a lab-rat. Is there a way to look at other
cultures that have been isolated more recently?

**Authors:** Fucoïdan usually was known as secreted by brown algae and diatoms. To the
best of our knowledge, no other microalgae of fucoïdan producers was reported before
our study. *Glossomastix* is one of the relatively few cultivated isolates known to produce
fucoïdan. Similar studies are on ongoing in the lab using fucoïdan-producing diatom
species of documented global abundance.

-----

**Reviewer #3:** Lines 275-280 and Extended Data Fig. 7: there are missing data. in the
third panel (residual carbohydrates), phosphate concentrations go down to 0, but in the
a, b, d, f, it only goes to 5 or 10 - where are the other data? It looks like they have been
removed and this is a big red flag to me.

**Authors:** For the Extended Data Fig. 7a-b, when the phosphate concentration was below
10 μM , the OD_{600} values of the bacteria did not conform to the logistic growth model, and
therefore the growth rate as well as the maximum biomass could not be calculated. For
Extended Data Fig. 7d-e, similar to the above, the logistic growth model does not fit at
low phosphate concentrations. When the phosphate concentration was increased to 40
μM , a large amount of precipitation appeared at the bottom of the 24-well plate and a
growth curve could not be obtained. Therefore, 40 μM and 50 μM phosphate
concentration samples were not used for subsequent analysis. We have added the
explanations for the missing data in the legend of Extended Data Fig. 9 as:

*“The absent data points of 0 μM phosphate concentration in panels a, b, d and e was*
*because the logistic growth model did not fit at low phosphate concentrations. The absent*
*data points of 40 and 50 μM phosphate concentrations in panels d, e and f was because*
*a large amount of precipitation appeared at the bottom of the 24-well plate and a growth*
*curve could not be obtained.”*

-----

**Reviewer #3:** Lines 284-285 and Extended Figures: I don't see a difference in these
slopes in 6b. Moreover in the curves, there are also decreases in the laminarian. The
difference is in the shape of the curve at 1 and 5 μM P for glycan. This requires further
exploration. I might standardize the growth relative to the maximum and then look at
differences.

**Authors:** Fig. 6b shows that biomass increases linearly with phosphate concentration.
And the two curves will intersect at a certain phosphate concentration. Here we want to
highlight the effect of phosphate concentration on biomass. It also shows that at low
phosphate concentrations, especially at 1-10 μM phosphate, there is a huge difference in
biomass due to two polysaccharides. The linear increase in biomass indicates that
phosphate in the medium is still limiting at this point.

-----

Analytical approach

**Reviewer #3:** Stats in parentheses are lacking (just naked p-values)- what was the test?
where are other metrics?

**Authors:** We thank the reviewer for pointing this out. We have supplemented the figure
legend with statistical methods.

-----

**Reviewer #3:** Parameters used on CheckM should be listed.

**Authors:** Done.

-----

**Reviewer #3:** Why is coverage for HMMER 35%?

**Authors:** This is a commonly used threshold, especially in the annotation of glycoside
hydrolases.

-----

Suggested improvements

**Reviewer #3:** I would add a set of chemostat experiments or even semi-continuous
growth experiments that show results consistent with the ones presented, so that P
limitation can be directly linked to processes described. Here, I would also take P
limitation to lower concentrations.

**Authors:** We added the semi-continuous experiment. The pre culture was grow in
medium with an inorganic phosphate source and then inoculated in medium without
phosphate. At the beginning of the resulting exponential phase half of the culture was
inoculated into fresh medium again without phosphate to maintain the cells in the
exponential phase. The remainder was kept as batch experiment. Phosphate
concentrations in the batch two, three, four and five were below the detection limit (~1
μM) of the phosphate quantification assay. Hence, we include nM-level phosphate in our
experiment now. The key result is that, the yield of fucose per cell in cultures with lower
phosphate content were higher than those in cultures with higher phosphate

concentration (Figure 2b). The related description of this experiment can be found on Line
120-132 as:

*“Moreover, we conducted semi continuous growth experiments where we increasingly*
*decreased the inorganic phosphate concentration while keeping other essential nutrients*
*at a constant concentration. The pre-culture was grown in medium with inorganic*
*phosphate and then inoculated in the same medium but without phosphate. When the*
*algae entered the exponential growth phase, half of the culture was inoculated into fresh*
*medium again without phosphate. The remainder was kept as batch experiment. During*
*the semi-continuous experiment, which included four transfers, we recorded cell numbers,*
*phosphate concentrations and the fucoïdan exudation rate. In the semi-continuous growth*
*and batch mode, after the cultivation cycle ended, the fucoïdan yield per cell was higher*
*in cultures with lower phosphate concentrations (Fig. 2f and Extended Data Fig. 3). The*
*semi-continuous growth experiment showed that when the culture stopped being diluted*
*on day 15, the fucoïdan concentration began to increase (Extended Data Fig. 3).”*

-----
**Reviewer #3:** There seem to be two storylines in the paper. The first is the bacteria can
digest fucan and P limitation changes these dynamics. The second is that P limitation
changes the community (i.e., distribution of fucan digesting bacteria, line 233). These
hypotheses need to be better spelled out in the beginning and sifted through.

**Authors:** To make the storyline clear to the reader, we wrote the Introduction part on Line
39-74 as:

*“Photosynthetic algae exude substantial quantities of fucoïdan that shows potential as*
*a carbon sink¹⁻⁴. Fucoïdan is a family of diverse polysaccharides, which share alpha-*
*configured fucose. During diatom blooms fucoïdan accumulates and forms particles, while*
*the structurally simple polysaccharide laminarin, a beta-glucan, gets rapidly degraded in*
*dissolved form². Within sinking particles fucoïdan can reach deeper waters and store*
*carbon in hundred to thousand year old sediments^{3,5}. The stability of fucoïdan has been*
*independently supported by incubation experiments. During research ship expeditions,*

*less complex polysaccharides including laminarin are degraded faster than fucoidan by*
*extant microbial communities⁶⁻⁸. Moreover, fucoidan degraders are highly specialized*
*bacteria that use hundreds of different fucoidanase to degrade fucoidan⁹. Bacteria with*
*homolog genes of those fucoidanases can be detected during algal blooms locally and*
*globally in the TaraOcean dataset⁹. The presence of fucoidanases indicates a globally*
*pervasive potential for fucoidan degradation by bacteria under certain conditions.*
*However, during algal blooms the bacterial fucoidanases are downregulated in the*
*presence of fucoidan². The bacteria primarily express laminarinases and other enzymes*
*for the degradation of simpler polysaccharides². Molecular mechanisms that decrease*
*bacterial synthesis of fucoidan-degrading enzymes and thereby promote the fucoidan*
*carbon sink remain unknown.*

*Previous research showed transporters for phosphate acquisition and enzymes for*
*laminarin degradation are highly expressed by bacteria during algal blooms¹⁰. Hence,*
*bacteria may require phosphate to degrade polysaccharides including laminarin and*
*fucoidan. It has been suggested that the accumulation of algae derived mucilage*
*polysaccharides occurs during algae blooms when phosphate becomes limiting¹¹. Under*
*these conditions the algae show reduced or absent growth (cell divisions) yet continue*
*carbon dioxide fixation into released polysaccharides. However, this accumulation also*
*requires that bacteria fail to degrade the polysaccharides that are released by algae. This*
*bacterial inability to degrade polysaccharides in response to phosphate limitation has to*
*the best of our knowledge not been tested. To investigate the role of phosphate for*
*fucoidan degradation at the molecular level we assembled a model system. As a fucoidan*
*producer we chose the microalgae *Glossomastix* sp. PLY432 isolated from the English*
*Channel¹². We purified and structurally characterized this previously unknown fucoidan*
*from *Glossomastix* sp. PLY432, the first fucoidan structure from a microalgae. We used*
*the purified fucoidan as sole carbon source to isolate a previously unknown marine*
*Verrucomicrobiaceae bacterium that was specialized on this fucoidan. Physiological*
*experiments showed that phosphate deprivation promotes fucoidan synthesis by algae*
*and inhibits fucoidan degradation by bacteria. Our results offer a new theory in the ocean*

*that phosphate deprivation of microbes could promote the fixation and sequestration of*
*carbon dioxide in the form of fucoidan.”*

-----

**Reviewer #3:** For a paper where so much rests on P limitation, Extended Data Fig 2
should be a main figure.

**Authors:** In the new version, as suggested by the reviewer, we added the semi-
continuous experiment (Figure 2f) and the results supporting our opinion that phosphate
deprivation promotes fucoidan exudation. So, we still keep Extended Data Fig 2 in the
Extended data.

-----

**Reviewer #3:** I would like to see some discussion of B-glycerol phosphate as a phosphate
source for this algae strain.

**Authors:** In Fig. 2a and 2d, at the same phosphate concentration, β -glycerol phosphate,
as a source of phosphorus, resulted in a maximum biomass of algae that was
approximately four times that of inorganic phosphate. We don't know if this huge
difference comes from phosphorus or glycerol. But these two sources of phosphorus give
a consistent phenomenon, with *Glossomastix* synthesizing fucan in large quantities
during the stationary phase (Fig. 2a and 2d).

-----

**Reviewer #3:** Lines 295-299: this is well articulated, I might use it in other sections, to
provide background/context/thread throughout.

**Authors:** Thanks for this comment. We have intergraded this part into Introduction. The
new text can be found on Line 39-54 as:

*“Photosynthetic algae exude substantial quantities of fucoidan that shows potential as a*
*carbon sink¹⁻⁴. Fucoidan is a family of diverse polysaccharides, which share alpha-*
*configured fucose. During diatom blooms fucoidan accumulates and forms particles, while*
*the structurally simple polysaccharide laminarin, a beta-glucan, gets rapidly degraded in*

*dissolved form*². *Within sinking particles fucoïdan can reach deeper waters and store*
*carbon in hundred to thousand year old sediments*^{3,5}. *The stability of fucoïdan has been*
*independently supported by incubation experiments. During research ship expeditions,*
*less complex polysaccharides including laminarin are degraded faster than fucoïdan by*
*extant microbial communities*⁶⁻⁸. *Moreover, fucoïdan degraders are highly specialized*
*bacteria that use hundreds of different fucoïdanase to degrade fucoïdan*⁹. *Bacteria with*
*homolog genes of those fucoïdanases can be detected during algal blooms locally and*
*globally in the TaraOcean dataset*⁹. *The presence of fucoïdanases indicates a globally*
*pervasive potential for fucoïdan degradation by bacteria under certain conditions.*
*However, during algal blooms the bacterial fucoïdanases are downregulated in the*
*presence of fucoïdan*². *The bacteria primarily express laminarinases and other enzymes*
*for the degradation of simpler polysaccharides*². *Molecular mechanisms that decrease*
*bacterial synthesis of fucoïdan-degrading enzymes and thereby promote the fucoïdan*
*carbon sink remain unknown.”*

-----

**Reviewer #3:** There are a number of places in the text that don't read well or have tense
issues (e.g., line 86, "Not growing algae continue synthesis of fucan under nutrient limiting
conditions.", line 101, "The data shows a sustained population of algal cells sustains the
fucan sequestration pathway."). The manuscript deserves another sentence level read for
clarity.

**Authors:** We recognize the problems with readability arising from unclear sentences and
wording. Thorough revisions have been made to the language.

-----

Clarity and context

**Reviewer #3:** The section between 253 and 263 needs more clarity and context. The
sentence 253-254 is cited, but the citation is about the first part of the sentence, not
anything to do with algal blooms. All we know is that Pst ABC system has an affinity
constant 200 - 400 nm. Experiments were done at 1 µM, which is more than 2X this affinity
constant. Lines 255-257 -- These all seem to be different papers. I also don't understand

how disparate references can be linked to make a conclusion here. If that is not true,
more context is needed in the text. Lines 260-263, I am missing this link. What I know
here is that these bacteria experience P limitation, they have evolved transporters to get
very low amounts of P, how is that related to glycan here?

**Authors:** In order to address this comment, we propose this revised paragraph on lines
273 to 286 of the new revised manuscript: “*In other ecosystems, bacteria such as*
*Bacteroides thetaiotaomicron*⁴⁷ *and Clostridium perfringens*⁴⁸ *that degrade recalcitrant*
*mucilage polysaccharides use high affinity phosphate transporters of the ABC-type*
*(Supplementary Table 8). This high affinity required to bind phosphate indicates that*
*phosphate is an important but limited resource for which bacteria must compete. Hence,*
*phosphate may be important for and influence the degradation of recalcitrant*
*polysaccharides. During algal blooms in the ocean, we previously found that bacteria*
*express the ABC transporter and other proteins used for phosphate and phosphonate*
*uptake*¹⁰. *They also express enzymes for the degradation and metabolism of laminarin*
*such as GH16 endo-beta-1,3 glucanases. However, the fucoidan degrading bacteria*
*showed downregulation or absence of fucoidanase expression*². *This absence of*
*degradation was further supported by the concentration of fucoidan increasing over*
*several month*². *Meanwhile, laminarin was effectively consumed by bacteria despite the*
*limited phosphate concentration*^{2,23}. *Conclusively, the availability of phosphate might*
*especially constrain the ability of bacteria specialized in the degradation of complex*
*glycans such as fucoidan.*”

-----

**Point by point response to the comments of referees in relation to the manuscript**
**NMICROBIOL-24041026A: “Phosphate deprivation restricts the degradation of**
**fucoidan by a Verrucomicrobium”**

**Summary of Reviewers’ Comments and Our Response**

We received comments from three reviewers. Two of them approved the manuscript for
publication, while the other one recommended it for publication pending minor corrections.
These issues have now been addressed in this revised version.

We sincerely thank the Editors and Reviewers for the time and effort they devoted to
evaluating our manuscript. We are grateful for their constructive comments, which we
have carefully addressed in our revisions. All suggested improvements have been
incorporated, and we believe these changes have significantly enhanced the overall
quality of the manuscript.

-----

**Responses to Reviewer #1**

-----

Reviewer #1 (Remarks to the Author):

**Reviewer #1:** Authors have addressed all concerns. Congratulations on a great piece of
work!

**Authors:** We appreciate the reviewer's positive assessment and are pleased that all
concerns have been satisfactorily addressed.

-----

**Responses to Reviewer #2**

-----

Reviewer #2 (Remarks to the Author):

**Reviewer #2:** The authors have done a good job addressing the major concerns that I
had in the first submitted version of this paper. The addition of the semi-continuous growth
experiment was particularly important in strengthening the manuscript. While there are
still some minor revisions that are needed to improve the clarity of this work, I feel that
authors have substantially improved this manuscript and I am happy to recommend it for
publication after these minor errors are addressed.

**Authors:** We thank the reviewer for this very positive evaluation.

-----

General Concerns:

**Reviewer #2:** I was surprised that the V_227 bacteria only drew down phosphate to ~1
μM in the growth yield experiments (Extended figures 10-11), as this level of phosphate
is still considered replete and would not typically be low enough to trigger the pho regulons
that control high affinity ABC phosphate transporters. It might be worth clarifying if this is
a result of experimental methodology and design, or if V_227 does not draw down
phosphate below ~1 μM . If more significant drawdown did occur, consider changing the
y-axis scale to make it easier to interpret the figure.

**Authors:** Thank the reviewer for pointing this out that V_227 bacteria only drew down
phosphate to ~1 μM in the growth yield experiments (Extended figures 10-11). That is
because of the limitation of phosphate quantification assay, which has a lowest detection
limit of approximately 1 μM . We have made it clear in the Method section of the main text
on Line 413-414 as:

*“This assay has a detection limit of approximately 1 μM , below which phosphate*
*concentrations cannot be accurately quantified.”*

-----

General Comments:

-----

**Reviewer #2:** Ln 141: There should be a comma after diatoms

-----

**Authors:** It has been revised.

-----

**Reviewer #2:** Ln 193: “Enrichments” should not be plural

-----

**Authors:** It has been corrected.

-----

**Reviewer #2:** Ln 209: there should be an “and” between rumen and ocean

-----

**Authors:** It has been revised.

-----

**Reviewer #2:** The paragraph beginning at Ln 212: This paragraph was hard to follow. I
suggest reviewing it for clarity and restructuring it so it is clear to the readers which
“fucoidan” is being discussed.

-----

**Authors:** To shorten our manuscript to 3500 words, we have to remove this paragraph in
this version.

-----

**Reviewer #2:** Ln 241 & Ln 382: Consider also citing Granzow and Repeta (2024) ES&T,
which also found acylpolysaccharides make up ~ 20 μm of DOC in the surface ocean.

-----

**Authors:** Thanks to the reviewer for referring this paper, which found acylpolysaccharides
contribute to ~20 μm of DOC in the surface ocean. We have added this citation.

-----

**Reviewer #2:** Ln 246: While P is often limiting in the Mediterranean and Sargasso sea,
many other oligotrophic regions of the ocean are primarily N limited and only secondarily

limited by P (See Moore et al 2013 Nat. Geosci). I think you can keep this sentence largely
unchanged, but I would temper the statement to be more precise.

**Authors:** We have tempered this sentence on Line 204-207 as:

*“Especially in oligotrophic regions of the surface ocean with a phosphate concentration*
*of 0-300 nM³⁷⁻³⁹, and during later phases of coastal algal blooms, phosphate can play an*
*important role in restricting bacterial growth⁴⁰, although the degree of phosphate limitation*
*varies among systems.”*

-----

**Reviewer #2:** Ln 247-251: The idea about Verrucomicrobiota becoming more abundant
on sinking particles due to increased dissolved phosphate is interesting but not well
described here. Consider rewriting or removing, as the dynamics between dissolved
nutrient concentrations and microbial succession on sinking particles is complicated and
deserves more nuance than described here.

**Authors:** We agree with the reviewer that the dynamics between dissolved nutrient
concentrations and microbial succession on sinking particles is complicated. We have
removed this part as suggested.

-----

**Reviewer #2:** Ln 277: Consider changing to “may be important for and may influence” to
improve readability

**Authors:** We have revised this.

-----

**Reviewer #2:** Ln 293: What does “low” mean in this sentence?

**Authors:** It was an error. We have removed ‘low’ from this sentence.

-----

**Reviewer #2:** Ln 313: What linear model are you referring to? It’s not clear in the text.

**Authors:** Linear relationship between optical density and cell abundance of V_227. We
have added the reference “Supplementary Table 8” in this sentence to make it clear.

-----

**Reviewer #2:** Ln 345-346: I’m not sure what this sentence is saying. Please revise.

**Authors:** We have revised this sentence on Line 282-284 as:

*“Given that more laminarin is required compared to fucoïdan to build similar quantities of*
*bacterial biomass, some of the carbon energy of laminarin is lost.”*

-----

**Reviewer #2:** Ln 347: What do you mean by “Without knowing what we lost?” is this in
reference to monosaccharide content?

**Authors:** To make it clear, we have revised this sentence on Line 284-286 as:

*“The TCA cycle rapidly extracts energy and releases carbon dioxide from glucose, which*
*may account for the lost carbon. Compared to laminarin, fucoïdan is twice as stable and*
*therefore of superior quality to store carbon.”*

-----

**Reviewer #2:** Ln 352: You use the term “stability index” which would typically imply a
formula that can be used to calculate the stability of a glycan, but no formula is given.
Consider changing to a different term, such as relationship or characteristic.

**Authors:** We have changed the ‘index’ to ‘ratio’ to make it more accurate.

-----

**Reviewer #2:** Ln 355-356: I am not sure what you mean by this sentence. Do you mean
that algae have evolved to increase their glycan structural complexity as a survival
strategy? Or do you mean that complex glycans are synthesized on the outer membrane
of algae?.

**Authors:** We meant both. We have revised this sentence on Line 296-299 as:

*“Complex glycans are synthesized on the outer membrane of eukaryotic cells where they*
*form the extracellular matrix. Algae and other eukaryotes have evolved to increase their*
*glycan structural complexity as a survival strategy in diverse environments, including*
*oceans, soils, intestines, and rhizosphere.”*

-----

**Reviewer #2:** Ln 382-383: The 2nd clause in this sentence makes it difficult to follow.
Consider rephrasing.

**Authors:** We have rephrased the sentence on Line 324-325 as:

*“These glycans hold 20% (~ 20 μ M) of the dissolved organic carbon in this system, which*
*is largely phosphate deprived (0-300 nM)³³⁻³⁶.”*

-----

**Reviewer #2:** Ln 391: The term “carbon energy compounds” is confusing. Consider
removing “energy” from the sentence.

**Authors:** We have removed “energy” in this sentence as suggested.

-----

**Reviewer #2:** Ln 392: Metabolites are not inherently labile or easy to consume. Fucoidan
is a metabolite. Picrotoxins are metabolites. You should be more specific about which
metabolites you are talking about in this sentence.

**Authors:** To make it more clear and precise, we revised this sentence on Line 332-333
as:

*“In contrast, bacteria that consume structurally more labile carbon compounds⁵⁶ would*
*rapidly consume the essential nutrients.”*

-----

**Reviewer #2:** Ln 398: “Contributes” should be “contribute”.

**Authors:** Revised.

-----

**Reviewer #2:** Ln 413-414: There should be commas after “show” and “taxa”.

**Authors:** Done.

-----

**Responses to Reviewer #3**

-----

Reviewer #3 (Remarks to the Author):

**Reviewer #3:** I approve the publication of this work as is.

**Authors:** We sincerely thank the reviewer for the approval.

-----